# Evaluation of Gut Microbiota Stability and Flexibility as a Response to Seasonal Variation in the Wild François' Langurs (*Trachypithecus francoisi*) in Limestone Forest

Hongying Liu,[a,b,c] Yuhui Li,[a,b,c] Jipeng Liang,[d] Dengpan Nong,[d] Youbang Li,[a,b,c] ⏺Zhonghao Huang[a,b,c]

[a]Key Laboratory of Ecology of Rare and Endangered Species and Environmental Protection (Guangxi Normal University), Ministry of Education, Guilin, China
[b]Guangxi Key Laboratory of Rare and Endangered Animal Ecology, Guangxi Normal University, Guilin, China
[c]College of Life Sciences, Guangxi Normal University, Guilin, China
[d]Administration Center of Guangxi Chongzuo White-Headed Langur National Nature Reserve, Chongzuo, China

**ABSTRACT** The coevolution between gut microbiota and the host markedly influences the digestive strategies of animals to cope with changes in food sources. We have explored the compositional structure and seasonal variation in the gut microbiota of François' langur in a limestone forest in Guangxi, southwest China, using 16S rRNA sequencing. Our results demonstrated that *Firmicutes* and *Bacteroidetes* were the dominant phyla in langurs, followed by *Oscillospiraceae*, *Christensenellaceae*, and *Lachnospiraceae* at the family level. The top five dominant phyla did not show significant seasonal variations, and only 21 bacterial taxa differed at the family level, indicating stability in gut the microbiota possibly with respect to foraging for several dominant plants and high-leaf feeding by the langurs. Moreover, rainfall and minimum humidity are important factors affecting the gut microbiota of the langurs, but they explain few changes in bacterial taxa. The activity budget and thyroid hormone levels of the langurs did not differ significantly between seasons, indicating that these langurs did not respond to seasonal changes in food by regulating behavior or reducing metabolism. The present study indicates that the gut microbiota's structure is related to digestion and energy absorption of these langurs, providing new perspectives on their adaptation to limestone forests.

**IMPORTANCE** François' langur is a primate that particularly lives in karst regions. The adaptation of wild animals to karst habitats has been a hot topic in behavioral ecology and conservation biology. In this study, gut microbiota, behavior, and thyroid hormone data were integrated to understand the interaction of the langurs and limestone forests from the physiological response, providing basic data for assessing the adaptation of the langurs to the habitats. The responses of the langurs to environmental changes were explored from the seasonal variations in gut microbiota, which would help to further understand the adaptive strategies of species to environmental changes.

**KEYWORDS** gut microbiota, stability, flexibility, François' langur, limestone forest

The microbiota that colonize the gastrointestinal tract of animals are closely related to the host and regulate digestion and nutrient absorption (1, 2), maintain immune homeostasis (3), and regulate host behavior (4), enabling the host to potentially adapt to environmental changes (1, 2, 5). Owing to mammals' inability to produce enzymes that break down cellulose, they rely on the gut microbiota to degrade foods (5–7). *Ruminococcaceae* and *Lachnospiraceae* frequently aid in fermenting cellulose and hemicellulose to produce short-chain fatty acids, thereby providing energy to the host (8, 9). The gut microbiota of the black howler monkey (*Alouatta pigra*) produces large amounts of volatile fatty acids to compensate for energy deficits during food shortages, even without involving behavioral adjustment (10). The evaluation of the gut microbiota has become an important approach

Address correspondence to Zhonghao Huang, hzh773@126.com.

The authors declare no conflict of interest.

in reflecting the physiology of the host and understanding the adaptative potential of species (1, 5).

Exploring the influence of external factors on the gut microbiota and the balance mechanism could assist in understanding the interaction between the host and the gut microbiota (11, 12). The gut microbiota community is dynamic and relatively stable (13, 14). The resilience of the gut microbiota community serves as a balancer in response to disturbances by forming new homeostasis (13, 14). Moreover, resilience occurs more frequently with changes in the abundance of bacterial taxa, with stability demonstrated over a longer period by the dominant taxa that make up the core microbiota community (13, 15). Host genetics and diet are predominant factors influencing gut microbiota, even driving niche differentiation of hosts (1, 12). Host genetics are considered the most significant factors in shaping the gut microbiota, given the large foregut or hindgut of phytophagous animals, which provides space for the bacteria to ferment plant fibers adequately (1, 16). Moreover, there have been reports that the ruminants yak (*Bos grunniens*) and Tibetan sheep (*Ovis aries*) share similar gut microbial communities (17). However, the gut microbiota of animals with similar genetics, even different geographical populations of the same species, differ significantly in their gut microbiota owing to host diets that are shaped by ecological factors (18).

Diet has been reported to play a determinant role in shaping the gut microbiota (19, 20), particularly in driving the coevolution of the host with the gut microbiota (6), with even greater influence than genetics in shaping the microbial composition of animals, as demonstrated in the colobine monkey species (21). The food resources of wild animals exhibit seasonal shifts and are influenced by climatic factors such as rainfall (22) and temperature (11). These adjustments in ecological responses are related to the corresponding changes in environmental selection pressures (10, 22, 23). Furthermore, white-headed black langurs (*Trachypithecus leucocephalus*) increase foraging and decrease movement to conserve energy to adapt to food patches (24), while rhesus macaques (*Macaca mulatta*) respond to resource scarcity and seasonal changes in high-altitude habitats by increasing plant items and using fallback foods (25). However, current studies predominantly focus on the behavioral strategies of the animals in response to spatial-temporal shifts in foods. Behavioral strategies alone are not enough to fully reflect the adaptation mechanisms of wildlife, and the changing regularity of the gut microbiota allows probing of the host-specific response to the habitat at a deeper level. The black howler monkeys respond to seasonal variations in food by changing the structure and function of the gut microbiota without activity adjustment (10), and volatile fatty acids metabolized by *Ruminococcaceae* compensate for the energy deficit during the dry seasons (10). Similarly, white-faced capuchins (*Cebus imitator*) (26), geladas (*Theropithecus gelada*) (22), and Tibetan macaques (*Macaca thibetana*) (9) also exhibit seasonal variations in their gut microbiota.

The François' langurs (*Trachypithecus francoisi*) are included in the IUCN Red List of Threatened Species (27). These langurs in Guangxi, China, are facing habitat challenges because of insufficient water sources, patchy foraging sites, and seasonal variations in food (28). They exhibit behavioral strategies, such as increased foraging time when food is reduced, alleviation of energy deficits (29), and increased intake of mature leaves and seeds, to accommodate environmental changes (30). These behavioral strategies are adopted in response to seasonal fluctuations in foods, which are correlated with the host's gut microbiota (9, 26, 31). Preliminary studies suggest that the gut microbiota of wild François' langurs predominantly comprise *Firmicutes* and *Verrucomicrobia* (32), while that of captive individuals is enriched in bacteria taxa such as *Clostridiaceae* and *Lachnospiraceae*, which assist in digesting plant fibers (21).

However, data on the seasonal changes in the gut microbiota of François' langurs are still unavailable. In this study, we have described the diversity and composition of the gut microbiota based on 152 feces samples of wild François' langurs sampled over 16 months, and then the influences of ecological factors on the gut microbiota were studied. We have also recorded the behavioral adjustment of these langurs in response to the changes in seasons, using activity budget data (29). Furthermore, we have measured the thyroid hormones to evaluate the energy metabolism of the langurs. Finally,

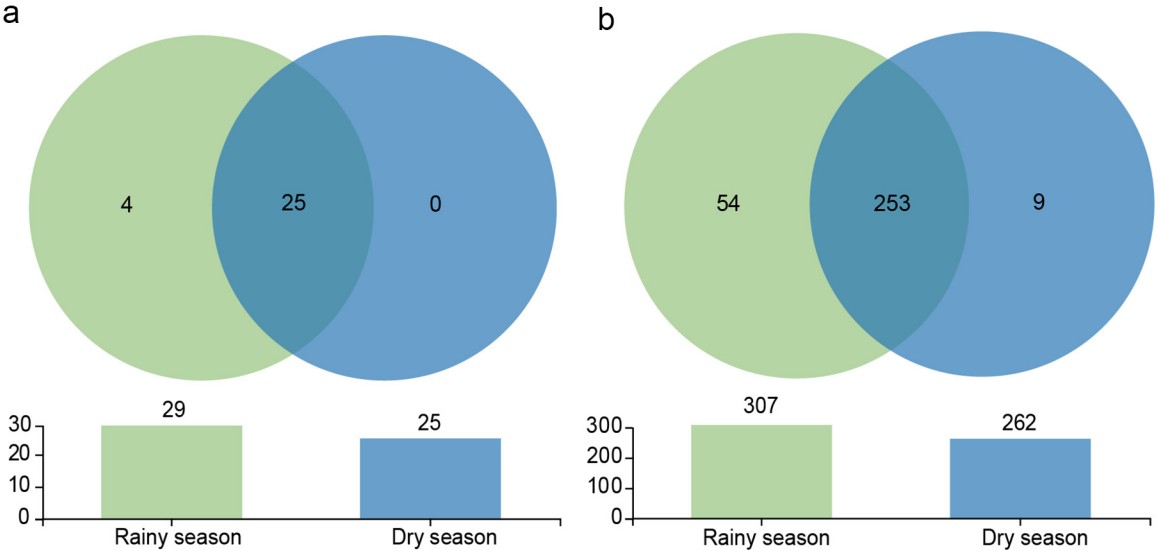

**FIG 1** Number of shared and specific taxa in the gut microbiota of François langurs at the phylum level (a) and family level (b).

we discussed the importance of the gut microbiota in responses to the ecological seasonality of the François' langurs by integrating data on the gut microbiota, thyroid hormones, and activity budgets.

## RESULTS

**Community composition in the gut microbiota of François' langurs.** The gut microbiota in the feces was clustered to 2,894 operational taxonomic units (OTUs) and was divided into 29 phyla, 70 classes, 175 orders, 316 families, 664 genera, and 1,041 species. Specifically, the unique gut microbiota in the rainy season included 4 phyla and 54 families, and 9 taxa at the family level were distinctive during the dry season. In both seasons, the bacterial taxa comprised 25 phyla and 253 families (Fig. 1).

At the phylum level, the microbial taxa of the langurs were dominated by *Firmicutes* (75.76% ± 11.84%), *Bacteroidetes* (10.29% ± 8.50%), and *Verrucomicrobia* (6.90% ± 6.00%) (see Table S4 in the supplemental material). At the family level, the top five dominant bacterial taxa were *Oscillospiraceae* (17.74% ± 6.86%), *Christensenellaceae* (15.22% ± 6.69%), *Lachnospiraceae* (9.27% ± 4.22%), *Akkermansiaceae* (6.77% ± 6.08%), and *Ruminococcaceae* (6.47% ± 9.15%) (Fig. 2; Table S5).

**Seasonal variations in community composition.** The top five bacterial taxa (*Firmicutes*, *Bacteroidetes*, *Verrucomicrobia*, *Actinobacteria*, and *Proteobacteria*) did not show significant seasonal variations at the phylum level, except *Cyanobacteria*, whose relative abundance was higher in the rainy season than in the dry season (Fig. 3a; Table S4). At the family level, only 21 bacterial taxa were distinctly different, such as the *Christensenellaceae* and the *Eubacterium coprostanoligenes* group, which were enriched in the rainy season. Whereas *Ruminococcaceae* increased during the dry season, most of them had a relative abundance of less than 1% (Fig. 3b; Table S5). We further identified representative bacterial taxa from the two seasons using a random forest model (area under the curve [AUC], 0.82), which revealed significant seasonal differences mostly in some low abundance taxa, such as *norank_o_ Izemoplasmatales* and the *norank_o_Clostridia_vadinBB60* group (Fig. 4).

**Seasonal differences in gut microbiota diversity.** The results of the generalized linear mixed model (GLMM) demonstrated that there was no significant difference in the Shannon and invsimpson indices between seasons (Shannon: $\chi^2 = 1.281$, df = 1, $P = 0.258$; invsimpson: $\chi^2 = 1.451$, df = 1, $P = 0.228$), and the Chao index did not show any significant change (Chao: $\chi^2 = 3.191$, df = 1, $P = 0.074$). Only the abundance-based coverage estimator (ACE) index in the rainy season was higher than that in the dry season (ACE: $\chi^2 = 5.236$, df = 1, $P = 0.022$) (Fig. 5; Tables S6 and S7).

The principal-coordinate analysis (PCoA) demonstrated seasonal changes in the beta

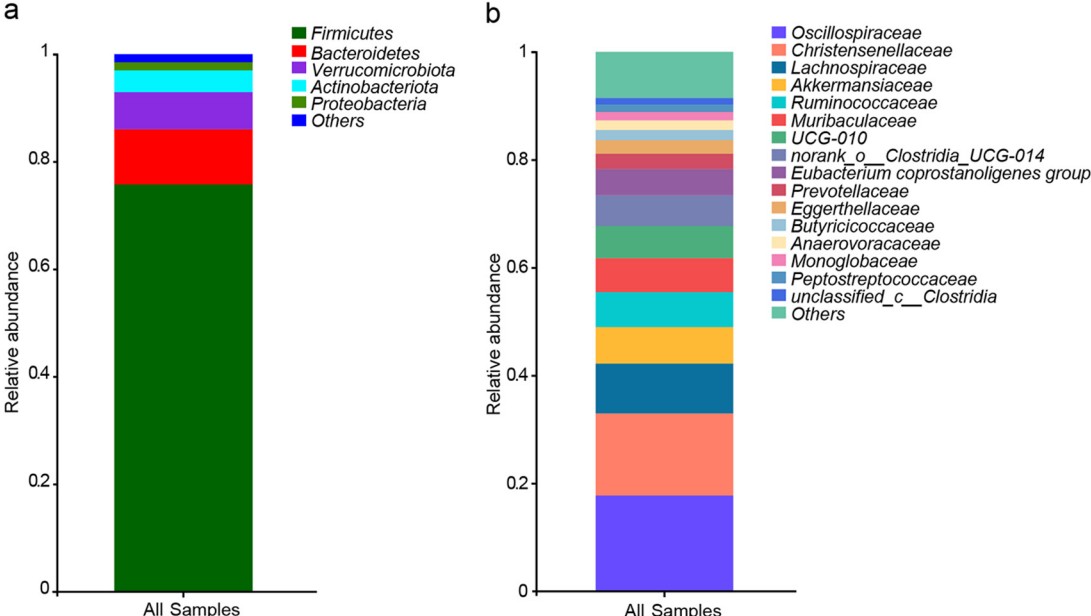

**FIG 2** Community composition of gut microbiota at the phylum level (a) and family level (b) in François langurs, all taxa with a relative abundance of less than 1% were classified as "others".

diversity of gut microbiota. At the phylum level, the structure of the gut microbiota did not show distinct differences in unweighted UniFrac distance ($R^2 = 0.012$, $P = 0.143$) or weighted UniFrac distance ($R^2 = 0.013$, $P = 0.115$) (Fig. S2a). At the family level, the gut microbiota structure showed significant seasonal differences in both unweighted UniFrac distance ($R^2 = 0.025$, $P = 0.001$) and weighted UniFrac distance ($R^2 = 0.032$, $P = 0.001$) (Fig. S2b), but $R^2$ showed that seasonal grouping had a low explanation for the differences.

**Effects of climate factors on gut microbiota.** The remaining climate factors were used for subsequent analysis (variance inflation factor [VIF] $< 10$) after removing the maximum temperature by VIF (Table S8). At the phylum level, redundancy analysis (RDA) indicated that explanatory power accumulated to 13.11% for axis 1 and axis 2, with minimum humidity having the greatest impact on seasonal differences in gut microbiota. A Monte Carlo permutation test showed that only relative humidity ($R^2 = 0.068$, $P = 0.003$) and minimum humidity ($R^2 = 0.136$, $P = 0.001$) had a significant effect on the differences (Fig. S3a, Table S9). At the family level, the two axes contributed to the explanatory power of 16.74%, with minimum humidity explaining the most variation. Further testing revealed that only sunshine duration has no significant effect on gut microbiota ($R^2 = 0.089$, $P = 0.094$). Moreover, the minimum humidity has the greatest impact ($R^2 = 0.466$, $P = 0.001$), followed by relative humidity ($R^2 = 0.274$, $P = 0.001$) and mean temperature ($R^2 = 0.241$, $P = 0.001$), and the lowest was rainfall ($R^2 = 0.138$, $P = 0.001$) (Fig. S3b, Table S9).

The variant partitioning analysis (VPA) indicated that climatic factors had less influence on gut microbiota community structure differences. At the phylum level, humidity alone explained the largest proportion of the changes in community structure. At the same time, sunshine duration accounted for the lowest proportion (0.36%), with low cointerpretation of these factor combinations. The remaining 88.61% of variations could not be explained by these climatic factors alone or together (Fig. S4a). At the family level, the highest explanation for structural differences was rainfall (8.93%) and the lowest was sunlight duration (0.41%). These climatic factors could not explain the remaining 80.83% of differences (Fig. S4b).

The heat map demonstrates correlations between these climatic factors and bacterial taxa (Fig. S5). There was a significant positive correlation between the abundance of *Firmicutes* and mean temperature ($r = 0.240$, $P = 0.003$, $n = 152$), relative humidity ($r = 0.264$, $P = 0.001$, $n = 152$), and minimum humidity ($r = 0.249$, $P = 0.002$, $n = 152$). The abundance of *Bacteroidetes* was negatively associated with the mean temperature ($r = -0.193$, $P = 0.017$, $n = 152$), rainfall ($r = -0.216$, $P = 0.007$, $n = 152$), relative humidity ($r = -0.440$,

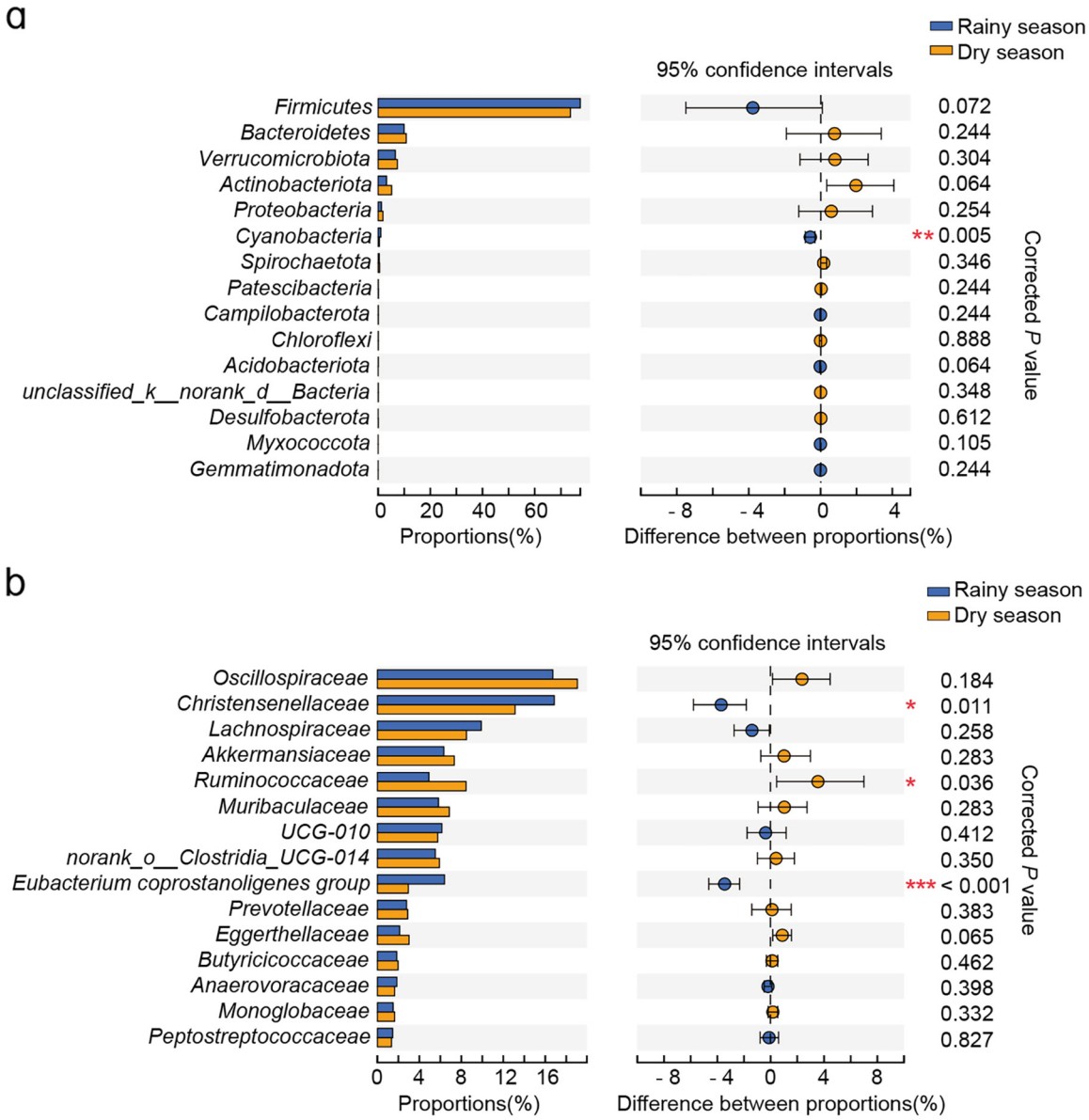

**FIG 3** Seasonal variations in the relative abundances of the gut microbiota. The differential results of the top 15 bacterial taxa at the phylum level (a) and family level (b). The graph on the left indicates the relative abundances of the bacteria taxa, and the graph on the right represents the differential results. Asterisks indicate a significant difference; *, $P < 0.05$; **, $P < 0.01$; ***, $P < 0.001$.

$P < 0.001$, $n = 152$), and minimum humidity ($r = -0.441$, $P < 0.001$, $n = 152$) but was positively associated with sunshine duration ($r = 0.208$, $P = 0.010$, $n = 152$) (Fig. S5a, Table S10). At the family level, the abundance of *Oscillospiraceae* was negatively correlated with mean temperature ($r = -0.325$, $P < 0.001$, $n = 152$), sunshine duration ($r = -0.172$, $P = 0.034$, $n = 152$), and minimum humidity ($r = -0.343$, $P < 0.001$, $n = 152$); the abundance of *Christensenellaceae* was positively correlated with mean temperature ($r = 0.286$, $P < 0.001$, $n = 152$), rainfall ($r = 0.327$, $P < 0.001$, $n = 152$), relative humidity ($r = 0.226$, $P = 0.005$, $n = 152$), and minimum humidity ($r = 0.424$, $P < 0.001$, $n = 152$) (Fig. S5b, Table S10).

**The relationship between thyroid hormones and the gut microbiota.** The concentrations of triiodothyronine (T3) and tetraiodothyronine (T4) in the langurs varied between months, with the highest value of T3 (21.676 ng/g) in July and the lowest in April (15.454 ng/g); meanwhile, the highest T4 (39.201 ng/g) was recorded in November, and the lowest was in July (28.996 ng/g) (Fig. 6). However, the GLMM results indicated that

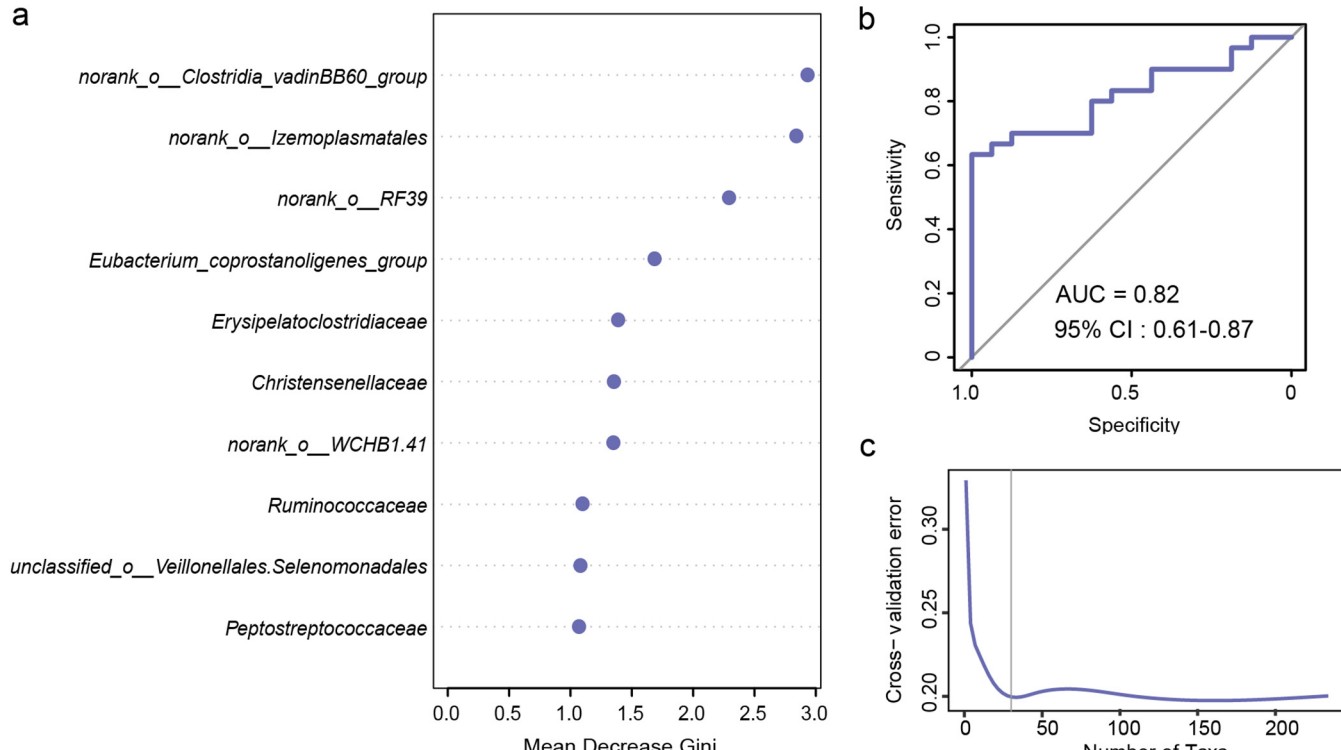

**FIG 4** Random forest model of representative bacterial taxa in seasonal groupings at the family level in the gut microbiota of François langurs (a), the ROC curve showing the accuracy of the random forest model in predicting a classification (b), and the OTU numbers corresponding to model accuracy (c).

there was no significant seasonal variation in T3 and T4 (T3: $\chi^2 = 2.765$, $P = 0.096$; T4: $\chi^2 = 0.089$, $P = 0.766$) (Tables S11 and S12). The random forest models revealed that the relative abundances of the vast majority of bacterial taxa were not significantly associated with T3 or T4, except that T3 was related to *Bdellovibrionota* and *Enterobacteriaceae* (Fig. 7 and 8; Fig. S6 to S8). The model constructed with four taxa at the phylum level explained 71.74% of the variation in T3, and three taxa at the family level had an interpretability of 59.35%. Further, T4 variation could be explained by two taxa at the phylum level by 31.15% and by four taxa at the family level by less than 26.02% (Fig. 7 and 8; Table S13).

**Activity budget.** The langurs spent 65.99% ± 5.44% of the day resting, 26.21% ± 3.89% moving, 4.12% ± 2.19% feeding, 2.67% ± 1.18% playing, and 1.01% ± 0.48% grooming each month on an average (Fig. 9). However, there were no significant differences in the activity budget during different seasons (resting: $\chi^2 = 0.013$, $P = 0.908$; moving: $\chi^2 = 0.099$, $P = 0.753$; feeding: $\chi^2 = 0.505$, $P = 0.477$; grooming: $\chi^2 = 0.560$, $P = 0.454$; playing: $\chi^2 = 2.583$, $P = 0.108$) (Table S14 and S15).

## DISCUSSION

**Gut microbiota composition.** *Firmicutes* and *Bacteroidetes* are the dominant bacterial taxa of the François' langurs at the phylum level, which is consistent with earlier studies of other primates (21, 32), including white-headed black langurs (33), Guizhou snub-nosed monkeys (*Rhinopithecus brelichi*) (23), and geladas (22). *Firmicutes* and *Bacteroidetes* are commonly found in the gut microbiota of mammals and are considered to provide the host with a strong digestive capacity (2, 34). Specifically, the *Firmicutes* can produce a variety of enzymes to help break down complex polysaccharides in foods in close association with the digestion and absorption of the host (2, 34, 35). *Bacteroidetes* can play a key role in degrading proteins and complex carbohydrates in foods (7, 34). These bacteria promote the degradation and absorption of ingested foods, and their high aggregation may be closely correlated to the langurs' diets (19, 31). In particular, the high enrichment of *Firmicutes* could be a response to diets rich in plants. More than 70% of these langurs diets consists of leaves (28). *Firmicutes* ferment substances such as cellulose and pectin to deal

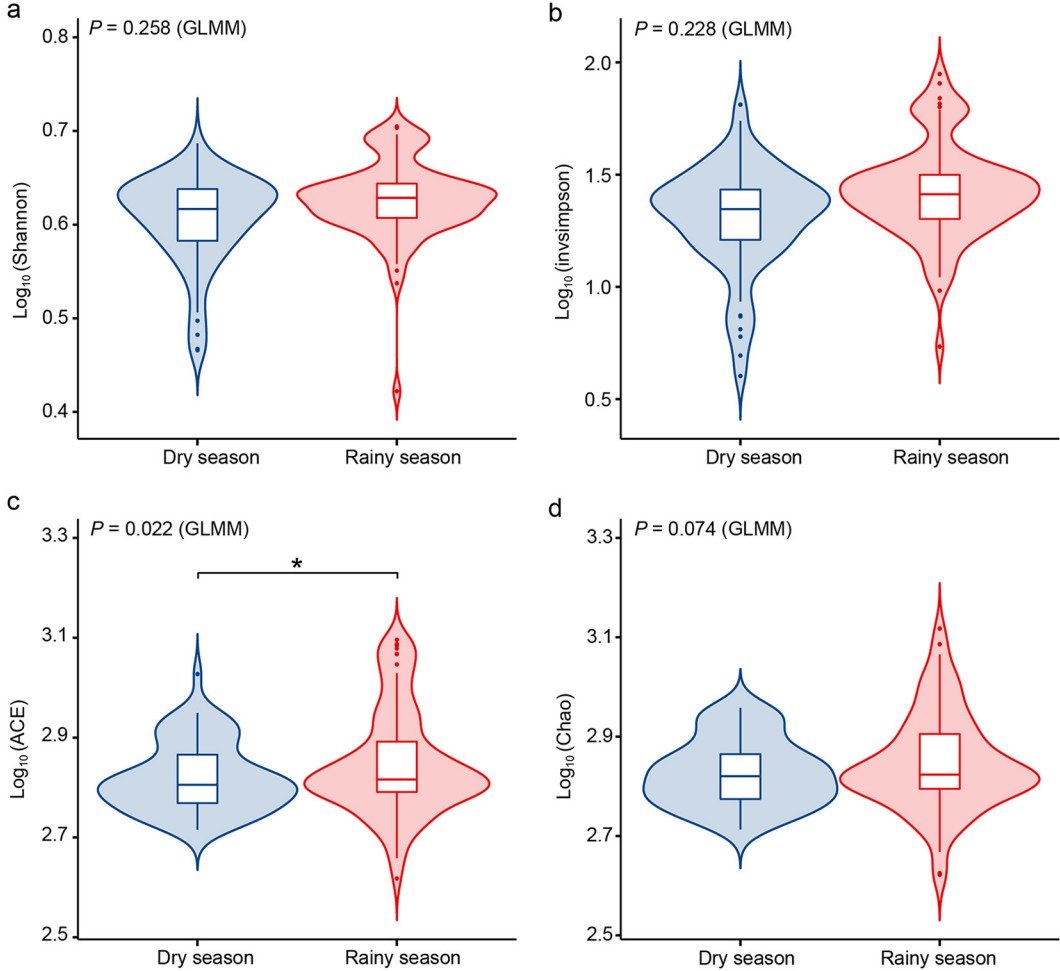

**FIG 5** Seasonal variations in the gut microbiota alpha diversity index based on GLMM. Asterisks indicate a significant difference; *, $P < 0.05$. (a) Shannon index; (b) invsimpson index; (c) ACE index; (d) Chao index.

with high-fiber foods, consequently revealing a high relative abundance (23, 35). This may be the digestion strategy of the François' langur in response to the leaf-rich diet, which is the same as that of the karst-dwelling white-headed black langurs (33).

Furthermore, 316 bacterial taxa were found in the gut microbiota at the family level, and the relative abundances of the top 15 families were greater than 90%, indicating that the gut microbiota of the François' langurs was composed of a small number of highly abundant bacterial taxa. This is similar to the human gut microbiota (15, 36). Indeed, the dominant species can support the system in a stable ecosystem that is effectively resistant to environmental disturbance (14, 37). Among these dominant bacterial taxa, the gut microbiota of the langurs were dominated by *Oscillospiraceae*, *Christensenellaceae*, *Lachnospiraceae*, and *Ruminococcaceae*, which can efficiently break down various complex fibers and provide energy to the host (8, 18). This may be related to the high-leaf diet of the langurs. The François' langur is a typical foregut-fermenting primate (16); their enlarged stomach provides the physiological structure for fermentation of fiber-rich foods by these bacterial taxa, which can help degrade polysaccharides (18) and mucins (38) and other substances. The enrichment of the bacterial taxa may respond to the high-fiber and low-quality diet of the langurs, which may be beneficial to enhance their adaptability to the limestone forest.

**Gut microbiota stability and flexibility.** There were no significant seasonal differences in the relative abundance of most bacterial taxa in the gut microbiota of the François' langurs, suggesting that their gut microbiota is relatively stable across seasons. Moreover, the stability of the gut microbiota was reported earlier in the humans

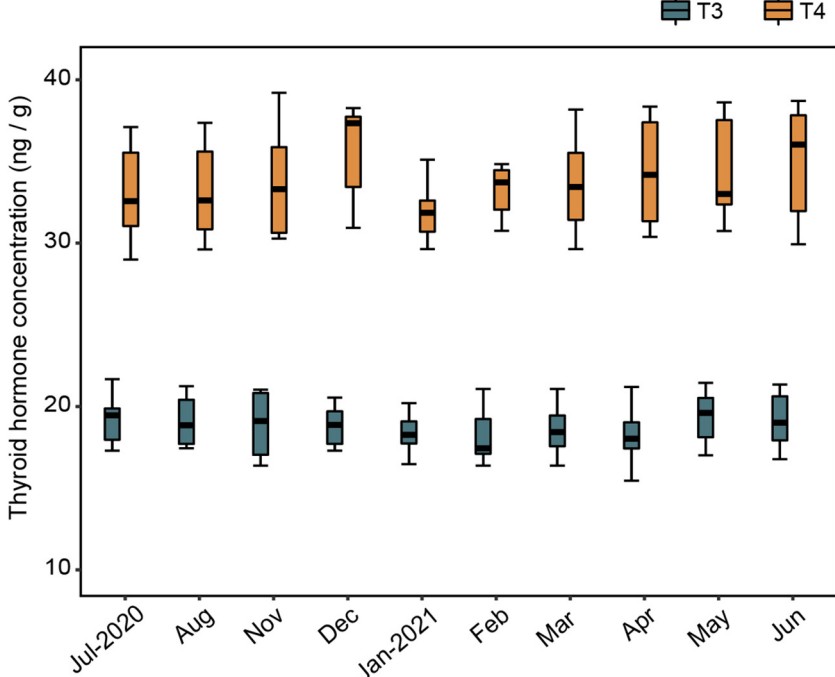

**FIG 6** Monthly variations in triiodothyronine (T3) and tetraiodothyronine (T4) of François' langurs.

(15, 36), and the human gut microbiota remains stable across longer durations (15). In fact, several studies have focused on the stability of nonhuman primate gut microbiota (39, 40), which may take into account long-term sampling and sampling difficulties. Similarly, the gut microbiota of these langurs remains relatively stable, possibly related to their leaf-eating habit. Next, we can expand the vertical sampling to better reflect the seasonal patterns of their gut microbiota.

The absence of significant seasonal differences in the top five dominant phyla (with a proportion of more than 98%) may be the digestive strategy adopted by the langurs in response to a high proportion of leaves in diets (30, 41). Furthermore, *Firmicutes* that efficiently ferment cellulose maintain relative stability across seasons include some taxa that specialize in decomposing the leaves from several major plant species, such as *Oscillospiraceae*, *Christensenellaceae*, and *Lachnospiraceae* (8, 18, 34). The dominant species were central to maintaining the function of the system (36, 37), and the presence of these dominant taxa in the François' langurs may be one of the important conditions to safeguard the normal functions of their gut microbiota. This may be analogous to the wild white-headed black langurs possessing a high abundance of *Firmicutes* digesting cellulose and carbohydrates (33), which is associated with the high leaf tropism of the rhesus macaques (33). Moreover, the abundance of the *Ruminococcaceae* family, which may effectively ferment cellulose, increases in the dry season; their response correlates with dry season onset, and their increase may provide a coping mechanism to the animals (8, 22); the other taxa with seasonal variations were low in abundance and belonged to phyla such as *Firmicutes* and *Bacteroidetes*. In addition, the low-abundance taxa are more susceptible to environmental changes due to their low proportions (15), which corresponds to the obvious seasonal variation of 21 low-abundance taxa at the family level in the langurs. Conversely, the high-abundance taxa are more resilient and are central to maintaining stability in the community structure, similar to those of the human gut microbiota (15, 36). Therefore, the elastic regulation in the gut microbiota of the langurs could function in response to the changes in seasonal food availability. However, overall, the gut microbiota of the langurs remains stable, suggesting that the langurs' gut microecosystem tolerates external disturbance.

There is no significant difference in activity budget and energy metabolism on across

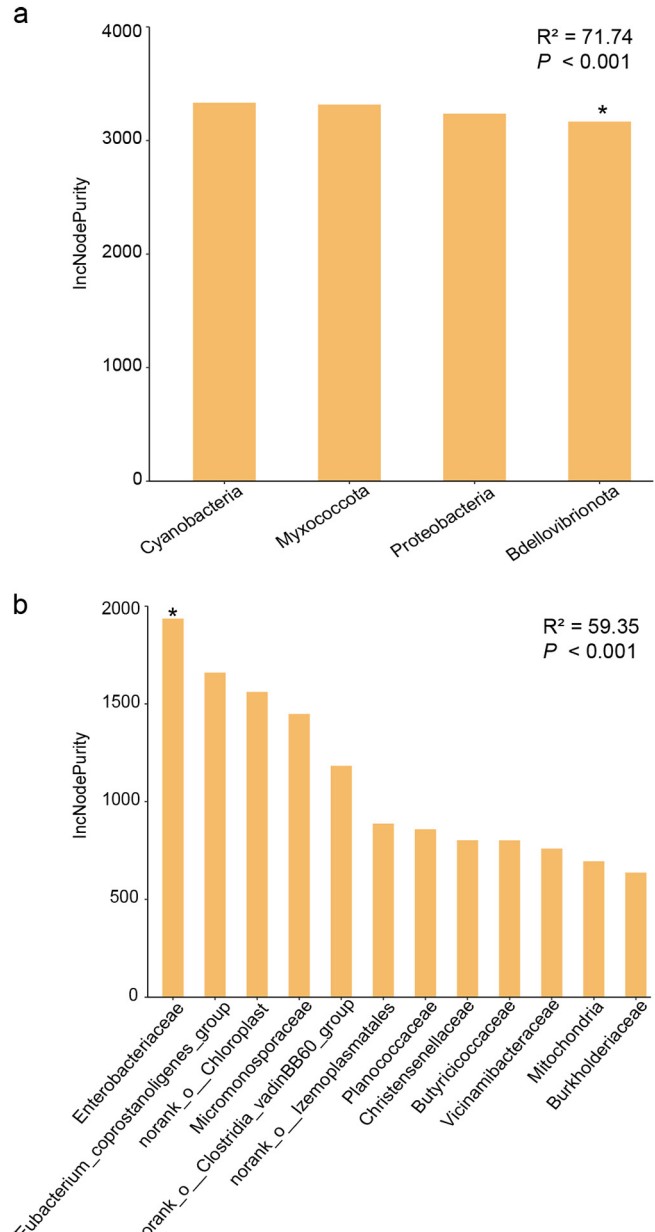

**FIG 7** The random forest model detects the bacterial taxa on T3 concentration fluctuations at the phylum level (a) and family level (b), with significant values determined for different bacterial taxa based on the sum of squared residuals representing the effect of each variable on the taxonomic nodes of the model. IncNodePurity, increase in node purity. *, $P < 0.05$.

seasons (Tables S11, S12, S14 and S15). We further hypothesize that the stability and elasticity of the gut microbiota are the key strategies for the langurs to adapt to the karst forests. Animals respond to environmental changes through behavioral adaptation as the first response (42); for example, white-headed black langurs reduce moving time and increase feeding time to meet their energy demand (43), and Assamese macaques (*Macaca assamensis*) mainly collect protein-rich foods to enhance their viability in low-quality habitats (24). However, behavioral adaptations are not always appropriate (42), and some animals respond with physiological strategies, such as the howler monkeys (*Alouatta palliata*), which reduce their thyroid hormone level to lower metabolic consumption to deal with food deprivation (44). We found no significant seasonal variations in activity budget, T3, and T4, suggesting that the langurs did not alleviate environmental stress by regulating behavioral patterns or reducing energy metabolism. Moreover, the gut microbiota of the langurs reaches

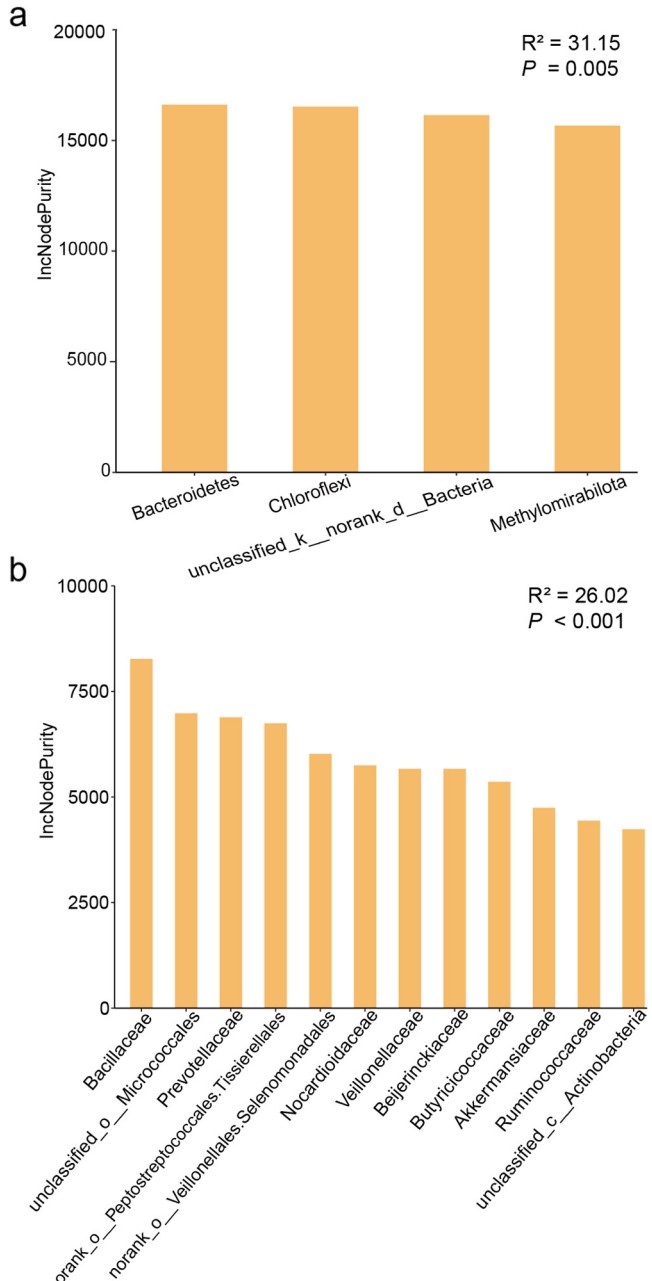

**FIG 8** The random forest model detects the bacterial taxa on T4 concentration fluctuations at the phylum level (a) and family level (b), with significant values determined for different bacterial taxa based on the sum of squared residuals representing the effect of each variable on the taxonomic nodes of the model. IncNodePurity, increase in node purity. *, $P < 0.05$.

an equilibrium in response to the food availability in the forest after transient fluctuations. The langurs in our study preferred a steady supply of several foods (41), as did the sympatric white-headed black langurs (45). The François' langurs shared a similar dietary composition across seasons (41), probably accounting for their relative stability in maintaining the dominant bacterial taxa. Similar host coevolution with gut microbiota is found in other primates (9, 10, 31), such as the gut microbiota of black howler monkeys, which provide an effective energy buffer for the hosts with resource fluctuations without requiring behavioral reactions (41). This suggests that energy resupply produced by the gut microbiota contributes to species adaptation to seasonal diet changes without changing behavioral patterns.

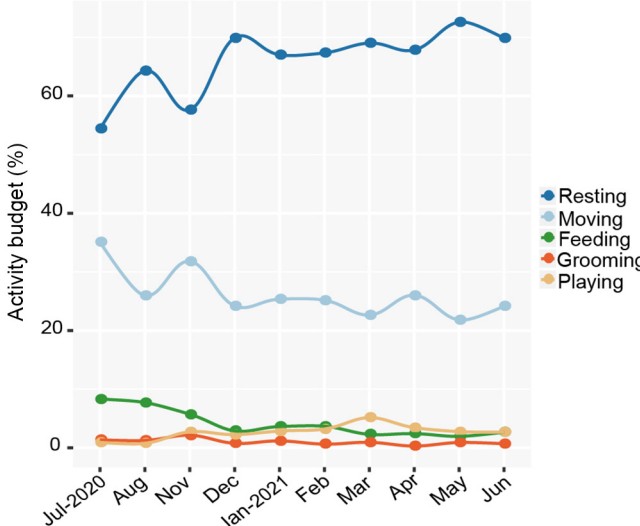

**FIG 9** Monthly activity budgets of François' langurs.

**Influence of climatic factors.** In the current study, we found significant correlations between most taxa and climatic variables, indicating that these bacteria are prevalent in response to climate change. However, VPA results showed that climatic variables could only explain <10% of the changes in community structure, likely because the langurs that experienced environmental effects in the limestone forest formed a coevolutionary adaptation strategy with the gut microbiota. Therefore, these ecological factors did not significantly affect the stability of the core taxa but prompted modest relative abundance changes in the gut microbiota. Specifically, rainfall and humidity are the main climate factors that explain the changes in the gut microbiota structure of the langurs, and this was previously found in other wild primates (20, 22, 26). But these two factors amount to only 8.93% of the maximum interpretation of the gut microbiota in the langurs. Rainfall is the climatic factor with the greatest impact on the gut microbiota of geladas, but by only 3.3% (22); it may affect the gut microbiota of animals by influencing vegetation growth (20, 22). Rainfall-induced changes in plant items and adjustments in foraging strategies may indirectly affect changes of the gut microbiota community of the langurs, which has also been seen in studies of other primates (20, 26), such as the western lowland gorillas (*Gorilla gorilla gorilla*) and chimpanzees (*Pan troglodytes*) (20).

Seasonal variations in plant items are particularly pronounced in limestone forests, where fruits and young leaves appear mainly in the rainy season (46). Primates living in the region, including François' langurs, often increase intake of fruits and young leaves during the rainy season (28, 43, 46). Moreover, increased biomass of vegetation during the rainy season (47) in turn alters food availability, changing the gut microbiota community. Indeed, in langurs, the abundance of *Christensenellaceae* and the *Eubacterium coprostanoligenes* group was slightly increased, and these organisms are adept at breaking down polysaccharides to deal with the foods prevalent in the rainy season. Therefore, the impact of climatic variables on the gut microbiota in langurs may be caused by affecting their food resources, suggesting the importance of protection and restoration of the fragmented habitats.

We admit that there are weaknesses in this study. The current study is based on a 16-month study period, likely leading to the tempering of our conclusions by the limited study period and relatively insufficient sequencing depth. However, we undoubtedly provide a general pattern of the seasonality in the gut microbiota of the limestone forest-dwelling François' langurs. Further study is required.

In conclusion, the gut microbiota of François' langurs in limestone forests is dominated by *Firmicutes* and *Bacteroidetes*, followed by *Oscillospiraceae*, *Christensenellaceae*, *Lachnospiraceae*, and *Ruminococcaceae* at the family level, which may assist in degrading the complex polysaccharides in plant items. Moreover, the five predominant phyla

remained relatively stable during the seasons, and only 21, with the low-abundance taxa at the family level, fluctuated appropriately, showing the stability and flexibility of the gut microbiota in these langurs. Specifically, the langurs exploit the structure and function of the gut microbiota to adapt to resource pressures from karst habitats, including changes in seasonal food availability caused by rainfall and humidity fluctuations and the continuous supply of mature leaves that are difficult to degrade. In addition, the strong metabolic capacity of the gut microbiota favors the langurs in acquiring energy from low-quality foods in the limestone forests, even without seasonal adjustments in activity budget and energy metabolism (thyroid hormones), enhancing the adaptation potential of the species.

## MATERIALS AND METHODS

**Study sites and animals.** The Guangxi Chongzuo White-Headed Langur National Nature Reserve (107°16′53″ to 107°59′46″E, 22°10′43″ to 22°36′55″N) is a typical karst landform with a height of 300 to 400 m (48). The reserve is in a tropical monsoon climate zone, where winter and spring are cold and dry and summer and autumn are hot and rainy (49). The monthly climate data for Chongzuo was collected from February 2019 to May 2020 and from July 2020 to June 2021. Rainfall during these two periods revealed that the flood season occurs from April to September (49), and the dry season occurs from October to March. The group of François' langurs ($n = 12$) that we studied comprised one adult male, six adult females, one subadult males, two juveniles, and two infants; the number of the group remained stable during the first period. The subadult left the group in October 2020, and then the group ($n = 11$) comprised an adult male, six adult females, and four juveniles.

**Sample collection.** Sterile tools were used to collect the inner parts of fresh feces from the sleeping site of the langurs in the early morning and packed into tubes for cryopreservation. The separation between feces was maintained to be greater than 2 m to avoid collecting feces from the same individual (whenever possible). We collected 152 fecal samples during the first period (Table S1) and 137 fecal samples during the second period (Table S2).

**Analysis of gut microbiota.** DNA was extracted from the samples using kits (E.Z.N.A. soil DNA kit) and checked by agarose gel electrophoresis; the purity of the DNA was determined using a NanoDrop 2000 device. Next, PCR amplification of 16S rRNA v3 to v4 was performed using primers 338F (5′-ACTCCTACGGGAGGCAGCAG-3′) and 806R (5′-GGACTACHVGGGTWTCTAAT-3′). The 20-$\mu$L PCR system was constructed using template DNA (10 ng), 5× FastPfu buffer (4 $\mu$L), 2.5 mM deoxynucleoside triphosphates (dNTPs; 2 $\mu$L), bovine serum albumin (0.2 $\mu$L), FastPfu polymerase (0.4 $\mu$L), primers (0.8 $\mu$L), and supplemental double-distilled water (ddH$_2$O). Subsequently, 3 min of predenaturation at 95℃, 30 s of denaturation at 95℃, 30 s of annealing at 53℃, and 45 s of extension at 72℃ for another 10 min. This cycle was repeated 30 times and held at 10℃ for 10 min. Three PCR amplifications were performed on each sample to obtain a PCR mixed product, and the products were quantified up to 0.5 ng/$\mu$L. A NEXTFLEX rapid DNA-Seq kit was used to construct the library, including adding Illumina adapters and removing adapters' self-connecting fragments by magnetic beads, PCR amplification to enrich library templates, and magnetic bead recovery of PCR products to obtain the final library. Finally, sequencing was completed with the Miseq PE300 platform. The analysis was performed using the Majorbio Cloud Platform. Table S3 shows the list of reagents and equipment used.

Optimized sequences were obtained after filtering low-quality raw sequences with Fastp (https://github.com/OpenGene/fastp), and the sequences were spliced using FLASH (http://www.cbcb.umd.edu/software/flash). The bases with quality less than 20 at the end of the reads were filtered, a sliding window of 50 bp was set to discard reads of <50 bp, and reads containing N were removed. The minimum overlap length of overlapping sequences was allowed to be 10 bp. The differentiation of samples relied on barcodes and primers at both ends of the sequence, with a barcode mismatch of 0 and primer mismatch of up to 2.

After removing nonrepeating unique sequences, operational taxonomic unit (OTU) clustering was performed on unique sequences based on 97% similarity (Uparse 7.0.1090; http://drive5.com/uparse/) to obtain the OTU table. All OTUs were aligned to SILVA (release138; https://www.arb-silva.de/) using the RDP classifier (version 2.11; http://sourceforge.net/projects/rdp-classifier/) to obtain the microbial information corresponding to each OTU with a confidence threshold set at 0.8. The sequence numbers of all samples are unified according to the minimum sequence number to reduce the effect of different sequencing depths. Since most OTUs cannot be accurately attributed to established genus or species levels, we completed the analysis at the family level.

The raw sequences were filtered after sufficient sequencing depth. We obtained a total of 9,085,392 original sequences, with an average of 59,772 ± 8,725 sequences for each sample, and 7,394,800 optimized sequences remained after filtering low-quality sequences, with an average of 48,650 ± 8,125 sequences for each sample. Good's coverage estimates of the 152 samples ranged from 98.82% to 99.79%, indicating that our sequencing results were sufficiently representative of the real situation of bacteria in the fecal samples. The rarefaction curve was used to demonstrate the rationality of the sequencing data (R 3.3.1) (Fig. S1).

Venn diagrams were used to demonstrate shared and unique taxa for two seasons, and histograms showed the relative abundance of the bacterial taxa. The Wilcoxon rank sum test was used to compare differences in composition between seasons at the phylum and family levels ($P < 0.05$), and the $P$ value was corrected using the false-discovery rate. The main bacterial taxa that significantly influenced seasonal differences were investigated using the random forest model (R 4.1.3; https://cran.r-project.org/). The training set of the model accounted for 80% of the data, and the test set accounted for 20%. Important OTUs were selected to establish

the prediction model by repeated 10-fold cross-validation five times to avoid a single test and the inadequacy of training data. The receiver operating characteristic curves were used to demonstrate the accuracy of the model.

The Shannon and invsimpson indices were calculated to reflect community diversity, and the Chao and ACE indices were used to reflect community richness. The generalized linear mixed model (GLMM) was used to compare the seasonal differences in the alpha diversity index (R 4.0.3) (24). Analysis of variance (ANOVA) was used to compare the difference between the two models with and without fixed effects to determine the influence of fixed effects on the response variable ($P < 0.05$). Initially, these indices were transformed to $\log_{10}(x)$ to improve the linear effect, with season as the fixed factor, the diversity index as the response variable, and the sample size as the random factor. Principal-coordinate analysis (PCoA) was used to calculate beta diversity to explore the similarities and differences in gut microbiota composition in two seasons and months under the unweighted UniFrac and weighted UniFrac distance.

Moreover, climate factors with strong multicollinearity by a variance inflation factor (VIF) were excluded until all VIFs were less than 10 for the next analyses. Redundancy analysis (RDA) was used to demonstrate the influence of climatic factors on the bacterial community, and a Monte Carlo permuted test was performed to determine whether these factors significantly influenced the gut microbiota ($P < 0.05$). Variant partitioning analysis (VPA) was used to further quantify the separate or multiple factors, such as rainfall, mean temperature, sunlight duration, and humidity (minimum and relative humidity), that contribute to differences in gut microbiota. Finally, the relationships between climatic factors and each bacterial taxa were obtained using heat maps of Spearman correlations. All of the analyses for our study (except those noted) were done in R 3.3.1.

**Assay of thyroid hormones.** The energy metabolism of wild animals can be obtained by monitoring thyroid hormones (44, 50), which can effectively evaluate the physiological state of animals in response to habitat changes (44). An enzyme-linked immunosorbent assay kit was used to quantify the concentrations of thyroid hormones (triiodothyronine [T3] and tetraiodothyronine [T4]) in the second batch of samples. Specifically, we utilized standard solutions that were diluted in sequence to set five concentrations to obtain a standard curve. Tube 5 was added with 150 $\mu$L standard solution and 150 $\mu$L diluent, tube 4 was added with 150 $\mu$L solution from tube 5 and 150 $\mu$L diluent, and so on to obtain 5 standard solutions and 1 blank tube. A blank well (without adding samples and enzyme-labeled antigen; the rest were the same), the standard wells, and the fecal sample wells were set on the microplate, and 50 $\mu$L of sample was added correspondingly, and the added fecal sample was diluted five times (40-$\mu$L dilution and 10-$\mu$L fecal sample). After incubation at 37℃ for 30 min, we repeated five washes to separate the substrate and hormone. Then, 50 $\mu$L enzyme-labeled reagent was added to the wells, and the wells were incubated and washed again. Reactions were terminated by hiding the mixtures from light for 15 min after addition of chromogens, and the absorbance in each well was examined at 450 nm using the spectrophotometer. Finally, we made a standard curve based on the concentration and absorbance of the standard solution and then calculated the T3 and T4 concentrations in the fecal samples, expressed as ng/g. A box chart was used to show the fluctuations of hormone content between months and seasons, and the GLMM was further established to compare the seasonal differences in T3 and T4. $\log_{10}(x)$ was used to improve the linear effect of the original data (24), with concentration as the response variable, sample size as the random factor, and season as the fixed factor; meanwhile, the ANOVA was used to test the seasonal differences in T3 and T4. Finally, a random forest model (51) was used to evaluate the effect of gut microbiota on T3 and T4.

**Analysis of activity budget.** We used instantaneous scan sampling to record various behaviors of the langurs (52). A scan lasted for 5 min, from left to right and recorded the animal's behavior, including resting, moving, feeding, playing, and grooming; the scan was done again at an interval of 10 min, thus sampling four times in 1 h. This sampling was stopped if the monkeys did not appear in our visual field range during the sampling time. Each scan was taken as an independent sample, and the ratio of the number of individuals in whom a certain behavior occurred to the total number of individuals observed indicated the activity budget for that behavior at the sampling. Finally, the activity budget per hour was taken as the base unit, and its average was taken to represent the monthly activity budget. If 4 samplings were not obtained within an hour, the data for the hour were not used. GLMM was used to analyze the seasonal change of the activity budget, with the sample size as the random factor, the activity budget as the response variable, and the season as the fixed factor.

**Ethics approval.** Our study collected fecal samples from the animals and did not require ethical approval because they did not involve animal tissue.

**Data availability.** The data sets presented in this study can be found in online repositories. The names of the repository/repositories and accession number(s) can be found under BioProject no. PRJNA900144.

## SUPPLEMENTAL MATERIAL

Supplemental material is available online only.
**SUPPLEMENTAL FILE 1**, XLSX file, 0.1 MB.
**SUPPLEMENTAL FILE 2**, PDF file, 1.9 MB.

## ACKNOWLEDGMENTS

This work was supported by the Guangxi Chongzuo White-Headed Langur National Nature Reserve.

We thank Feiyun He, Xianfeng Huang, and Da Li for their help with sample collection.

This work was supported by the National Natural Science Foundation of China (no. 32170488 and 31960106) and the Innovation Project of Guangxi Graduate Education, China (YCSW2023130).

We have no relevant financial or nonfinancial interests to disclose.

Zhonghao Huang designed the study. Hongying Liu analyzed the data and wrote the manuscript. Yuhui Li, Jipeng Liang, and Dengpan Nong collected samples. Zhonghao Huang and Youbang Li revised the manuscript. All authors read and approved the submitted manuscript.

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
