## [Reviewer comments · Microbiology Spectrum]

Microbiology Spectrum

Evaluation of gut microbiota stability and flexibility as a response to seasonal variation in the wild François' langurs (*Trachypithecus francoisi*) in limestone forest

Hongying Liu, Yuhui Li, JiPeng Liang, Dengpan Nong, Youbang Li, and Zhonghao Huang

Corresponding Author(s): Zhonghao Huang, Guangxi Normal University

Review Timeline:

Submission Date:	December 12, 2022
Editorial Decision:	February 17, 2023
Revision Received:	March 13, 2023
Editorial Decision:	April 21, 2023
Revision Received:	May 10, 2023
Editorial Decision:	May 24, 2023
Revision Received:	June 7, 2023
Accepted:	June 10, 2023

Editor: John Chaston

Reviewer(s): The reviewers have opted to remain anonymous.

Transaction Report:

DOI: <https://doi.org/10.1128/spectrum.05091-22>

February 17, 2023

Dr. Zhonghao Huang
Guangxi Normal University
Guilin
China

Re: Spectrum05091-22 (Evaluation of gut microbiota stability and flexibility as a response to seasonal variation in the wild François' langurs (*Trachypithecus francoisi*) in limestone forest)

Dear Dr. Zhonghao Huang:

Thank you for submitting your manuscript to Microbiology Spectrum. It has now been reviewed by three experts in the field, and they have raised issues that must be addressed before I can consider the manuscript appropriate for acceptance to Microbiology Spectrum. I encourage you to consider submitting a revised version of the manuscript, and, if you do so, to pay attention to themes raised by multiple reviewers. Additionally, one reviewer raised concerns about the differences in this study relative to one of your previous studies, and I encourage you to be sure to document these differences clearly in your response. Evaluating if this is a replication finding or not is an important part of the review process (some replication findings are useful whereas others are not).

Link Not Available

Sincerely,

John Chaston

Journals Department
Reviewer comments:

Reviewer #1 (Comments for the Author):

The authors studied the compositional structure and seasonal variation in the gut microbiota of François's langur by 16S rRNA

sequencing and fecal hormone assay. The design of this study is good, and the results can provide data basis for wildlife protection. The writing of the article is relatively rough, and English writing needs professional modification and improvement.

The main questions of this manuscript are:

1. The logic of content writing and key content are not highlighted: for example, the title of the manuscript indicates that it is an assessment of the gut microbiota of langurs under seasonal variation, while the abstract does not summarize the flora related to seasonal changes; The results did not show the information of genus level flora. The accuracy and recognition of 16S rRNA amplicon sequencing technology in the analysis of flora at the genus level are relatively high now. It is suggested that the author can summarize and display the flora at the genus level under the key flora at family level, so that the analysis will be more in-depth and enhance the readability.
2. The description of the contents and statements in many parts of this article is vague: For example, some contents of the method steps are not clearly written, the sentence "We studied two cohorts of wild Francois' langurs with stable numbers over the periods" led me to believe that this study studied two cohorts of wild Francois' langurs with stable numbers over the periods. In addition, why does the determination of fecal hormones in the results not use the samples in Table s1, but use the fecal samples in Table s2 collected at different times? This makes me very confused when reading this manuscript!
3. Many of the words used in the full text are inaccurate: for example, the title in Table s16 uses "several primates", but this table clearly listed 11 animals, so it can be written directly as "11 primates". In addition, the remarks under this table are not properly written as "1-Firmicutes"; the "invsimpson" in Table s17 does not need to be italicized; The notional words in the title of Figure 2 need not be capitalized, and the four figures in the figure 2 need to be written with a, b, c and d respectively; What do a and b in Figure 3 represent? Figures 2 and 5 also need to refer to a, b, c and d.
4. Some writing of the full text are inconsistent: for example, the letter P should be capitalized; "triiodothyronine (T3) and tetraiodothyronine (T4)" are unified if the initial letters are all capitalized.

Reviewer #2 (Comments for the Author):

The manuscript described microbiome characterization by seasonal variation in the wild Francois's langurs living in limestone forest. This paper revealed that microbiome represented relatively stable during the seasons. Only the lower bacteria taxa at the family level showed variable depending on seasons.

Major revision

1. Significant different microbiota composition at phylum and family level was identified in wild Francois langurs reared in the same place between this paper and a previous paper published in this research team (<https://doi.org/10.21203/rs.3.rs-2377898/v1>) (Bacteroidetes 10.29% {plus minus} 8.50% vs 4.82% {plus minus} 1.41%; Actinobacteria 4.05% {plus minus} 5.35% vs 9.11% {plus minus} 8.20%; Ocsillospiraceae 17.74%{plus minus}6.86, 30.21% {plus minus} 4.87% etc.). The reason for this difference should be explained.
2. Direct proof should be added regarding increased level of Christensenellaceae is associated with dealing with foods such as young leaves and fruits.
3. The energy metabolism have been known for association with various factors including other hormones.

Reviewer #3 (Comments for the Author):

The manuscript reflects a broad and robust dataset that reports on the fecal (not the gut) microbiota of wild langurs, and also reports on the collection of some other animal data and the correlation between these values and microbiota abundance. The authors use numerous statistical tests and they are generally appropriately applied; however there are superior tests that the authors could and should use in their analyses. Finally, I am very concerned that the paper is not very easy to read, and the authors could do some work to improve the structure of the results and discussion.

Tests: The authors use ANOVA and adonis to discriminate differences in overall microbiota composition between treatments, and also use Wilcoxon tests to discriminate changes in abundance of individual microbes. I recommend the authors replace these analyses with PERMANOVA (adonis is PERMANOVA, but when run through some applications such as the qiime2 software, it may not be possible to use multiple covariates. If the authors move to R, they should be able to perform a PERMANOVA using the vegan package (function is named adonis). Also, a method such as ANCOM or LefSe could be used to identify individual microbes that vary in abundance between treatments.

Also, I am concerned about how the authors interpret their tests. For example, at line 245-56 the authors report that there is no difference in T3 or T4 levels with season; yet then they go on to report microbial abundances that covaried with T3 or T4 levels. This does not make sense, since there is no variation to calculate correlations for. Note that L251 says the analysis revealed

bacteria that had significant effects on T3 and T4, when in reality they only identified taxa whose abundance is correlated with T3 and T4.

Language and interpretations: There are some issues with interpretations. For example, at L279-90, 283-4, 287-8, 292-6, 298, 299-302, 302-3, 303-4, 305-6, 311 and 312-4 (I don't list beyond here, but these represent many issues in a short span), the interpretations are overstated. In some instances I think this is because of language - for example, at L283-4 the authors are likely citing a previous finding, but it is phrased as a conclusion. Others, maybe also be languages, such as L 311 "gut microbiota varied significantly on smaller timescales [similar to humans]" the structure of the sentence implies that human microbiomes vary across small time scales but not large ones and cites papers that only support stability. I think the authors only intend those references to apply to the statement about long stability, but the sentence needs to be restructured to reflect this. Finally, for this issue about long-term stability and short-term variation I think this needs to be better explained in the text, as any short term variability must by definition mean there is long-term variability (but the cited studies are correct, the difference between the results just needs to be reconciled better in the text). Generally, I think the authors need to pay very close attention to the intro and discussion to make sure findings are not over interpreted.

Use of supporting information: I think some the information in the supporting information could be moved to the main text to keep the reading clearer. Certainly there need to be more details in the methods (L121 - what are the kits? what are the per parameters? L123 - what software packages were used on the MajorBio Cloud platform, etc). Also, the authors' selection of figures adds some confusion to the manuscript. For example, there are no main text figures supporting measuring T3 or T4 levels, animal activity (these are in supplemental), and there is no taxon plot showing the overall microbiomes of the samples. These changes may not be required, but I recommend the authors pay attention to them to ensure the manuscript is clear and direct.

Staff Comments:

Preparing Revision Guidelines

Please return the manuscript within 60 days; if you cannot complete the modification within this time period, please contact me. If you do not wish to modify the manuscript and prefer to submit it to another journal, please notify me of your decision immediately so that the manuscript may be formally withdrawn from consideration by Microbiology Spectrum.

Responses to Reviewers' Comments

NOTE: The reviewers' comments are in black, and our responses are in red. The line numbers in our responses are those in the clean version of current manuscript.

Associate Editor Comments to Author:

Thank you for submitting your manuscript to Microbiology Spectrum. It has now been reviewed by three experts in the field, and they have raised issues that must be addressed before I can consider the manuscript appropriate for acceptance to Microbiology Spectrum. I encourage you to consider submitting a revised version of the manuscript, and, if you do so, to pay attention to themes raised by multiple reviewers. Additionally, one reviewer raised concerns about the differences in this study relative to one of your previous studies, and I encourage you to be sure to document these differences clearly in your response. Evaluating if this is a replication finding or not is an important part of the review process (some replication findings are useful whereas others are not).

Thanks for your comments. In current version, we revised manuscript according to the comments from editors and reviewers.

Our study is different from the previous study. Current study aims to examine the seasonal shift pattern of the gut microbiota of François's langurs, whereas the previous one attempts to detect the effect of captivity on the gut microbiota of the wild François's langurs. Moreover, current works cover a completed year circle and provide data based on 152 fecal samples from the wild langurs. The previous paper is based on 32 fecal samples (15 from the provisioned members and 17 from the wild individuals), which provides limited descriptive information without any data on seasonality. In addition, we combined behavioral and fecal hormone data into current study. Actually, the fecal samples used in the two studies were different because of the different sampling times and number of samples.

Reviewer(s)' Comments to Author:

Reviewer 1:

The authors studied the compositional structure and seasonal variation in the gut microbiota of François's langur by 16S rRNA sequencing and fecal hormone assay. The design of this study is good, and the results can provide data basis for wildlife protection. The writing of the article is relatively rough, and English writing needs professional modification and improvement.

The main questions of this manuscript are:

1. The logic of content writing and key content are not highlighted: for example, the title of the manuscript indicates that it is an assessment of the gut microbiota of langurs under seasonal

variation, while the abstract does not summarize the flora related to seasonal changes; The results did not show the information of genus level flora. The accuracy and recognition of 16S rRNA amplicon sequencing technology in the analysis of flora at the genus level are relatively high now. It is suggested that the author can summarize and display the flora at the genus level under the key flora at family level, so that the analysis will be more in-depth and enhance the readability.

We deeply appreciate for your comments, and we have summarized the seasonal changes of the gut microbiota in the langurs in the abstract, please see L28-31.

Indeed, we tried to analyze the compositional structure of the gut microbiota of the langurs at the genus level, but more than 250 bacterial taxa out of 664 were undefined, as were some taxa in the top 5 dominant families (Table 1), likely due to the abundant novel species in the gut of study langurs. And about 50% of the top 15 bacterial taxa in the genus level with high relative abundances were undefined (Figure 1). This is why the subsequent classification of bacterial taxa only reached the family level.

Figure 1 Composition of gut microbiota at the genus level of the François's langurs, showing only the top 15 dominant bacterial taxa.

Table 1 Relative abundance of all bacterial taxa in gut microbiota at the genus level.

OTU ID	relative abundance
p__Firmicutes; c__Clostridia; o__Christensenellales; f__Christensenellaceae; g__Christensenellaceae_R-7_group	0.151
p__Firmicutes; c__Clostridia; o__Christensenellales; f__Christensenellaceae; g__norank_f__Christensenellaceae	0.001

p__Firmicutes; c__Clostridia; o__Christensenellales; f__Christensenellaceae; g__unclassified_f__Christensenellaceae	< 0.001
p__Firmicutes; c__Clostridia; o__Oscillospirales; f__Oscillospiraceae; g__NK4A214_group	0.102
p__Firmicutes; c__Clostridia; o__Oscillospirales; f__Oscillospiraceae; g__UCG-005	0.043
p__Firmicutes; c__Clostridia; o__Oscillospirales; f__Oscillospiraceae; g__unclassified_f__Oscillospiraceae	0.014
p__Firmicutes; c__Clostridia; o__Oscillospirales; f__Oscillospiraceae; g__UCG-002	0.009
p__Firmicutes; c__Clostridia; o__Oscillospirales; f__Oscillospiraceae; g__norank_f__Oscillospiraceae	0.006
p__Firmicutes; c__Clostridia; o__Oscillospirales; f__Oscillospiraceae; g__UCG-007	0.001
p__Firmicutes; c__Clostridia; o__Oscillospirales; f__Oscillospiraceae; g__Oscillibacter	0.001
p__Firmicutes; c__Clostridia; o__Oscillospirales; f__Oscillospiraceae; g__Colidextribacter	0.001
p__Firmicutes; c__Clostridia; o__Oscillospirales; f__Oscillospiraceae; g__Intestinimonas	< 0.001
p__Firmicutes; c__Clostridia; o__Oscillospirales; f__Oscillospiraceae; g__UCG-003	< 0.001
p__Firmicutes; c__Clostridia; o__Oscillospirales; f__Oscillospiraceae; g__Papillibacter	< 0.001
p__Verrucomicrobiota; c__Verrucomicrobiae; o__Verrucomicrobiales; f__Akkermansiaceae; g__Akkermansia	0.068
p__Firmicutes; c__Clostridia; o__Oscillospirales; f__Ruminococcaceae; g__Ruminococcus	0.048
p__Firmicutes; c__Clostridia; o__Oscillospirales; f__Ruminococcaceae; g__norank_f__Ruminococcaceae	0.008
p__Firmicutes; c__Clostridia; o__Oscillospirales; f__Ruminococcaceae; g__CAG-352	0.003
p__Firmicutes; c__Clostridia; o__Oscillospirales; f__Ruminococcaceae; g__Faecalibacterium	0.002
p__Firmicutes; c__Clostridia; o__Oscillospirales; f__Ruminococcaceae; g__Eubacterium_siraenum_group	0.002
p__Firmicutes; c__Clostridia; o__Oscillospirales; f__Ruminococcaceae; g__unclassified_f__Ruminococcaceae	0.002
p__Firmicutes; c__Clostridia; o__Oscillospirales; f__Ruminococcaceae; g__Candidatus_Soleaferrea	< 0.001
p__Firmicutes; c__Clostridia; o__Oscillospirales; f__Ruminococcaceae; g__Paludicola	< 0.001
p__Firmicutes; c__Clostridia; o__Oscillospirales; f__Ruminococcaceae; g__UBA1819	< 0.001
p__Firmicutes; c__Clostridia; o__Oscillospirales; f__Ruminococcaceae; g__Pygmaibacter	< 0.001
p__Firmicutes; c__Clostridia; o__Oscillospirales; f__Ruminococcaceae; g__Subdoligranulum	< 0.001
p__Firmicutes; c__Clostridia; o__Lachnospirales; f__Lachnospiraceae; g__unclassified_f__Lachnospiraceae	0.04
p__Firmicutes; c__Clostridia; o__Lachnospirales; f__Lachnospiraceae; g__Marvinbryantia	0.021
p__Firmicutes; c__Clostridia; o__Lachnospirales; f__Lachnospiraceae; g__Eubacterium_ruminantium_group	0.009
p__Firmicutes; c__Clostridia; o__Lachnospirales; f__Lachnospiraceae; g__Blautia	0.005
p__Firmicutes; c__Clostridia; o__Lachnospirales; f__Lachnospiraceae; g__Anaerostipes	0.003
p__Firmicutes; c__Clostridia; o__Lachnospirales; f__Lachnospiraceae; g__Lachnospiraceae_AC2044_group	0.003
p__Firmicutes; c__Clostridia; o__Lachnospirales; f__Lachnospiraceae; g__Dorea	0.003
p__Firmicutes; c__Clostridia; o__Lachnospirales; f__Lachnospiraceae; g__Roseburia	0.002
p__Firmicutes; c__Clostridia; o__Lachnospirales; f__Lachnospiraceae; g__Acetitomaculum	0.001
p__Firmicutes; c__Clostridia; o__Lachnospirales; f__Lachnospiraceae; g__Tyzzerella	0.001
p__Firmicutes; c__Clostridia; o__Lachnospirales; f__Lachnospiraceae; g__Eubacterium_xylophilum_group	0.001
p__Firmicutes; c__Clostridia; o__Lachnospirales; f__Lachnospiraceae; g__Anaerospobacter	0.001
p__Firmicutes; c__Clostridia; o__Lachnospirales; f__Lachnospiraceae; g__Lachnospira	< 0.001
p__Firmicutes; c__Clostridia; o__Lachnospirales; f__Lachnospiraceae; g__Frisingicoccus	< 0.001
p__Firmicutes; c__Clostridia; o__Lachnospirales; f__Lachnospiraceae; g__Tuzzerella	< 0.001
p__Firmicutes; c__Clostridia; o__Lachnospirales; f__Lachnospiraceae; g__Lachnospiraceae_NK4A136_group	< 0.001
p__Firmicutes; c__Clostridia; o__Lachnospirales; f__Lachnospiraceae; g__Ruminococcus_torques_group	< 0.001
p__Firmicutes; c__Clostridia; o__Lachnospirales; f__Lachnospiraceae; g__norank_f__Lachnospiraceae	< 0.001
p__Firmicutes; c__Clostridia; o__Lachnospirales; f__Lachnospiraceae; g__Lachnospiraceae_UCG-010	< 0.001

p__Firmicutes; c__Clostridia; o__Lachnospirales; f__Lachnospiraceae; g__Shuttleworthia	< 0.001
p__Firmicutes; c__Clostridia; o__Lachnospirales; f__Lachnospiraceae; g__Lachnospiraceae_NK3A20_group	< 0.001
p__Firmicutes; c__Clostridia; o__Lachnospirales; f__Lachnospiraceae; g__GCA-900066575	< 0.001
p__Firmicutes; c__Clostridia; o__Lachnospirales; f__Lachnospiraceae; g__Eubacterium_oxidoreducens_group	< 0.001
p__Firmicutes; c__Clostridia; o__Lachnospirales; f__Lachnospiraceae; g__Lachnospiraceae_FCS020_group	< 0.001
p__Firmicutes; c__Clostridia; o__Lachnospirales; f__Lachnospiraceae; g__Eubacterium_ventriosum_group	< 0.001
p__Firmicutes; c__Clostridia; o__Lachnospirales; f__Lachnospiraceae; g__Ruminococcus_gauvreauii_group	< 0.001
p__Firmicutes; c__Clostridia; o__Lachnospirales; f__Lachnospiraceae; g__Bacteroides_pectinophilus_group	< 0.001
p__Firmicutes; c__Clostridia; o__Lachnospirales; f__Lachnospiraceae; g__Lachnospiraceae_UCG-006	< 0.001
p__Firmicutes; c__Clostridia; o__Lachnospirales; f__Lachnospiraceae; g__Lachnoclostridium	< 0.001
p__Firmicutes; c__Clostridia; o__Lachnospirales; f__Lachnospiraceae; g__Eubacterium_hallii_group	< 0.001
p__Firmicutes; c__Clostridia; o__Lachnospirales; f__Lachnospiraceae; g__Lachnospiraceae_UCG-001	< 0.001
p__Firmicutes; c__Clostridia; o__Lachnospirales; f__Lachnospiraceae; g__Coproccoccus	< 0.001
p__Firmicutes; c__Clostridia; o__Lachnospirales; f__Lachnospiraceae; g__A2	< 0.001
p__Firmicutes; c__Clostridia; o__Lachnospirales; f__Lachnospiraceae; g__Butyrivibrio	< 0.001
p__Bacteroidota; c__Bacteroidia; o__Bacteroidales; f__Muribaculaceae; g__norank_f__Muribaculaceae	0.063
p__Firmicutes; c__Clostridia; o__Oscillospirales; f__UCG-010; g__norank_f__UCG-010	0.06
p__Firmicutes; c__Clostridia; o__Clostridia_UCG-014; f__norank_o__Clostridia_UCG-014; g__norank_f__norank_o__Clostridia_UCG-014	0.057
p__Firmicutes; c__Clostridia; o__Oscillospirales; f__Eubacterium_coprostanoligenes_group; g__norank_f__Eubacterium_coprostanoligenes_group	0.049
p__Firmicutes; c__Clostridia; o__Oscillospirales; f__Butyricoccaceae; g__UCG-009	0.019
p__Firmicutes; c__Clostridia; o__Monoglobales; f__Monoglobaceae; g__Monoglobus	0.016
p__Bacteroidota; c__Bacteroidia; o__Bacteroidales; f__Prevotellaceae; g__unclassified_f__Prevotellaceae	0.015
p__Actinobacteriota; c__Coriobacteriia; o__Coriobacteriales; f__Eggerthellaceae; g__unclassified_f__Eggerthellaceae	0.015
p__Firmicutes; c__Clostridia; o__Peptostreptococcales-Tissierellales; f__Anaerovoracaceae; g__Family_XIII_AD3011_group	0.015
p__Firmicutes; c__Clostridia; o__Peptostreptococcales-Tissierellales; f__Peptostreptococcaceae; g__Intestinibacter	0.014
p__Firmicutes; c__Clostridia; o__unclassified_c__Clostridia; f__unclassified_c__Clostridia; g__unclassified_c__Clostridia	0.012
p__Bacteroidota; c__Bacteroidia; o__Bacteroidales; f__Prevotellaceae; g__Prevotella	0.012
p__Actinobacteriota; c__Coriobacteriia; o__Coriobacteriales; f__Eggerthellaceae; g__Enterorhabdus	0.01
p__Proteobacteria; c__Gammaproteobacteria; o__Pseudomonadales; f__Moraxellaceae; g__Acinetobacter	0.007
p__Firmicutes; c__Clostridia; o__Clostridia_vadinBB60_group; f__norank_o__Clostridia_vadinBB60_group; g__norank_f__norank_o__Clostridia_vadinBB60_group	0.005
p__Cyanobacteria; c__Cyanobacteriia; o__Chloroplast; f__norank_o__Chloroplast; g__norank_f__norank_o__Chloroplast	0.005
p__Firmicutes; c__Bacilli; o__RF39; f__norank_o__RF39; g__norank_f__norank_o__RF39	0.005
p__Firmicutes; c__Clostridia; o__Oscillospirales; f__unclassified_o__Oscillospirales; g__unclassified_o__Oscillospirales	0.004
p__Spirochaetota; c__Spirochaetia; o__Spirochaetales; f__Spirochaetaceae; g__Treponema	0.004
p__Actinobacteriota; c__Actinobacteria; o__Micrococcales; f__Micrococcaceae; g__Citricoccus	0.004
p__Bacteroidota; c__Bacteroidia; o__Bacteroidales; f__Rikenellaceae; g__unclassified_f__Rikenellaceae	0.004
p__Bacteroidota; c__Bacteroidia; o__Bacteroidales; f__Bacteroidaceae; g__Bacteroides	0.004
p__Proteobacteria; c__Gammaproteobacteria; o__Enterobacteriales; f__Enterobacteriaceae; g__Escherichia-Shigella	0.004
p__Actinobacteriota; c__Actinobacteria; o__Corynebacteriales; f__Corynebacteriaceae; g__Corynebacterium	0.003
p__Firmicutes; c__Clostridia; o__Peptococcales; f__Peptococcaceae; g__norank_f__Peptococcaceae	0.002

p__Cyanobacteria; c__Vampirivibrionia; o__Gastranaerophilales; f__norank_o__Gastranaerophilales; g__norank_f__norank_o__Gastranaerophilales	0.002
p__Firmicutes; c__Bacilli; o__Izemoplasmatales; f__norank_o__Izemoplasmatales; g__norank_f__norank_o__Izemoplasmatales	0.002
p__Firmicutes; c__Clostridia; o__Eubacteriales; f__Anaerofustaceae; g__Anaerofustis	0.002
p__Bacteroidota; c__Bacteroidia; o__Bacteroidales; f__Rikenellaceae; g__Rikenellaceae_RC9_gut_group	0.002
p__Firmicutes; c__Clostridia; o__Peptostreptococcales-Tissierellales; f__Anaerovoracaceae; g__Family_XIII_UCG-001	0.002
p__Actinobacteriota; c__Coriobacteriia; o__Coriobacteriales; f__Coriobacteriales_Incertae_Sedis; g__norank_f__Coriobacteriales_Incertae_Sedis	0.001
p__Actinobacteriota; c__Coriobacteriia; o__Coriobacteriales; f__Atopobiaceae; g__norank_f__Atopobiaceae	0.001
p__Verrucomicrobiota; c__Kiritimatiellae; o__WCHB1-41; f__norank_o__WCHB1-41; g__norank_f__norank_o__WCHB1-41	0.001
p__Firmicutes; c__Clostridia; o__norank_c__Clostridia; f__norank_o__norank_c__Clostridia; g__norank_f__norank_o__norank_c__Clostridia	0.001
p__Firmicutes; c__Clostridia; o__Peptostreptococcales-Tissierellales; f__Anaerovoracaceae; g__Eubacterium_brachy_group	0.001
p__Actinobacteriota; c__Actinobacteria; o__Micrococcales; f__Dermatophilaceae; g__Dermatophilus	0.001
p__Bacteroidota; c__Bacteroidia; o__Bacteroidales; f__Prevotellaceae; g__Prevotellaceae_UCG-001	0.001
p__Campilobacterota; c__Campylobacteria; o__Campylobacterales; f__Campylobacteraceae; g__Campylobacter	0.001
p__Patescibacteria; c__Saccharimonadia; o__Saccharimonadales; f__Saccharimonadaceae; g__Candidatus_Saccharimonas	0.001
p__Firmicutes; c__Clostridia; o__Oscillospirales; f__norank_o__Oscillospirales; g__norank_f__norank_o__Oscillospirales	0.001
p__Actinobacteriota; c__Coriobacteriia; o__Coriobacteriales; f__norank_o__Coriobacteriales; g__norank_f__norank_o__Coriobacteriales	0.001
p__Firmicutes; c__Clostridia; o__Eubacteriales; f__Eubacteriaceae; g__Pseudoramibacter	< 0.001
p__Bacteroidota; c__Bacteroidia; o__unclassified_c__Bacteroidia; f__unclassified_c__Bacteroidia; g__unclassified_c__Bacteroidia	< 0.001
p__Bacteroidota; c__Bacteroidia; o__Bacteroidales; f__p-2534-18B5_gut_group; g__norank_f__p-2534-18B5_gut_group	< 0.001
p__Firmicutes; c__Bacilli; o__Bacillales; f__Planococcaceae; g__Solibacillus	< 0.001
p__Proteobacteria; c__Gammaproteobacteria; o__Burkholderiales; f__Oxalobacteraceae; g__Oxalobacter	< 0.001
p__Firmicutes; c__Bacilli; o__Erysipelotrichales; f__Erysipelotrichaceae; g__Erysipelotrichaceae_UCG-003	< 0.001
p__Actinobacteriota; c__Coriobacteriia; o__Coriobacteriales; f__Eggerthellaceae; g__norank_f__Eggerthellaceae	< 0.001
p__Proteobacteria; c__Gammaproteobacteria; o__Enterobacteriales; f__Enterobacteriaceae; g__Klebsiella	< 0.001
p__Actinobacteriota; c__Actinobacteria; o__Pseudonocardiales; f__Pseudonocardiaceae; g__Pseudonocardia	< 0.001
p__Chloroflexi; c__Chloroflexia; o__Thermomicrobiales; f__JG30-KF-CM45; g__norank_f__JG30-KF-CM45	< 0.001
p__Firmicutes; c__Clostridia; o__Peptostreptococcales-Tissierellales; f__Anaerovoracaceae; g__unclassified_f__Anaerovoracaceae	< 0.001
p__Spirochaetota; c__Brachyspirae; o__Brachyspirales; f__Brachyspiraceae; g__Brachyspira	< 0.001
p__Firmicutes; c__Bacilli; o__Lactobacillales; f__Lactobacillaceae; g__Lactobacillus	< 0.001
p__Actinobacteriota; c__Actinobacteria; o__Micrococcales; f__Micrococcaceae; g__Kocuria	< 0.001
p__Bacteroidota; c__Bacteroidia; o__Bacteroidales; f__unclassified_o__Bacteroidales; g__unclassified_o__Bacteroidales	< 0.001
p__Bacteroidota; c__Bacteroidia; o__Bacteroidales; f__norank_o__Bacteroidales; g__norank_f__norank_o__Bacteroidales	< 0.001
p__Proteobacteria; c__Alphaproteobacteria; o__Rhodospirillales; f__norank_o__Rhodospirillales; g__norank_f__norank_o__Rhodospirillales	< 0.001
p__Firmicutes; c__Bacilli; o__Bacillales; f__Planococcaceae; g__Kurthia	< 0.001
p__Proteobacteria; c__unclassified_p__Proteobacteria; o__unclassified_p__Proteobacteria; f__unclassified_p__Proteobacteria; g__unclassified_p__Proteobacteria	< 0.001
p__Actinobacteriota; c__Coriobacteriia; o__Coriobacteriales; f__Atopobiaceae; g__Olsenella	< 0.001
p__Actinobacteriota; c__Coriobacteriia; o__Coriobacteriales; f__unclassified_o__Coriobacteriales;	< 0.001

g_unclassified_o_Coriobacteriales	
p_Firmicutes; c_Clostridia; o_Oscillospirales; f_Butyricocccaceae; g_unclassified_f_Butyricocccaceae	< 0.001
p_unclassified_k_norank_d_Bacteria; c_unclassified_k_norank_d_Bacteria; o_unclassified_k_norank_d_Bacteria; f_unclassified_k_norank_d_Bacteria; g_unclassified_k_norank_d_Bacteria	< 0.001
p_Actinobacteriota; c_Actinobacteria; o_Propionibacteriales; f_Propionibacteriaceae; g_Luteococcus	< 0.001
p_Firmicutes; c_Clostridia; o_Clostridiales; f_Clostridiaceae; g_Clostridium_sensu_stricto_1	< 0.001
p_Actinobacteriota; c_Actinobacteria; o_Micrococcales; f_Micrococccaceae; g_unclassified_f_Micrococccaceae	< 0.001
p_Firmicutes; c_Clostridia; o_Peptostreptococcales-Tissierellales; f_Peptostreptococccaceae; g_unclassified_f_Peptostreptococccaceae	< 0.001
p_Actinobacteriota; c_Actinobacteria; o_Micrococcales; f_unclassified_o_Micrococcales; g_unclassified_o_Micrococcales	< 0.001
p_Proteobacteria; c_Gammaproteobacteria; o_Pseudomonadales; f_Pseudomonadaceae; g_Pseudomonas	< 0.001
p_Proteobacteria; c_Alphaproteobacteria; o_Paracaedibacterales; f_Paracaedibacteraceae; g_norank_f_Paracaedibacteraceae	< 0.001
p_Firmicutes; c_Bacilli; o_Erysipelotrichales; f_Erysipelotrichaceae; g_unclassified_f_Erysipelotrichaceae	< 0.001
p_Firmicutes; c_Negativicutes; o_Veillonellales-Selenomonadales; f_norank_o_Veillonellales-Selenomonadales; g_norank_f_norank_o_Veillonellales-Selenomonadales	< 0.001
p_Actinobacteriota; c_Actinobacteria; o_Corynebacteriales; f_Nocardiaceae; g_Rhodococcus	< 0.001
p_Actinobacteriota; c_Actinobacteria; o_unclassified_c_Actinobacteria; f_unclassified_c_Actinobacteria; g_unclassified_c_Actinobacteria	< 0.001
p_Desulfobacterota; c_Desulfovibrionia; o_Desulfovibrionales; f_Desulfovibrionaceae; g_Desulfovibrio	< 0.001
p_Proteobacteria; c_Gammaproteobacteria; o_Burkholderiales; f_unclassified_o_Burkholderiales; g_unclassified_o_Burkholderiales	< 0.001
p_Proteobacteria; c_Gammaproteobacteria; o_Burkholderiales; f_Burkholderiaceae; g_Burkholderia-Caballeronia-Paraburkholderia	< 0.001
p_Actinobacteriota; c_Actinobacteria; o_Micrococcales; f_Bogoriellaceae; g_Georgenia	< 0.001
p_Proteobacteria; c_Alphaproteobacteria; o_Rhodobacterales; f_Rhodobacteraceae; g_Paracoccus	< 0.001
p_Proteobacteria; c_Alphaproteobacteria; o_Rhodobacterales; f_Rhodobacteraceae; g_unclassified_f_Rhodobacteraceae	< 0.001
p_Bacteroidota; c_Bacteroidia; o_Bacteroidales; f_Rikenellaceae; g_Alistipes	< 0.001
p_Firmicutes; c_Bacilli; o_Bacillales; f_Bacillaceae; g_Bacillus	< 0.001
p_Proteobacteria; c_Alphaproteobacteria; o_Rhizobiales; f_Rhizobiaceae; g_unclassified_f_Rhizobiaceae	< 0.001
p_Proteobacteria; c_Alphaproteobacteria; o_Rhizobiales; f_Devesiaceae; g_Devesia	< 0.001
p_Bacteroidota; c_Bacteroidia; o_Flavobacteriales; f_Flavobacteriaceae; g_Flavobacterium	< 0.001
p_Actinobacteriota; c_Actinobacteria; o_Propionibacteriales; f_Nocardioidaceae; g_Nocardioides	< 0.001
p_Actinobacteriota; c_Actinobacteria; o_Microtrichales; f_Ilumatobacteraceae; g_norank_f_Ilumatobacteraceae	< 0.001
p_Actinobacteriota; c_Actinobacteria; o_Micromonosporales; f_Micromonosporaceae; g_unclassified_f_Micromonosporaceae	< 0.001
p_Actinobacteriota; c_Thermoleophilia; o_Solirubrobacterales; f_Solirubrobacteraceae; g_Solirubrobacter	< 0.001
p_Proteobacteria; c_Gammaproteobacteria; o_Burkholderiales; f_Comamonadaceae; g_unclassified_f_Comamonadaceae	< 0.001
p_Bacteroidota; c_Bacteroidia; o_Bacteroidales; f_Prevotellaceae; g_norank_f_Prevotellaceae	< 0.001
p_Actinobacteriota; c_Thermoleophilia; o_Solirubrobacterales; f_67-14; g_norank_f_67-14	< 0.001
p_Proteobacteria; c_Alphaproteobacteria; o_Acetobacteriales; f_Acetobacteraceae; g_Craurococcus-Caldovatus	< 0.001
p_Bacteroidota; c_Bacteroidia; o_Bacteroidales; f_Porphyrionadaceae; g_Porphyrionas	< 0.001
p_Firmicutes; c_Bacilli; o_Erysipelotrichales; f_Erysipelotrichaceae; g_Solobacterium	< 0.001
p_Actinobacteriota; c_Actinobacteria; o_Micrococcales; f_Brevibacteriaceae; g_Brevibacterium	< 0.001
p_Firmicutes; c_Bacilli; o_Erysipelotrichales; f_Erysipelotrichaceae; g_norank_f_Erysipelotrichaceae	< 0.001

p__Proteobacteria; c__Gammaproteobacteria; o__Xanthomonadales; f__Xanthomonadaceae; g__Luteimonas	< 0.001
p__Proteobacteria; c__Alphaproteobacteria; o__Sphingomonadales; f__Sphingomonadaceae; g__Sphingomonas	< 0.001
p__Firmicutes; c__Clostridia; o__Lachnospirales; f__unclassified_o__Lachnospirales; g__unclassified_o__Lachnospirales	< 0.001
p__Actinobacteriota; c__Actinobacteria; o__Corynebacteriales; f__Mycobacteriaceae; g__Mycobacterium	< 0.001
p__Proteobacteria; c__Alphaproteobacteria; o__Rhizobiales; f__Rhizobiaceae; g__Allorhizobium-Neorhizobium-Pararhizobium-Rhizobium	< 0.001
p__Firmicutes; c__Clostridia; o__Clostridiales; f__Clostridiaceae; g__unclassified_f__Clostridiaceae	< 0.001
p__Patescibacteria; c__Saccharimonadia; o__Saccharimonadales; f__Saccharimonadaceae; g__TM7a	< 0.001
p__Proteobacteria; c__Alphaproteobacteria; o__Rickettsiales; f__Mitochondria; g__norank_f__Mitochondria	< 0.001
p__Acidobacteriota; c__Vicinamibacteria; o__Vicinamibacterales; f__Vicinamibacteraceae; g__norank_f__Vicinamibacteraceae	< 0.001
p__Firmicutes; c__Negativicutes; o__Veillonellales-Selenomonadales; f__Veillonellaceae; g__Megasphaera	< 0.001
p__Actinobacteriota; c__Coriobacteriia; o__Coriobacteriales; f__Atopobiaceae; g__Atopobium	< 0.001
p__Proteobacteria; c__Alphaproteobacteria; o__Sphingomonadales; f__Sphingomonadaceae; g__Ellin6055	< 0.001
p__Firmicutes; c__Bacilli; o__Lactobacillales; f__Carnobacteriaceae; g__Desemzia	< 0.001
p__Firmicutes; c__Clostridia; o__Oscillospirales; f__Clostridium_methylpentosum_group; g__norank_f__Clostridium_methylpentosum_group	< 0.001
p__Proteobacteria; c__Alphaproteobacteria; o__Caulobacterales; f__Caulobacteraceae; g__Brevundimonas	< 0.001
p__Proteobacteria; c__Gammaproteobacteria; o__Burkholderiales; f__Sutterellaceae; g__Parasutterella	< 0.001
p__Proteobacteria; c__Gammaproteobacteria; o__Burkholderiales; f__Oxalobacteraceae; g__Massilia	< 0.001
p__Firmicutes; c__Bacilli; o__Lactobacillales; f__Carnobacteriaceae; g__Marinilactibacillus	< 0.001
p__Firmicutes; c__Bacilli; o__Lactobacillales; f__Carnobacteriaceae; g__Atopostipes	< 0.001
p__Proteobacteria; c__Alphaproteobacteria; o__Sphingomonadales; f__Sphingomonadaceae; g__unclassified_f__Sphingomonadaceae	< 0.001
p__Bacteroidota; c__Bacteroidia; o__Bacteroidales; f__Rikenellaceae; g__Rikenella	< 0.001
p__Actinobacteriota; c__Actinobacteria; o__Micrococcales; f__Microbacteriaceae; g__Microbacterium	< 0.001
p__Proteobacteria; c__Alphaproteobacteria; o__Rickettsiales; f__norank_o__Rickettsiales; g__norank_f__norank_o__Rickettsiales	< 0.001
p__Firmicutes; c__Clostridia; o__Lachnospirales; f__Defluviitaleaceae; g__Defluviitaleaceae_UCG-011	< 0.001
p__Actinobacteriota; c__Rubrobacteria; o__Rubrobacterales; f__Rubrobacteriaceae; g__Rubrobacter	< 0.001
p__Firmicutes; c__Bacilli; o__Bacillales; f__Planococcaceae; g__Lysinibacillus	< 0.001
p__Bacteroidota; c__Bacteroidia; o__Flavobacteriales; f__Weeksellaceae; g__Chryseobacterium	< 0.001
p__Actinobacteriota; c__Coriobacteriia; o__Coriobacteriales; f__Eggerthellaceae; g__DNF00809	< 0.001
p__Firmicutes; c__Bacilli; o__unclassified_c__Bacilli; f__unclassified_c__Bacilli; g__unclassified_c__Bacilli	< 0.001
p__Patescibacteria; c__Saccharimonadia; o__Saccharimonadales; f__norank_o__Saccharimonadales; g__norank_f__norank_o__Saccharimonadales	< 0.001
p__Acidobacteriota; c__Vicinamibacteria; o__Vicinamibacterales; f__norank_o__Vicinamibacterales; g__norank_f__norank_o__Vicinamibacterales	< 0.001
p__Actinobacteriota; c__Actinobacteria; o__Micrococcales; f__Micrococcaceae; g__Enteractinococcus	< 0.001
p__Actinobacteriota; c__Actinobacteria; o__Micrococcales; f__Dermatophilaceae; g__unclassified_f__Dermatophilaceae	< 0.001
p__Firmicutes; c__Bacilli; o__Erysipelotrichales; f__Erysipelatoclostridiaceae; g__Coprobacillus	< 0.001
p__Actinobacteriota; c__Actinobacteria; o__Micrococcales; f__Dermabacteraceae; g__unclassified_f__Dermabacteraceae	< 0.001
p__Actinobacteriota; c__Actinobacteria; o__Micrococcales; f__Dermabacteraceae; g__Brachy bacterium	< 0.001
p__Firmicutes; c__Negativicutes; o__Veillonellales-Selenomonadales; f__unclassified_o__Veillonellales-Selenomonadales; g__unclassified_o__Veillonellales-Selenomonadales	< 0.001

p__Proteobacteria; c__Gammaproteobacteria; o__Burkholderiales; f__Comamonadaceae; g__Comamonas	< 0.001
p__Firmicutes; c__Clostridia; o__Peptostreptococcales-Tissierellales; f__Anaerovoracaceae; g__Mogibacterium	< 0.001
p__Firmicutes; c__Clostridia; o__Oscillospirales; f__norank_o__Oscillospirales; g__Hydrogenoanaerobacterium	< 0.001
p__Actinobacteriota; c__Actinobacteria; o__Propionibacteriales; f__Propionibacteriaceae; g__Microlunatus	< 0.001
p__Actinobacteriota; c__Acidimicrobiia; o__Microtrichales; f__norank_o__Microtrichales; g__norank_f__norank_o__Microtrichales	< 0.001
p__Proteobacteria; c__Alphaproteobacteria; o__Sphingomonadales; f__Sphingomonadaceae; g__Novosphingobium	< 0.001
p__Actinobacteriota; c__Coriobacteriia; o__Coriobacteriales; f__Atopobiaceae; g__unclassified_f__Atopobiaceae	< 0.001
p__Proteobacteria; c__Alphaproteobacteria; o__Rhodobacterales; f__Rhodobacteraceae; g__Amaricoccus	< 0.001
p__Proteobacteria; c__Alphaproteobacteria; o__Rhizobiales; f__Beijerinckiaceae; g__Microvirga	< 0.001
p__Proteobacteria; c__Alphaproteobacteria; o__Tistrellales; f__Geminicoccaceae; g__norank_f__Geminicoccaceae	< 0.001
p__Firmicutes; c__Bacilli; o__Erysipelotrichales; f__Erysipelatoclostridiaceae; g__UCG-004	< 0.001
p__Firmicutes; c__Clostridia; o__Peptostreptococcales-Tissierellales; f__Anaerovoracaceae; g__Eubacterium_nodatum_group	< 0.001
p__Bacteroidota; c__Bacteroidia; o__Bacteroidales; f__Marinifilaceae; g__Butyricimonas	< 0.001
p__Proteobacteria; c__Alphaproteobacteria; o__Rhizobiales; f__Xanthobacteraceae; g__norank_f__Xanthobacteraceae	< 0.001
p__Actinobacteriota; c__Actinobacteria; o__Micromonosporales; f__Micromonosporaceae; g__Actinoplanes	< 0.001
p__Firmicutes; c__Clostridia; o__Peptococcales; f__Peptococcaceae; g__Peptococcus	< 0.001
p__Actinobacteriota; c__Actinobacteria; o__Micrococcales; f__Intrasporangiaceae; g__Ornithinimicrobium	< 0.001
p__Firmicutes; c__Bacilli; o__Erysipelotrichales; f__Erysipelotrichaceae; g__Allobaculum	< 0.001
p__Firmicutes; c__Bacilli; o__Bacillales; f__Planococcaceae; g__Planococcus	< 0.001
p__Chloroflexi; c__Anaerolineae; o__SBR1031; f__A4b; g__norank_f__A4b	< 0.001
p__Actinobacteriota; c__Actinobacteria; o__Corynebacteriales; f__Dietziaceae; g__Dietzia	< 0.001
p__Actinobacteriota; c__Actinobacteria; o__Frankiales; f__Geodermatophilaceae; g__unclassified_f__Geodermatophilaceae	< 0.001
p__Verrucomicrobiota; c__Lentisphaeria; o__Victivallales; f__vadinBE97; g__norank_f__vadinBE97	< 0.001
p__Actinobacteriota; c__Actinobacteria; o__Micrococcales; f__Cellulomonadaceae; g__Cellulomonas	< 0.001
p__Proteobacteria; c__Gammaproteobacteria; o__Burkholderiales; f__Burkholderiaceae; g__Ralstonia	< 0.001
p__Firmicutes; c__Bacilli; o__Erysipelotrichales; f__Erysipelatoclostridiaceae; g__unclassified_f__Erysipelatoclostridiaceae	< 0.001
p__Cyanobacteria; c__Cyanobacteriia; o__Cyanobacteriales; f__Chroococcidiopsaceae; g__Chroococcidiopsis_SAG_2023	< 0.001
p__Bacteroidota; c__Bacteroidia; o__Bacteroidales; f__Prevotellaceae; g__Prevotellaceae_UCG-003	< 0.001
p__Actinobacteriota; c__Actinobacteria; o__Micrococcales; f__Microbacteriaceae; g__unclassified_f__Microbacteriaceae	< 0.001
p__Firmicutes; c__Negativicutes; o__Veillonellales-Selenomonadales; f__Selenomonadaceae; g__unclassified_f__Selenomonadaceae	< 0.001
p__Firmicutes; c__Bacilli; o__Staphylococcales; f__Staphylococcaceae; g__Salinicoccus	< 0.001
p__Firmicutes; c__unclassified_p__Firmicutes; o__unclassified_p__Firmicutes; f__unclassified_p__Firmicutes; g__unclassified_p__Firmicutes	< 0.001
p__Firmicutes; c__Bacilli; o__Lactobacillales; f__Aerococcaceae; g__Aerococcus	< 0.001
p__Proteobacteria; c__Alphaproteobacteria; o__Rhizobiales; f__Beijerinckiaceae; g__Methylobacterium-Methylorubrum	< 0.001
p__Actinobacteriota; c__Thermoleophilia; o__Gaiellales; f__norank_o__Gaiellales; g__norank_f__norank_o__Gaiellales	< 0.001
p__Actinobacteriota; c__Actinobacteria; o__Corynebacteriales; f__Nocardiaceae; g__unclassified_f__Nocardiaceae	< 0.001
p__Proteobacteria; c__Gammaproteobacteria; o__Steroidobacterales; f__Steroidobacteraceae; g__Steroidobacter	< 0.001
p__Acidobacteriota; c__Blastocatellia; o__Pyrinomonadales; f__Pyrinomonadaceae; g__RB41	< 0.001
p__Proteobacteria; c__Alphaproteobacteria; o__Rhodobacterales; f__Rhodobacteraceae; g__Rubellimicrobium	< 0.001
p__Spirochaetota; c__Spirochaetia; o__Spirochaetales; f__Spirochaetaceae; g__unclassified_f__Spirochaetaceae	< 0.001
p__Actinobacteriota; c__Actinobacteria; o__Propionibacteriales; f__Nocardioidaceae; g__unclassified_f__Nocardioidaceae	< 0.001

p__Myxococcota; c__Polyangia; o__Haliangiales; f__Haliangiaceae; g__Haliangium	< 0.001
p__Campilobacterota; c__Campylobacteria; o__Campylobacterales; f__Helicobacteraceae; g__Helicobacter	< 0.001
p__Actinobacteriota; c__Acidimicrobiia; o__Microtrichales; f__Ilumatobacteraceae; g__unclassified_f__Ilumatobacteraceae	< 0.001
p__Bacteroidota; c__Bacteroidia; o__Bacteroidales; f__Muribaculaceae; g__unclassified_f__Muribaculaceae	< 0.001
p__Proteobacteria; c__Alphaproteobacteria; o__Rhizobiales; f__Hyphomicrobiaceae; g__Pedomicrobium	< 0.001
p__Actinobacteriota; c__Acidimicrobiia; o__Actinomarinales; f__norank_o__Actinomarinales; g__norank_f__norank_o__Actinomarinales	< 0.001
p__Actinobacteriota; c__Actinobacteria; o__Micrococcales; f__Microbacteriaceae; g__Agrococcus	< 0.001
p__Actinobacteriota; c__Actinobacteria; o__Micromonosporales; f__Micromonosporaceae; g__Virgisporangium	< 0.001
p__Actinobacteriota; c__Actinobacteria; o__Pseudonocardiales; f__Pseudonocardiaceae; g__Actinomycetospora	< 0.001
p__Proteobacteria; c__Alphaproteobacteria; o__Sphingomonadales; f__Sphingomonadaceae; g__Altererythrobacter	< 0.001
p__Bacteroidota; c__Bacteroidia; o__Bacteroidales; f__Prevotellaceae; g__Alloprevotella	< 0.001
p__Deinococcota; c__Deinococci; o__Deinococcales; f__Trueperaceae; g__Truepera	< 0.001
p__Acidobacteriota; c__Acidobacteriae; o__Bryobacteriales; f__Bryobacteraceae; g__Bryobacter	< 0.001
p__Proteobacteria; c__Gammaproteobacteria; o__Burkholderiales; f__Oxalobacteraceae; g__Herbaspirillum	< 0.001
p__Cyanobacteria; c__Cyanobacteriia; o__Cyanobacteriales; f__unclassified_o__Cyanobacteriales; g__unclassified_o__Cyanobacteriales	< 0.001
p__Proteobacteria; c__Alphaproteobacteria; o__Rickettsiales; f__unclassified_o__Rickettsiales; g__unclassified_o__Rickettsiales	< 0.001
p__Proteobacteria; c__Alphaproteobacteria; o__Caulobacterales; f__Caulobacteraceae; g__norank_f__Caulobacteraceae	< 0.001
p__Actinobacteriota; c__Actinobacteria; o__Streptomycetales; f__Streptomycetaceae; g__Streptomyces	< 0.001
p__Chloroflexi; c__Chloroflexia; o__Thermomicrobiales; f__AKYG1722; g__norank_f__AKYG1722	< 0.001
p__Actinobacteriota; c__Actinobacteria; o__Pseudonocardiales; f__Pseudonocardiaceae; g__Crossiella	< 0.001
p__Myxococcota; c__Polyangia; o__Polyangiales; f__Sandaracinaceae; g__norank_f__Sandaracinaceae	< 0.001
p__Proteobacteria; c__Gammaproteobacteria; o__Xanthomonadales; f__Xanthomonadaceae; g__norank_f__Xanthomonadaceae	< 0.001
p__Chloroflexi; c__Chloroflexia; o__Chloroflexales; f__Roseiflexaceae; g__norank_f__Roseiflexaceae	< 0.001
p__Proteobacteria; c__Alphaproteobacteria; o__Rhizobiales; f__norank_o__Rhizobiales; g__norank_f__norank_o__Rhizobiales	< 0.001
p__Proteobacteria; c__Alphaproteobacteria; o__Tistrellales; f__Geminicoccaceae; g__Candidatus_Alysiosphaera	< 0.001
p__Actinobacteriota; c__Acidimicrobiia; o__Microtrichales; f__Iamiaceae; g__Iamia	< 0.001
p__Bacteroidota; c__Bacteroidia; o__Sphingobacteriales; f__Sphingobacteriaceae; g__Pedobacter	< 0.001
p__Gemmatimonadota; c__Longimicrobia; o__Longimicrobiales; f__Longimicrobiaceae; g__norank_f__Longimicrobiaceae	< 0.001
p__Proteobacteria; c__Alphaproteobacteria; o__Rhizobiales; f__Xanthobacteraceae; g__Bradyrhizobium	< 0.001
p__Firmicutes; c__Bacilli; o__Erysipelotrichales; f__Erysipelotrichaceae; g__Catenisphaera	< 0.001
p__Actinobacteriota; c__Actinobacteria; o__Micromonosporales; f__Micromonosporaceae; g__Dactylosporangium	< 0.001
p__Proteobacteria; c__Alphaproteobacteria; o__Caulobacterales; f__Hyphomonadaceae; g__Hirschia	< 0.001
p__Firmicutes; c__Clostridia; o__Peptostreptococcales-Tissierellales; f__norank_o__Peptostreptococcales-Tissierellales; g__Helcococcus	< 0.001
p__Actinobacteriota; c__Actinobacteria; o__Micrococcales; f__Cellulomonadaceae; g__Pseudactinotalea	< 0.001
p__Proteobacteria; c__Alphaproteobacteria; o__Rhizobiales; f__Xanthobacteraceae; g__unclassified_f__Xanthobacteraceae	< 0.001
p__Firmicutes; c__Bacilli; o__Lactobacillales; f__Streptococcaceae; g__Streptococcus	< 0.001
p__Actinobacteriota; c__Actinobacteria; o__Micrococcales; f__Micrococcaceae; g__Garicola	< 0.001
p__Chloroflexi; c__Chloroflexia; o__Kallotenuales; f__AKIW781; g__norank_f__AKIW781	< 0.001
p__Chloroflexi; c__JG30-KF-CM66; o__norank_c__JG30-KF-CM66; f__norank_o__norank_c__JG30-KF-CM66; g__norank_f__norank_o__norank_c__JG30-KF-CM66	< 0.001

p__Firmicutes; c__Bacilli; o__Lactobacillales; f__Aerococcaceae; g__unclassified_f__Aerococcaceae	< 0.001
p__Bacteroidota; c__Bacteroidia; o__Bacteroidales; f__Bacteroidales_RF16_group; g__norank_f__Bacteroidales_RF16_group	< 0.001
p__Proteobacteria; c__Alphaproteobacteria; o__Rhizobiales; f__Beijerinckiaceae; g__Bosea	< 0.001
p__Chloroflexi; c__Chloroflexia; o__Chloroflexales; f__Chloroflexaceae; g__FFCH7168	< 0.001
p__Proteobacteria; c__Alphaproteobacteria; o__Reyranelles; f__Reyranelleaceae; g__Reyranela	< 0.001
p__Chloroflexi; c__TK10; o__norank_c__TK10; f__norank_o__norank_c__TK10; g__norank_f__norank_o__norank_c__TK10	< 0.001
p__Proteobacteria; c__Alphaproteobacteria; o__Rhizobiales; f__Rhizobiaceae; g__Mesorhizobium	< 0.001
p__Proteobacteria; c__Alphaproteobacteria; o__Rhizobiales; f__Beijerinckiaceae; g__Psychroglaciecola	< 0.001
p__Nitrospirota; c__Nitrospiria; o__Nitrospirales; f__Nitrospiraceae; g__Nitrospira	< 0.001
p__Proteobacteria; c__Alphaproteobacteria; o__Rhodobacterales; f__Rhodobacteraceae; g__Limibaculum	< 0.001
p__Proteobacteria; c__Gammaproteobacteria; o__Burkholderiales; f__Oxalobacteraceae; g__Duganella	< 0.001
p__Acidobacteriota; c__Blastocatellia; o__Blastocatellales; f__Blastocatellaceae; g__Blastocatella	< 0.001
p__Actinobacteriota; c__Actinobacteria; o__Micrococcales; f__Intrasporangiaceae; g__unclassified_f__Intrasporangiaceae	< 0.001
p__Bacteroidota; c__Bacteroidia; o__Sphingobacteriales; f__Sphingobacteriaceae; g__Sphingobacterium	< 0.001
p__Acidobacteriota; c__Vicinamibacteria; o__Vicinamibacterales; f__Vicinamibacteraceae; g__unclassified_f__Vicinamibacteraceae	< 0.001
p__Proteobacteria; c__Alphaproteobacteria; o__Acetobacterales; f__Acetobacteraceae; g__Roseomonas	< 0.001
p__Proteobacteria; c__Alphaproteobacteria; o__Acetobacterales; f__Acetobacteraceae; g__unclassified_f__Acetobacteraceae	< 0.001
p__Actinobacteriota; c__Actinobacteria; o__Frankiales; f__Geodermatophilaceae; g__Blastococcus	< 0.001
p__Firmicutes; c__Negativicutes; o__Acidaminococcales; f__Acidaminococcaceae; g__Acidaminococcus	< 0.001
p__Firmicutes; c__Bacilli; o__Erysipelotrichales; f__Erysipelatoclostridiaceae; g__Erysipelatoclostridium	< 0.001
p__Actinobacteriota; c__Actinobacteria; o__Propionibacteriales; f__Nocardoidaceae; g__Aeromicrobium	< 0.001
p__Proteobacteria; c__Alphaproteobacteria; o__Rhizobiales; f__Amb-16S-1323; g__norank_f__Amb-16S-1323	< 0.001
p__Proteobacteria; c__Alphaproteobacteria; o__Rhizobiales; f__D05-2; g__norank_f__D05-2	< 0.001
p__Firmicutes; c__Negativicutes; o__Veillonellales-Selenomonadales; f__Veillonellaceae; g__Negativicoccus	< 0.001
p__Firmicutes; c__Bacilli; o__Bacillales; f__Planococcaceae; g__Psychrobacillus	< 0.001
p__Firmicutes; c__Clostridia; o__Peptostreptococcales-Tissierellales; f__norank_o__Peptostreptococcales-Tissierellales; g__Peptoniphilus	< 0.001
p__Actinobacteriota; c__Acidimicrobiia; o__IMCC26256; f__norank_o__IMCC26256; g__norank_f__norank_o__IMCC26256	< 0.001
p__Gemmatimonadota; c__Gemmatimonadetes; o__Gemmatimonadales; f__Gemmatimonadaceae; g__Gemmatimonas	< 0.001
p__Firmicutes; c__Bacilli; o__Lactobacillales; f__unclassified_o__Lactobacillales; g__unclassified_o__Lactobacillales	< 0.001
p__Actinobacteriota; c__Actinobacteria; o__Micrococcales; f__Microbacteriaceae; g__Agromyces	< 0.001
p__Firmicutes; c__Bacilli; o__Staphylococcales; f__Staphylococcaceae; g__Aliicoccus	< 0.001
p__Actinobacteriota; c__Actinobacteria; o__Propionibacteriales; f__Propionibacteriaceae; g__unclassified_f__Propionibacteriaceae	< 0.001
p__Actinobacteriota; c__Thermoleophilia; o__Gaiellales; f__Gaiellaceae; g__Gaiella	< 0.001
p__Actinobacteriota; c__Actinobacteria; o__Euzebyales; f__Euzebyaceae; g__norank_f__Euzebyaceae	< 0.001
p__Proteobacteria; c__Alphaproteobacteria; o__Caulobacterales; f__Hyphomonadaceae; g__SWB02	< 0.001
p__Chloroflexi; c__Gitt-GS-136; o__norank_c__Gitt-GS-136; f__norank_o__norank_c__Gitt-GS-136; g__norank_f__norank_o__norank_c__Gitt-GS-136	< 0.001
p__Actinobacteriota; c__Actinobacteria; o__Streptosporangiales; f__Thermomonosporaceae; g__Actinomadura	< 0.001
p__Proteobacteria; c__Gammaproteobacteria; o__Burkholderiales; f__TRA3-20; g__norank_f__TRA3-20	< 0.001
p__Firmicutes; c__Bacilli; o__Lactobacillales; f__Streptococcaceae; g__Lactococcus	< 0.001
p__Proteobacteria; c__Gammaproteobacteria; o__Burkholderiales; f__Burkholderiaceae; g__Limnobacter	< 0.001
p__Cyanobacteria; c__Cyanobacteriia; o__Cyanobacteriales; f__Nostocaceae; g__unclassified_f__Nostocaceae	< 0.001

p__Proteobacteria; c__Alphaproteobacteria; o__Caulobacterales; f__Caulobacteraceae; g__Phenylobacterium	< 0.001
p__Firmicutes; c__Bacilli; o__Lactobacillales; f__Aerococcaceae; g__Facklamia	< 0.001
p__Proteobacteria; c__Alphaproteobacteria; o__Rhizobiales; f__Rhodomicrobiaceae; g__Rhodomicrobium	< 0.001
p__Firmicutes; c__Negativicutes; o__Veillonellales-Selenomonadales; f__Selenomonadaceae; g__Selenomonas	< 0.001
p__Firmicutes; c__Clostridia; o__Peptostreptococcales-Tissierellales; f__norank_o__Peptostreptococcales-Tissierellales; g__Ezakiella	< 0.001
p__Acidobacteriota; c__Blastocatellia; o__11-24; f__norank_o__11-24; g__norank_f__norank_o__11-24	< 0.001
p__Proteobacteria; c__Gammaproteobacteria; o__PLTA13; f__norank_o__PLTA13; g__norank_f__norank_o__PLTA13	< 0.001
p__Elusimicrobiota; c__Elusimicrobia; o__Elusimicrobiales; f__Elusimicrobiaceae; g__Elusimicrobium	< 0.001
p__Gemmatimonadota; c__Gemmatimonadetes; o__Gemmatimonadales; f__Gemmatimonadaceae; g__norank_f__Gemmatimonadaceae	< 0.001
p__Proteobacteria; c__Gammaproteobacteria; o__Burkholderiales; f__Comamonadaceae; g__Pseudorhodofera	< 0.001
p__Bacteroidota; c__Bacteroidia; o__Flavobacteriales; f__Weeksellaceae; g__norank_f__Weeksellaceae	< 0.001
p__Actinobacteriota; c__Acidimicrobiia; o__unclassified_c__Acidimicrobiia; f__unclassified_c__Acidimicrobiia; g__unclassified_c__Acidimicrobiia	< 0.001
p__Proteobacteria; c__Alphaproteobacteria; o__Rhizobiales; f__Rhizobiaceae; g__Aureimonas	< 0.001
p__Actinobacteriota; c__Actinobacteria; o__Kineosporiales; f__Kineosporiaceae; g__Kineosporia	< 0.001
p__Firmicutes; c__Bacilli; o__Lactobacillales; f__Carnobacteriaceae; g__norank_f__Carnobacteriaceae	< 0.001
p__Firmicutes; c__Clostridia; o__Peptostreptococcales-Tissierellales; f__norank_o__Peptostreptococcales-Tissierellales; g__Anaerococcus	< 0.001
p__Actinobacteriota; c__Actinobacteria; o__Frankiales; f__Sporichthyaceae; g__norank_f__Sporichthyaceae	< 0.001
p__Proteobacteria; c__Alphaproteobacteria; o__Reyranelles; f__Reyranelleaceae; g__norank_f__Reyranelleaceae	< 0.001
p__Proteobacteria; c__Alphaproteobacteria; o__Caulobacterales; f__Caulobacteraceae; g__Caulobacter	< 0.001
p__Chloroflexi; c__Anaerolineae; o__Caldilineales; f__Caldilineaceae; g__norank_f__Caldilineaceae	< 0.001
p__Proteobacteria; c__Gammaproteobacteria; o__Xanthomonadales; f__Xanthomonadaceae; g__Lysobacter	< 0.001
p__Myxococcota; c__bacteriap25; o__norank_c__bacteriap25; f__norank_o__norank_c__bacteriap25; g__norank_f__norank_o__norank_c__bacteriap25	< 0.001
p__Planctomycetota; c__Planctomycetes; o__Pirellulales; f__Pirellulaceae; g__p-1088-a5_gut_group	< 0.001
p__Acidobacteriota; c__Blastocatellia; o__Blastocatellales; f__Blastocatellaceae; g__norank_f__Blastocatellaceae	< 0.001
p__Proteobacteria; c__Gammaproteobacteria; o__CCD24; f__norank_o__CCD24; g__norank_f__norank_o__CCD24	< 0.001
p__Firmicutes; c__Clostridia; o__Oscillospirales; f__Butyricicoccaceae; g__Butyricicoccus	< 0.001
p__Actinobacteriota; c__Actinobacteria; o__Actinomycetales; f__Actinomycetaceae; g__Varibaculum	< 0.001
p__Actinobacteriota; c__Actinobacteria; o__Micrococcales; f__Promicromonosporaceae; g__Isopticola	< 0.001
p__Verrucomicrobiota; c__Lentisphaeria; o__Victivallales; f__Victivallaceae; g__Victivallis	< 0.001
p__Myxococcota; c__Myxococcia; o__Myxococcales; f__Myxococcaceae; g__unclassified_f__Myxococcaceae	< 0.001
p__Chloroflexi; c__Dehalococcoidia; o__S085; f__norank_o__S085; g__norank_f__norank_o__S085	< 0.001
p__Chloroflexi; c__Anaerolineae; o__SBR1031; f__A4b; g__OLB13	< 0.001
p__Actinobacteriota; c__Thermoleophilia; o__Solirubrobacterales; f__Solirubrobacteraceae; g__unclassified_f__Solirubrobacteraceae	< 0.001
p__Bacteroidota; c__Bacteroidia; o__Cytophagales; f__Microscillaceae; g__norank_f__Microscillaceae	< 0.001
p__Proteobacteria; c__Alphaproteobacteria; o__Rhizobiales; f__Devosiaceae; g__Pelagibacterium	< 0.001
p__Acidobacteriota; c__Viciniabacteria; o__Viciniabacteriales; f__Viciniabacteraceae; g__Viciniabacter	< 0.001
p__Proteobacteria; c__Gammaproteobacteria; o__Burkholderiales; f__Hydrogenophilaceae; g__Thiobacillus	< 0.001

p__Firmicutes; c__Clostridia; o__Peptostreptococcales-Tissierellales; f__Anaerovoracaceae; g__S5-A14a	< 0.001
p__Firmicutes; c__Bacilli; o__Bacillales; f__Bacillaceae; g__Oceanobacillus	< 0.001
p__Actinobacteriota; c__Thermoleophilia; o__Gaiellales; f__unclassified_o__Gaiellales; g__unclassified_o__Gaiellales	< 0.001
p__Actinobacteriota; c__Actinobacteria; o__Kineosporiales; f__Kineosporiaceae; g__unclassified_f__Kineosporiaceae	< 0.001
p__Firmicutes; c__Bacilli; o__Staphylococcales; f__Staphylococcaceae; g__Staphylococcus	< 0.001
p__Proteobacteria; c__Gammaproteobacteria; o__Xanthomonadales; f__Xanthomonadaceae; g__unclassified_f__Xanthomonadaceae	< 0.001
p__Cyanobacteria; c__Cyanobacteriia; o__norank_c__Cyanobacteriia; f__norank_o__norank_c__Cyanobacteriia; g__norank_f__norank_o__norank_c__Cyanobacteriia	< 0.001
p__Proteobacteria; c__Alphaproteobacteria; o__Rhizobiales; f__Rhizobiaceae; g__Shinella	< 0.001
p__Deinococcota; c__Deinococci; o__Deinococcales; f__Deinococcaceae; g__Deinococcus	< 0.001
p__Chloroflexi; c__KD4-96; o__norank_c__KD4-96; f__norank_o__norank_c__KD4-96; g__norank_f__norank_o__norank_c__KD4-96	< 0.001
p__Entotheonellaota; c__Entotheonellia; o__Entotheonellales; f__Entotheonellaceae; g__Candidatus_Entotheonella	< 0.001
p__Actinobacteriota; c__Actinobacteria; o__Pseudonocardiales; f__Pseudonocardiaceae; g__Actinophytocola	< 0.001
p__Firmicutes; c__Negativicutes; o__Acidaminococcales; f__Acidaminococcaceae; g__Phascolarctobacterium	< 0.001
p__Actinobacteriota; c__Acidimicrobiia; o__Microtrichales; f__Ilumatobacteraceae; g__Ilumatobacter	< 0.001
p__Bacteroidota; c__Bacteroidia; o__Flavobacteriales; f__Flavobacteriaceae; g__norank_f__Flavobacteriaceae	< 0.001
p__Actinobacteriota; c__Actinobacteria; o__Propionibacteriales; f__Nocardiodaceae; g__Marmoricola	< 0.001
p__Proteobacteria; c__Alphaproteobacteria; o__Rhizobiales; f__Rhizobiaceae; g__Ochrobactrum	< 0.001
p__Actinobacteriota; c__Actinobacteria; o__Frankiales; f__Cryptosporangiaceae; g__Cryptosporangium	< 0.001
p__Proteobacteria; c__Gammaproteobacteria; o__Xanthomonadales; f__Xanthomonadaceae; g__Pseudoxanthomonas	< 0.001
p__Patescibacteria; c__Saccharimonadia; o__Saccharimonadales; f__unclassified_o__Saccharimonadales; g__unclassified_o__Saccharimonadales	< 0.001
p__Firmicutes; c__Bacilli; o__Lactobacillales; f__Carnobacteriaceae; g__Jeotgalibaca	< 0.001
p__Proteobacteria; c__Gammaproteobacteria; o__Gammaproteobacteria_Incertae_Sedis; f__unclassified_o__Gammaproteobacteria_Incertae_Sedis; g__Acidibacter	< 0.001
p__Cyanobacteria; c__Cyanobacteriia; o__Cyanobacteriales; f__Chroococcidiopsaceae; g__norank_f__Chroococcidiopsaceae	< 0.001
p__Actinobacteriota; c__Acidimicrobiia; o__Microtrichales; f__Microtrichaceae; g__norank_f__Microtrichaceae	< 0.001
p__Proteobacteria; c__Alphaproteobacteria; o__norank_c__Alphaproteobacteria; f__norank_o__norank_c__Alphaproteobacteria; g__norank_f__norank_o__norank_c__Alphaproteobacteria	< 0.001
p__Firmicutes; c__Bacilli; o__Bacillales; f__Bacillaceae; g__unclassified_f__Bacillaceae	< 0.001
p__Cyanobacteria; c__Cyanobacteriia; o__Cyanobacteriales; f__Phormidiaceae; g__Tychonema_CCAP_1459-11B	< 0.001
p__Bacteroidota; c__Bacteroidia; o__Cytophagales; f__Cyclobacteriaceae; g__norank_f__Cyclobacteriaceae	< 0.001
p__Proteobacteria; c__Alphaproteobacteria; o__Sphingomonadales; f__Sphingomonadaceae; g__norank_f__Sphingomonadaceae	< 0.001
p__Firmicutes; c__Bacilli; o__Erysipelotrichales; f__Erysipelotrichaceae; g__Dubosiella	< 0.001
p__Bacteroidota; c__Bacteroidia; o__Chitinophagales; f__Chitinophagaceae; g__Flavisolibacter	< 0.001
p__Actinobacteriota; c__Actinobacteria; o__Propionibacteriales; f__Nocardiodaceae; g__Mumia	< 0.001
p__Methylomirabilota; c__Methylomirabilia; o__Rokubacteriales; f__norank_o__Rokubacteriales; g__norank_f__norank_o__Rokubacteriales	< 0.001
p__Proteobacteria; c__Alphaproteobacteria; o__Rhizobiales; f__Devosiaceae; g__norank_f__Devosiaceae	< 0.001
p__Acidobacteriota; c__Viciniabacteria; o__Viciniabacteriales; f__Viciniabacteraceae; g__Luteitalea	< 0.001
p__Actinobacteriota; c__Actinobacteria; o__Kineosporiales; f__Kineosporiaceae; g__Pseudokineococcus	< 0.001
p__Bacteroidota; c__Rhodothermia; o__Rhodothermales; f__Rhodothermaceae; g__Rubrivirga	< 0.001

p__Proteobacteria; c__Alphaproteobacteria; o__Rhizobiales; f__Hyphomicrobiaceae; g__Hyphomicrobium	< 0.001
p__Actinobacteriota; c__Actinobacteria; o__Pseudonocardiales; f__Pseudonocardiaceae; g__Saccharopolyspora	< 0.001
p__Proteobacteria; c__Alphaproteobacteria; o__Tistrellales; f__Geminicoccaceae; g__Geminicoccus	< 0.001
p__Proteobacteria; c__Gammaproteobacteria; o__Cellvibrionales; f__Cellvibrionaceae; g__Cellvibrio	< 0.001
p__Proteobacteria; c__Alphaproteobacteria; o__Rhizobiales; f__A0839; g__norank_f__A0839	< 0.001
p__Proteobacteria; c__Gammaproteobacteria; o__Burkholderiales; f__Sutterellaceae; g__Sutterella	< 0.001
p__Chloroflexi; c__Chloroflexia; o__Chloroflexales; f__Chloroflexaceae; g__unclassified_f__Chloroflexaceae	< 0.001
p__Acidobacteriota; c__Blastocatellia; o__Blastocatellales; f__Blastocatellaceae; g__Stenotrophobacter	< 0.001
p__Actinobacteriota; c__Actinobacteria; o__Corynebacteriales; f__Nocardiaceae; g__Gordonia	< 0.001
p__Actinobacteriota; c__Actinobacteria; o__Streptosporangiales; f__Thermomonosporaceae; g__Actinocorallia	< 0.001
p__Chloroflexi; c__Chloroflexia; o__Chloroflexales; f__Chloroflexaceae; g__Candidatus_Chloroploca	< 0.001
p__Proteobacteria; c__Gammaproteobacteria; o__Burkholderiales; f__Burkholderiaceae; g__Cupriavidus	< 0.001
p__Myxococcota; c__Polyangia; o__Polyangiales; f__Brii41; g__norank_f__Brii41	< 0.001
p__Bacteroidota; c__Bacteroidia; o__Chitinophagales; f__Chitinophagaceae; g__unclassified_f__Chitinophagaceae	< 0.001
p__Actinobacteriota; c__Actinobacteria; o__Streptosporangiales; f__Nocardiopsaceae; g__Nocardiopsis	< 0.001
p__Firmicutes; c__Bacilli; o__Lactobacillales; f__Enterococcaceae; g__Enterococcus	< 0.001
p__Proteobacteria; c__Alphaproteobacteria; o__Dongiiales; f__Dongiaceae; g__Dongia	< 0.001
p__Firmicutes; c__Bacilli; o__Paenibacillales; f__Paenibacillaceae; g__Paenibacillus	< 0.001
p__Gemmatimonadota; c__S0134_terrestrial_group; o__norank_c__S0134_terrestrial_group; f__norank_o__norank_c__S0134_terrestrial_group; g__norank_f__norank_o__norank_c__S0134_terrestrial_group	< 0.001
p__Proteobacteria; c__Gammaproteobacteria; o__Burkholderiales; f__Burkholderiaceae; g__Lautropia	< 0.001
p__Proteobacteria; c__Alphaproteobacteria; o__Rhizobiales; f__Rhizobiales_Incertae_Sedis; g__Nordella	< 0.001
p__Actinobacteriota; c__Actinobacteria; o__Actinomycetales; f__Actinomycetaceae; g__Actinomyces	< 0.001
p__Entotheonellaota; c__Entotheonellia; o__Entotheonellales; f__Entotheonellaceae; g__norank_f__Entotheonellaceae	< 0.001
p__Proteobacteria; c__Gammaproteobacteria; o__Burkholderiales; f__Alcaligenaceae; g__Verticiella	< 0.001
p__Bacteroidota; c__Bacteroidia; o__Cytophagales; f__Cytophagaceae; g__Siphonobacter	< 0.001
p__Actinobacteriota; c__MB-A2-108; o__norank_c__MB-A2-108; f__norank_o__norank_c__MB-A2-108; g__norank_f__norank_o__norank_c__MB-A2-108	< 0.001
p__Actinobacteriota; c__Actinobacteria; o__Frankiales; f__Geodermatophilaceae; g__Antricoccus	< 0.001
p__Bacteroidota; c__Bacteroidia; o__Cytophagales; f__Cytophagaceae; g__Rhodocytophaga	< 0.001
p__Actinobacteriota; c__Thermoleophilia; o__Solirubrobacterales; f__unclassified_o__Solirubrobacterales; g__unclassified_o__Solirubrobacterales	< 0.001
p__Firmicutes; c__Syntrophomonadia; o__Syntrophomonadales; f__Syntrophomonadaceae; g__unclassified_f__Syntrophomonadaceae	< 0.001
p__Proteobacteria; c__Gammaproteobacteria; o__Burkholderiales; f__Nitrosomonadaceae; g__Nitrospira	< 0.001
p__Proteobacteria; c__Alphaproteobacteria; o__Puniceispirillales; f__Puniceispirillales_Incertae_Sedis; g__Constrictibacter	< 0.001
p__Chloroflexi; c__Chloroflexia; o__Chloroflexales; f__Chloroflexaceae; g__Oscillochloris	< 0.001
p__Bacteroidota; c__Bacteroidia; o__Cytophagales; f__Spirosomaceae; g__Larkinella	< 0.001
p__Bacteroidota; c__Rhodothermia; o__Rhodothermales; f__Rhodothermaceae; g__norank_f__Rhodothermaceae	< 0.001
p__Bacteroidota; c__Bacteroidia; o__Bacteroidales; f__Rikenellaceae; g__dgA-11_gut_group	< 0.001
p__Actinobacteriota; c__Actinobacteria; o__Kineosporiales; f__Kineosporiaceae; g__Quadrisphaera	< 0.001
p__Chloroflexi; c__unclassified_p__Chloroflexi; o__unclassified_p__Chloroflexi; f__unclassified_p__Chloroflexi; g__unclassified_p__Chloroflexi	< 0.001

p__Proteobacteria; c__Gammaproteobacteria; o__Burkholderiales; f__Comamonadaceae; g__Rhizobacter	< 0.001
p__Actinobacteriota; c__Actinobacteria; o__Frankiales; f__unclassified_o__Frankiales; g__unclassified_o__Frankiales	< 0.001
p__Bacteroidota; c__Bacteroidia; o__Cytophagales; f__Cyclobacteriaceae; g__unclassified_f__Cyclobacteriaceae	< 0.001
p__Bacteroidota; c__Bacteroidia; o__Chitinophagales; f__Chitinophagaceae; g__Puia	< 0.001
p__Acidobacteriota; c__Aminicenantia; o__Aminicenantales; f__norank_o__Aminicenantales; g__norank_f__norank_o__Aminicenantales	< 0.001
p__Planctomycetota; c__Planctomycetes; o__Gemmatales; f__Gemmataceae; g__norank_f__Gemmataceae	< 0.001
p__Firmicutes; c__Bacilli; o__Erysipelotrichales; f__Erysipelatoclostridiaceae; g__Sharpea	< 0.001
p__Chloroflexi; c__Chloroflexia; o__Chloroflexales; f__Roseiflexaceae; g__Roseiflexus	< 0.001
p__Sumerlaeota; c__Sumerlaeia; o__Sumerlaeales; f__Sumerlaeaceae; g__Sumerlaea	< 0.001
p__Bacteroidota; c__Bacteroidia; o__Chitinophagales; f__Chitinophagaceae; g__Chitinophaga	< 0.001
p__Firmicutes; c__Bacilli; o__Bacillales; f__unclassified_o__Bacillales; g__unclassified_o__Bacillales	< 0.001
p__Actinobacteriota; c__Coriobacteriia; o__Coriobacteriales; f__Eggerthellaceae; g__Slackia	< 0.001
p__Proteobacteria; c__Alphaproteobacteria; o__Acetobacterales; f__Acetobacteraceae; g__norank_f__Acetobacteraceae	< 0.001
p__Firmicutes; c__Bacilli; o__Bacillales; f__Marinococcaceae; g__Natribacillus	< 0.001
p__Proteobacteria; c__Alphaproteobacteria; o__Rhizobiales; f__Rhizobiales_Incertae_Sedis; g__Bauldia	< 0.001
p__Firmicutes; c__Clostridia; o__Peptococcales; f__Peptococcaceae; g__unclassified_f__Peptococcaceae	< 0.001
p__Proteobacteria; c__Alphaproteobacteria; o__Caulobacterales; f__Caulobacteraceae; g__unclassified_f__Caulobacteraceae	< 0.001
p__Bacteroidota; c__Bacteroidia; o__Cytophagales; f__Microscillaceae; g__Chryseolinea	< 0.001
p__Gemmatimonadota; c__Gemmatimonadetes; o__Gemmatimonadales; f__Gemmatimonadaceae; g__Roseisolibacter	< 0.001
p__Acidobacteriota; c__Blastocatellia; o__Blastocatellales; f__Blastocatellaceae; g__Aridibacter	< 0.001
p__Proteobacteria; c__Alphaproteobacteria; o__Caulobacterales; f__Caulobacteraceae; g__Asticcacaulis	< 0.001
p__Actinobacteriota; c__Actinobacteria; o__Kineosporiales; f__Kineosporiaceae; g__Kineococcus	< 0.001
p__Proteobacteria; c__Alphaproteobacteria; o__Elsterales; f__norank_o__Elsterales; g__norank_f__norank_o__Elsterales	< 0.001
p__Chloroflexi; c__Chloroflexia; o__Chloroflexales; f__Herpetosiphonaceae; g__Herpetosiphon	< 0.001
p__Chloroflexi; c__OLB14; o__norank_c__OLB14; f__norank_o__norank_c__OLB14; g__norank_f__norank_o__norank_c__OLB14	< 0.001
p__Proteobacteria; c__Alphaproteobacteria; o__Acetobacterales; f__Acetobacteraceae; g__Rhodovarius	< 0.001
p__Actinobacteriota; c__Actinobacteria; o__Micromonosporales; f__Micromonosporaceae; g__Krasilnikovia	< 0.001
p__Bacteroidota; c__Bacteroidia; o__Cytophagales; f__Hymenobacteraceae; g__Pontibacter	< 0.001
p__Bacteroidota; c__Bacteroidia; o__Chitinophagales; f__Chitinophagaceae; g__norank_f__Chitinophagaceae	< 0.001
p__Actinobacteriota; c__Actinobacteria; o__Micromonosporales; f__Micromonosporaceae; g__norank_f__Micromonosporaceae	< 0.001
p__Firmicutes; c__Bacilli; o__Bacillales; f__Bacillaceae; g__Virgibacillus	< 0.001
p__Patescibacteria; c__Saccharimonadia; o__Saccharimonadales; f__Saccharimonadaceae; g__norank_f__Saccharimonadaceae	< 0.001
p__Proteobacteria; c__Gammaproteobacteria; o__Burkholderiales; f__Alcaligenaceae; g__unclassified_f__Alcaligenaceae	< 0.001
p__Bacteroidota; c__Bacteroidia; o__Cytophagales; f__Spirosomaceae; g__Spirosoma	< 0.001
p__Actinobacteriota; c__Actinobacteria; o__Propionibacteriales; f__Nocardoidaceae; g__Kribbella	< 0.001
p__Myxococcota; c__Myxococcia; o__Myxococcales; f__Myxococcaceae; g__Archangium	< 0.001
p__Cyanobacteria; c__Cyanobacteriia; o__Cyanobacteriales; f__Phormidiaceae; g__Phormidium_IAM_M-71	< 0.001
p__Proteobacteria; c__Gammaproteobacteria; o__R7C24; f__norank_o__R7C24; g__norank_f__norank_o__R7C24	< 0.001
p__Myxococcota; c__Polyangia; o__Nannocystales; f__Nannocystaceae; g__Nannocystis	< 0.001
p__Bacteroidota; c__Bacteroidia; o__Bacteroidales; f__Tannerellaceae; g__Parabacteroides	< 0.001
p__Proteobacteria; c__Alphaproteobacteria; o__Thalassobaculales; f__norank_o__Thalassobaculales;	< 0.001

g_norank_f_norank_o_Thalassobaculales	
p_Firmicutes; c_Negativicutes; o_Veillonellales-Selenomonadales; f_Veillonellaceae; g_norank_f_Veillonellaceae	< 0.001
p_Chloroflexi; c_Chloroflexia; o_Chloroflexales; f_Roseiflexaceae; g_unclassified_f_Roseiflexaceae	< 0.001
p_Bacteroidota; c_Bacteroidia; o_Sphingobacteriales; f_env.OPS_17; g_norank_f_env.OPS_17	< 0.001
p_Bacteroidota; c_Bacteroidia; o_Bacteroidales; f_Prevotellaceae; g_Prevotellaceae_UCG-004	< 0.001
p_Bdellovibrionota; c_Bdellovibrionia; o_Bdellovibrionales; f_Bdellovibrionaceae; g_Bdellovibrionia	< 0.001
p_Bacteroidota; c_Bacteroidia; o_Chitinophagales; f_Chitinophagaceae; g_Edaphobaculum	< 0.001
p_Proteobacteria; c_Gammaproteobacteria; o_Xanthomonadales; f_Rhodanobacteraceae; g_norank_f_Rhodanobacteraceae	< 0.001
p_Bacteroidota; c_Bacteroidia; o_Cytophagales; f_Cyclobacteriaceae; g_Mariniradius	< 0.001
p_Proteobacteria; c_Gammaproteobacteria; o_Burkholderiales; f_Nitrosomonadaceae; g_Ellin6067	< 0.001
p_Verrucomicrobiota; c_Verrucomicrobiae; o_Chthoniobacteriales; f_Chthoniobacteraceae; g_Chthoniobacter	< 0.001
p_Bacteroidota; c_Bacteroidia; o_Bacteroidales; f_Marinifilaceae; g_Odoribacter	< 0.001
p_Proteobacteria; c_Alphaproteobacteria; o_Rhodobacterales; f_Rhodobacteraceae; g_Ketogulonicigenium	< 0.001
p_Fibrobacterota; c_Fibrobacteria; o_Fibrobacteriales; f_Fibrobacteraceae; g_Fibrobacter	< 0.001
p_Bacteroidota; c_Bacteroidia; o_Chitinophagales; f_Chitinophagaceae; g_Terrimonas	< 0.001
p_Proteobacteria; c_Gammaproteobacteria; o_Enterobacteriales; f_unclassified_o_Enterobacteriales; g_unclassified_o_Enterobacteriales	< 0.001
p_Actinobacteriota; c_Actinobacteria; o_Streptosporangiales; f_Streptosporangiaceae; g_Nonomurea	< 0.001
p_Gemmatimonadota; c_Longimicrobia; o_Longimicrobiales; f_Longimicrobiaceae; g_YC-ZSS-LKJ147	< 0.001
p_Firmicutes; c_Bacilli; o_Bacillales; f_Bacillaceae; g_Halobacillus	< 0.001
p_Proteobacteria; c_Gammaproteobacteria; o_Cardiobacteriales; f_Wohlfahrtiimonadaceae; g_Ignatzschineria	< 0.001
p_Actinobacteriota; c_Actinobacteria; o_Actinomycetales; f_Actinomycetaceae; g_Trueperella	< 0.001
p_Proteobacteria; c_Gammaproteobacteria; o_Burkholderiales; f_Alcaligenaceae; g_Parapusillimonas	< 0.001
p_Myxococcota; c_Myxococcia; o_Myxococcales; f_Myxococcaceae; g_norank_f_Myxococcaceae	< 0.001
p_Verrucomicrobiota; c_Verrucomicrobiae; o_Verrucomicrobiales; f_Rubritaleaceae; g_Luteolibacter	< 0.001
p_Proteobacteria; c_Alphaproteobacteria; o_Azospirillales; f_Azospirillaceae; g_Skermanella	< 0.001
p_Acidobacteriota; c_Blastocatellia; o_DS-100; f_norank_o_DS-100; g_norank_f_norank_o_DS-100	< 0.001
p_Bacteroidota; c_Bacteroidia; o_Chitinophagales; f_Saprospiraceae; g_OLB8	< 0.001
p_Proteobacteria; c_Gammaproteobacteria; o_Burkholderiales; f_SC-I-84; g_norank_f_SC-I-84	< 0.001
p_Cyanobacteria; c_Cyanobacteria; o_Cyanobacteriales; f_Chroococciopsaceae; g_Gloeocapsa_PCC-7428	< 0.001
p_Firmicutes; c_Bacilli; o_Lactobacillales; f_Lactobacillaceae; g_Pediococcus	< 0.001
p_Firmicutes; c_Clostridia; o_Peptostreptococcales-Tissierellales; f_norank_o_Peptostreptococcales-Tissierellales; g_W5053	< 0.001
p_Actinobacteriota; c_Actinobacteria; o_Propionibacteriales; f_Propionibacteriaceae; g_Propioniciqlava	< 0.001
p_Proteobacteria; c_Gammaproteobacteria; o_Xanthomonadales; f_Xanthomonadaceae; g_Arenimonas	< 0.001
p_Gemmatimonadota; c_Gemmatimonadetes; o_Gemmatimonadales; f_Gemmatimonadaceae; g_unclassified_f_Gemmatimonadaceae	< 0.001
p_Proteobacteria; c_Alphaproteobacteria; o_Rhodobacterales; f_Rhodobacteraceae; g_norank_f_Rhodobacteraceae	< 0.001
p_Actinobacteriota; c_unclassified_p_Actinobacteriota; o_unclassified_p_Actinobacteriota; f_unclassified_p_Actinobacteriota; g_unclassified_p_Actinobacteriota	< 0.001
p_Acidobacteriota; c_Blastocatellia; o_Blastocatellales; f_Blastocatellaceae; g_unclassified_f_Blastocatellaceae	< 0.001
p_Myxococcota; c_Polyangia; o_Polyangiales; f_Polyangiaceae; g_norank_f_Polyangiaceae	< 0.001
p_Firmicutes; c_Bacilli; o_Erysipelotrichales; f_Erysipelotrichaceae; g_ZOR0006	< 0.001
p_Proteobacteria; c_Gammaproteobacteria; o_Xanthomonadales; f_Rhodanobacteraceae;	< 0.001

g_unclassified_f_Rhodanobacteraceae	
p_Acidobacteriota; c_Thermoanaerobaculia; o_Thermoanaerobaculales; f_Thermoanaerobaculaceae; g_Subgroup_10	< 0.001
p_Firmicutes; c_Bacilli; o_Lactobacillales; f_Leuconostocaceae; g_Leuconostoc	< 0.001
p_Desulfobacterota; c_norank_p_Desulfobacterota; o_norank_c_norank_p_Desulfobacterota; f_norank_o_norank_c_norank_p_Desulfobacterota; g_norank_f_norank_o_norank_c_norank_p_Desulfobacterota	< 0.001
p_Chloroflexi; c_Anaerolineae; o_SBR1031; f_norank_o_SBR1031; g_norank_f_norank_o_SBR1031	< 0.001
p_Firmicutes; c_Bacilli; o_Erysipelotrichales; f_unclassified_o_Erysipelotrichales; g_unclassified_o_Erysipelotrichales	< 0.001
p_Actinobacteriota; c_Actinobacteria; o_Micromonosporales; f_Micromonosporaceae; g_Rhizocola	< 0.001
p_Chloroflexi; c_Anaerolineae; o_Ardenticatenales; f_Ardenticatenaceae; g_norank_f_Ardenticatenaceae	< 0.001
p_Actinobacteriota; c_Acidimicrobiia; o_Microtrichales; f_Ilumatobacteraceae; g_CL500-29_marine_group	< 0.001
p_Proteobacteria; c_Gammaproteobacteria; o_Burkholderiales; f_Sutterellaceae; g_norank_f_Sutterellaceae	< 0.001
p_Proteobacteria; c_Alphaproteobacteria; o_Micropepsales; f_Micropepsaceae; g_norank_f_Micropepsaceae	< 0.001
p_NB1-j; c_norank_p_NB1-j; o_norank_c_norank_p_NB1-j; f_norank_o_norank_c_norank_p_NB1-j; g_norank_f_norank_o_norank_c_norank_p_NB1-j	< 0.001
p_Actinobacteriota; c_Actinobacteria; o_Nitriliruptorales; f_Nitriliruptoraceae; g_norank_f_Nitriliruptoraceae	< 0.001
p_Proteobacteria; c_Alphaproteobacteria; o_Rhizobiales; f_Beijerinckiaceae; g_norank_f_Beijerinckiaceae	< 0.001
p_Cyanobacteria; c_Cyanobacteriia; o_Cyanobacteriales; f_norank_o_Cyanobacteriales; g_norank_f_norank_o_Cyanobacteriales	< 0.001
p_Firmicutes; c_Bacilli; o_Bacillales; f_Marinococcaceae; g_Geomicrobium	< 0.001
p_Bacteroidota; c_Bacteroidia; o_Cytophagales; f_Spirosomaceae; g_Dyadobacter	< 0.001
p_Firmicutes; c_Bacilli; o_Erysipelotrichales; f_Erysipelotrichaceae; g_Turicibacter	< 0.001
p_Actinobacteriota; c_Thermoleophilia; o_Solirubrobacterales; f_Solirubrobacteraceae; g_Parviterribacter	< 0.001
p_Firmicutes; c_Clostridia; o_norank_c_Clostridia; f_Hungateiclostridiaceae; g_Ruminiclostridium	< 0.001
p_Proteobacteria; c_Gammaproteobacteria; o_Burkholderiales; f_Oxalobacteraceae; g_norank_f_Oxalobacteraceae	< 0.001
p_Proteobacteria; c_Alphaproteobacteria; o_Sphingomonadales; f_Sphingomonadaceae; g_Sphingoaurantiacus	< 0.001
p_Proteobacteria; c_Alphaproteobacteria; o_Azospirillales; f_Azospirillaceae; g_norank_f_Azospirillaceae	< 0.001
p_Proteobacteria; c_Alphaproteobacteria; o_Caulobacteriales; f_Caulobacteraceae; g_PMMR1	< 0.001
p_Firmicutes; c_Bacilli; o_Paenibacillales; f_Paenibacillaceae; g_Ammoniphilus	< 0.001
p_Verrucomicrobiota; c_Verrucomicrobiae; o_Verrucomicrobiales; f_Verrucomicrobiaceae; g_Roseimicrobium	< 0.001
p_Patescibacteria; c_Saccharimonadia; o_Saccharimonadales; f_LWQ8; g_norank_f_LWQ8	< 0.001
p_Actinobacteriota; c_Acidimicrobiia; o_Microtrichales; f_Microtrichaceae; g_Sva0996_marine_group	< 0.001
p_Actinobacteriota; c_Thermoleophilia; o_Solirubrobacterales; f_Solirubrobacteraceae; g_Conexibacter	< 0.001
p_Actinobacteriota; c_Coriobacteriia; o_Coriobacteriales; f_Atopobiaceae; g_Libanicoccus	< 0.001
p_Verrucomicrobiota; c_Verrucomicrobiae; o_Opitutales; f_Puniceicoccaceae; g_norank_f_Puniceicoccaceae	< 0.001
p_Bacteroidota; c_Bacteroidia; o_Cytophagales; f_Microscillaceae; g_Ohtaekwangia	< 0.001
p_Acidobacteriota; c_Acidobacteriae; o_Solibacterales; f_Solibacteraceae; g_Candidatus_Solibacter	< 0.001
p_Myxococcota; c_Polyangia; o_Polyangiales; f_Polyangiaceae; g_Pajaroellobacter	< 0.001
p_Firmicutes; c_Bacilli; o_Bacillales; f_Planococcaceae; g_Sporosarcina	< 0.001
p_Actinobacteriota; c_Coriobacteriia; o_Coriobacteriales; f_Coriobacteriaceae; g_Collinsella	< 0.001
p_Proteobacteria; c_Gammaproteobacteria; o_Xanthomonadales; f_Xanthomonadaceae; g_Stenotrophomonas	< 0.001
p_Firmicutes; c_Clostridia; o_Clostridiales; f_Clostridiaceae; g_Clostridium_sensu_stricto_13	< 0.001
p_Verrucomicrobiota; c_Verrucomicrobiae; o_Chthoniobacteriales; f_Xiphinematobacteraceae; g_Candidatus_Xiphinematobacter	< 0.001

p__Acidobacteriota; c__Subgroup_18; o__norank_c__Subgroup_18; f__norank_o__norank_c__Subgroup_18; g__norank_f__norank_o__norank_c__Subgroup_18	< 0.001
p__Bdellovibrionota; c__Bdellovibrionia; o__Bdellovibrionales; f__Bdellovibrionaceae; g__OM27_clade	< 0.001
p__Bdellovibrionota; c__Oligoflexia; o__Oligoflexiales; f__norank_o__Oligoflexiales; g__Oligoflexus	< 0.001
p__Proteobacteria; c__Alphaproteobacteria; o__unclassified_c__Alphaproteobacteria; f__unclassified_c__Alphaproteobacteria; g__unclassified_c__Alphaproteobacteria	< 0.001
p__Planctomycetota; c__Planctomycetes; o__Isosphaerales; f__Isosphaeraceae; g__unclassified_f__Isosphaeraceae	< 0.001
p__Proteobacteria; c__Gammaproteobacteria; o__Aeromonadales; f__Succinivibrionaceae; g__Anaerobiospirillum	< 0.001
p__Myxococcota; c__Polyangia; o__Polyangiiales; f__Polyangiaceae; g__Sorangium	< 0.001
p__Bacteroidota; c__Bacteroidia; o__Chitinophagales; f__Chitinophagaceae; g__Niabella	< 0.001
p__Proteobacteria; c__Alphaproteobacteria; o__Rhodospirillales; f__Rhodospirillaceae; g__norank_f__Rhodospirillaceae	< 0.001
p__Proteobacteria; c__Alphaproteobacteria; o__Rhodobacterales; f__Rhodobacteraceae; g__Roseobacter_clade_CHAB-I-5_lineage	< 0.001
p__Bacteroidota; c__Bacteroidia; o__Chitinophagales; f__Chitinophagaceae; g__Filimonas	< 0.001
p__Acidobacteriota; c__Vicinamibacteria; o__Subgroup_17; f__norank_o__Subgroup_17; g__norank_f__norank_o__Subgroup_17	< 0.001
p__Proteobacteria; c__Alphaproteobacteria; o__Rhizobiales; f__Xanthobacteraceae; g__Starkeya	< 0.001
p__Proteobacteria; c__Gammaproteobacteria; o__Burkholderiales; f__norank_o__Burkholderiales; g__norank_f__norank_o__Burkholderiales	< 0.001
p__Actinobacteriota; c__Actinobacteria; o__Corynebacteriales; f__Nocardiaceae; g__Nocardia	< 0.001
p__Proteobacteria; c__Alphaproteobacteria; o__Rhizobiales; f__unclassified_o__Rhizobiales; g__unclassified_o__Rhizobiales	< 0.001
p__Firmicutes; c__Bacilli; o__Staphylococcales; f__Staphylococcaceae; g__Jeotgalicoccus	< 0.001
p__Bacteroidota; c__Bacteroidia; o__Flavobacteriales; f__NS9_marine_group; g__norank_f__NS9_marine_group	< 0.001
p__Cyanobacteria; c__Cyanobacteriia; o__Cyanobacteriales; f__Microcystaceae; g__Chalicogloea_CCALA_975	< 0.001
p__Proteobacteria; c__Gammaproteobacteria; o__Oceanospirillales; f__Halomonadaceae; g__Carmimonas	< 0.001
p__Chloroflexi; c__Anaerolineae; o__Caldilineales; f__Caldilineaceae; g__Litorilinea	< 0.001
p__Myxococcota; c__Polyangia; o__Blfdi19; f__norank_o__Blfdi19; g__norank_f__norank_o__Blfdi19	< 0.001
p__Actinobacteriota; c__Thermoleophilia; o__Solirubrobacterales; f__Solirubrobacteraceae; g__Patulibacter	< 0.001
p__Actinobacteriota; c__Actinobacteria; o__Streptosporangiales; f__Streptosporangiaceae; g__unclassified_f__Streptosporangiaceae	< 0.001
p__Cyanobacteria; c__Cyanobacteriia; o__Leptolyngbyales; f__Leptolyngbyaceae; g__Leptolyngbya_PCC-6306	< 0.001
p__Bacteroidota; c__Bacteroidia; o__Chitinophagales; f__Chitinophagaceae; g__Aurantisolimonas	< 0.001
p__Actinobacteriota; c__Coriobacteriia; o__Coriobacteriales; f__Eggerthellaceae; g__Senegalimassilia	< 0.001
p__Firmicutes; c__Bacilli; o__Erysipelotrichales; f__Erysipelatoclostridiaceae; g__Asteroleplasma	< 0.001
p__Bacteroidota; c__Bacteroidia; o__Cytophagales; f__Microscillaceae; g__Flexibacter	< 0.001
p__Proteobacteria; c__Alphaproteobacteria; o__Kiloniellales; f__Fodinicurvataceae; g__norank_f__Fodinicurvataceae	< 0.001
p__Proteobacteria; c__Alphaproteobacteria; o__Acetobacterales; f__Acetobacteraceae; g__Acidicaldus	< 0.001
p__Proteobacteria; c__Alphaproteobacteria; o__Rhizobiales; f__Beijerinckiaceae; g__Camelimonas	< 0.001
p__Bacteroidota; c__Bacteroidia; o__Sphingobacteriales; f__AKYH767; g__norank_f__AKYH767	< 0.001
p__Actinobacteriota; c__Actinobacteria; o__Propionibacteriales; f__unclassified_o__Propionibacteriales; g__unclassified_o__Propionibacteriales	< 0.001
p__Bacteroidota; c__Bacteroidia; o__Flavobacteriales; f__Weeksellaceae; g__Moheibacter	< 0.001
p__Bacteroidota; c__Rhodothermia; o__Rhodothermales; f__Rhodothermaceae; g__unclassified_f__Rhodothermaceae	< 0.001
p__Gemmatimonadota; c__Longimicrobia; o__Longimicrobiales; f__Longimicrobiaceae; g__Longimicrobium	< 0.001
p__Proteobacteria; c__Alphaproteobacteria; o__Rhizobiales; f__KF-JG30-B3; g__norank_f__KF-JG30-B3	< 0.001
p__Proteobacteria; c__Alphaproteobacteria; o__Sphingomonadales; f__Sphingomonadaceae; g__Sphingobium	< 0.001

p__Bacteroidota; c__Bacteroidia; o__Chitinophagales; f__Chitinophagaceae; g__Niastella	< 0.001
p__Proteobacteria; c__Alphaproteobacteria; o__Rhizobiales; f__Beijerinckiaceae; g__1174-901-12	< 0.001
p__Bacteroidota; c__Bacteroidia; o__Flavobacteriales; f__Blattabacteriaceae; g__unclassified_f__Blattabacteriaceae	< 0.001
p__Planctomycetota; c__Planctomycetes; o__Isosphaerales; f__Isosphaeraceae; g__norank_f__Isosphaeraceae	< 0.001
p__Proteobacteria; c__Gammaproteobacteria; o__Cellvibrionales; f__Spongiibacteraceae; g__BD1-7_clade	< 0.001
p__Planctomycetota; c__Phycisphaerae; o__Phycisphaerales; f__Phycisphaeraceae; g__SM1A02	< 0.001
p__Actinobacteriota; c__Actinobacteria; o__Bifidobacteriales; f__Bifidobacteriaceae; g__Gardnerella	< 0.001
p__Proteobacteria; c__Gammaproteobacteria; o__EPR3968-O8a-Bc78; f__norank_o__EPR3968-O8a-Bc78; g__norank_f__norank_o__EPR3968-O8a-Bc78	< 0.001
p__Proteobacteria; c__Alphaproteobacteria; o__Rickettsiales; f__SM2D12; g__norank_f__SM2D12	< 0.001
p__Actinobacteriota; c__Actinobacteria; o__Nitriliruptorales; f__Nitriliruptoraceae; g__unclassified_f__Nitriliruptoraceae	< 0.001
p__Proteobacteria; c__Gammaproteobacteria; o__unclassified_c__Gammaproteobacteria; f__unclassified_c__Gammaproteobacteria; g__unclassified_c__Gammaproteobacteria	< 0.001
p__Planctomycetota; c__Planctomycetes; o__Isosphaerales; f__Isosphaeraceae; g__Paludisphaera	< 0.001
p__Fusobacteriota; c__Fusobacteriia; o__Fusobacteriales; f__Fusobacteriaceae; g__Cetobacterium	< 0.001
p__Bacteroidota; c__Bacteroidia; o__Sphingobacteriales; f__Sphingobacteriaceae; g__Arcticibacter	< 0.001
p__Proteobacteria; c__Alphaproteobacteria; o__Rhizobiales; f__Labraceae; g__Labrys	< 0.001
p__Bacteroidota; c__Bacteroidia; o__Cytophagales; f__Hymenobacteraceae; g__Hymenobacter	< 0.001
p__Proteobacteria; c__Alphaproteobacteria; o__Alphaproteobacteria_Incertae_Sedis; f__unclassified_o__Alphaproteobacteria_Incertae_Sedis; g__unclassified_o__Alphaproteobacteria_Incertae_Sedis	< 0.001
p__Proteobacteria; c__Gammaproteobacteria; o__CCM19a; f__norank_o__CCM19a; g__norank_f__norank_o__CCM19a	< 0.001
p__Proteobacteria; c__Gammaproteobacteria; o__Burkholderiales; f__Nitrosomonadaceae; g__MND1	< 0.001
p__Chloroflexi; c__Chloroflexia; o__Chloroflexales; f__unclassified_o__Chloroflexales; g__unclassified_o__Chloroflexales	< 0.001
p__Actinobacteriota; c__Actinobacteria; o__Bifidobacteriales; f__Bifidobacteriaceae; g__Bifidobacterium	< 0.001
p__Bacteroidota; c__Bacteroidia; o__Chitinophagales; f__Chitinophagaceae; g__Ferruginibacter	< 0.001
p__Proteobacteria; c__Gammaproteobacteria; o__Xanthomonadales; f__Rhodanobacteraceae; g__Ahniella	< 0.001
p__Chloroflexi; c__Anaerolineae; o__Anaerolineales; f__Anaerolineaceae; g__unclassified_f__Anaerolineaceae	< 0.001
p__Firmicutes; c__Bacilli; o__Thermoactinomycetales; f__Thermoactinomycetaceae; g__Geothermomicrobium	< 0.001
p__Firmicutes; c__Clostridia; o__Peptostreptococcales-Tissierellales; f__unclassified_o__Peptostreptococcales-Tissierellales; g__unclassified_o__Peptostreptococcales-Tissierellales	< 0.001
p__Gemmatimonadota; c__BD2-11_terrestrial_group; o__norank_c__BD2-11_terrestrial_group; f__norank_o__norank_c__BD2-11_terrestrial_group; g__norank_f__norank_o__norank_c__BD2-11_terrestrial_group	< 0.001
p__Chloroflexi; c__Anaerolineae; o__SBR1031; f__unclassified_o__SBR1031; g__unclassified_o__SBR1031	< 0.001
p__Actinobacteriota; c__Actinobacteria; o__Nitriliruptorales; f__Nitriliruptoraceae; g__Egicoccus	< 0.001
p__Myxococcota; c__Myxococcia; o__Myxococcales; f__Myxococcaceae; g__Cystobacter	< 0.001
p__Proteobacteria; c__Alphaproteobacteria; o__Azospirillales; f__Azospirillaceae; g__unclassified_f__Azospirillaceae	< 0.001
p__Bdellovibrionota; c__Bdellovibrionia; o__Bacteriovorales; f__Bacteriovoraceae; g__Peredibacter	< 0.001
p__Cyanobacteria; c__Cyanobacteriia; o__Cyanobacteriales; f__Chroococciopsaceae; g__Aliterella	< 0.001
p__Planctomycetota; c__Phycisphaerae; o__Phycisphaerales; f__Phycisphaeraceae; g__norank_f__Phycisphaeraceae	< 0.001
p__Firmicutes; c__Limnochordia; o__norank_c__Limnochordia; f__norank_o__norank_c__Limnochordia; g__Hydrogenispora	< 0.001
p__Proteobacteria; c__Gammaproteobacteria; o__Burkholderiales; f__Rhodocyclaceae; g__Propionivibrio	< 0.001
p__Proteobacteria; c__Alphaproteobacteria; o__Micavibrionales; f__norank_o__Micavibrionales; g__norank_f__norank_o__Micavibrionales	< 0.001

p__Bacteroidota; c__Rhodothermia; o__Balneolales; f__Balneolaceae; g__unclassified_f__Balneolaceae	< 0.001
p__Armatimonadota; c__Armatimonadia; o__Armatimonadales; f__norank_o__Armatimonadales; g__norank_f__norank_o__Armatimonadales	< 0.001
p__Firmicutes; c__Bacilli; o__Acholeplasmatales; f__Acholeplasmataceae; g__Candidatus_Phytoplasma	< 0.001
p__Acidobacteriota; c__Acidobacteriae; o__norank_c__Acidobacteriae; f__norank_o__norank_c__Acidobacteriae; g__Paludibaculum	< 0.001
p__Proteobacteria; c__Alphaproteobacteria; o__Defluviicoceales; f__norank_o__Defluviicoceales; g__norank_f__norank_o__Defluviicoceales	< 0.001
p__Proteobacteria; c__Alphaproteobacteria; o__Sphingomonadales; f__Sphingomonadaceae; g__Sphingopyxis	< 0.001
p__Cyanobacteria; c__Cyanobacteriia; o__Cyanobacteriales; f__Nostocaceae; g__Scytonema_VB-61278	< 0.001
p__Proteobacteria; c__Gammaproteobacteria; o__Burkholderiales; f__Comamonadaceae; g__Ramlibacter	< 0.001
p__Myxococcota; c__Polyangia; o__Polyangiales; f__Polyangiaceae; g__Chondromyces	< 0.001
p__WS2; c__norank_p__WS2; o__norank_c__norank_p__WS2; f__norank_o__norank_c__norank_p__WS2; g__norank_f__norank_o__norank_c__norank_p__WS2	< 0.001
p__Proteobacteria; c__Alphaproteobacteria; o__Rickettsiales; f__Rickettsiaceae; g__unclassified_f__Rickettsiaceae	< 0.001
p__Proteobacteria; c__Gammaproteobacteria; o__Burkholderiales; f__Rhodocyclaceae; g__unclassified_f__Rhodocyclaceae	< 0.001
p__Firmicutes; c__Clostridia; o__Oscillospirales; f__Butyricoccaceae; g__UCG-008	< 0.001
p__Bacteroidota; c__Bacteroidia; o__Chitinophagales; f__Chitinophagaceae; g__Pseudoflavitalea	< 0.001
p__Proteobacteria; c__Gammaproteobacteria; o__Cardiobacteriales; f__Wohlfahrtiimonadaceae; g__unclassified_f__Wohlfahrtiimonadaceae	< 0.001
p__Planctomycetota; c__Phycisphaerae; o__Tepidisphaerales; f__norank_o__Tepidisphaerales; g__norank_f__norank_o__Tepidisphaerales	< 0.001
p__Bacteroidota; c__Bacteroidia; o__Bacteroidales; f__Dysgonomonadaceae; g__Dysgonomonas	< 0.001
p__Bacteroidota; c__Bacteroidia; o__Sphingobacteriales; f__Sphingobacteriaceae; g__Parapedobacter	< 0.001
p__Acidobacteriota; c__Acidobacteriae; o__unclassified_c__Acidobacteriae; f__unclassified_c__Acidobacteriae; g__unclassified_c__Acidobacteriae	< 0.001
p__Cyanobacteria; c__Cyanobacteriia; o__Cyanobacteriales; f__Chroococcidiopsaceae; g__Chroococcidiopsis_PCC_7203	< 0.001
p__Actinobacteriota; c__Coriobacteriia; o__Coriobacteriales; f__Coriobacteriales_Incertae_Sedis; g__Phoenicibacter	< 0.001
p__Proteobacteria; c__Alphaproteobacteria; o__Rhizobiales; f__Beijerinckiaceae; g__unclassified_f__Beijerinckiaceae	< 0.001
p__Armatimonadota; c__norank_p__Armatimonadota; o__norank_c__norank_p__Armatimonadota; f__norank_o__norank_c__norank_p__Armatimonadota; g__norank_f__norank_o__norank_c__norank_p__Armatimonadota	< 0.001
p__Verrucomicrobiota; c__Verrucomicrobiae; o__Chthoniobacteriales; f__Chthoniobacteraceae; g__Candidatus_Udaeobacter	< 0.001
p__Proteobacteria; c__Gammaproteobacteria; o__Burkholderiales; f__Nitrosomonadaceae; g__Nitrosomonas	< 0.001
p__Planctomycetota; c__Planctomycetes; o__norank_c__Planctomycetes; f__norank_o__norank_c__Planctomycetes; g__norank_f__norank_o__norank_c__Planctomycetes	< 0.001
p__Firmicutes; c__Bacilli; o__Staphylococcales; f__Gemellaceae; g__Gemella	< 0.001
p__Actinobacteriota; c__Coriobacteriia; o__Coriobacteriales; f__Atopobiaceae; g__Coriobacteriaceae_UCG-003	< 0.001
p__WPS-2; c__norank_p__WPS-2; o__norank_c__norank_p__WPS-2; f__norank_o__norank_c__norank_p__WPS-2; g__norank_f__norank_o__norank_c__norank_p__WPS-2	< 0.001
p__Actinobacteriota; c__Actinobacteria; o__Micrococcales; f__Demequinaceae; g__Demequina	< 0.001
p__Actinobacteriota; c__Thermoleophilia; o__Solirubrobacteriales; f__Solirubrobacteraceae; g__norank_f__Solirubrobacteraceae	< 0.001
p__Actinobacteriota; c__Actinobacteria; o__Bifidobacteriales; f__Bifidobacteriaceae; g__unclassified_f__Bifidobacteriaceae	< 0.001
p__Patescibacteria; c__Saccharimonadia; o__Saccharimonadales; f__S32; g__TM7	< 0.001

p__Bacteroidota; c__Bacteroidia; o__Cytophagales; f__Spirosomaceae; g__unclassified_f__Spirosomaceae	< 0.001
p__Cyanobacteria; c__Cyanobacteriia; o__Cyanobacteriales; f__Nostocaceae; g__Calothrix_PCC-6303	< 0.001
p__Bacteroidota; c__Bacteroidia; o__Sphingobacteriales; f__Sphingobacteriaceae; g__Pseudopedobacter	< 0.001
p__Gemmatimonadota; c__Longimicrobia; o__Longimicrobiales; f__Longimicrobiaceae; g__unclassified_f__Longimicrobiaceae	< 0.001
p__Planctomycetota; c__Phycisphaerae; o__Phycisphaerales; f__Phycisphaeraceae; g__unclassified_f__Phycisphaeraceae	< 0.001
p__Proteobacteria; c__Gammaproteobacteria; o__Steroidobacteriales; f__Steroidobacteraceae; g__norank_f__Steroidobacteraceae	< 0.001
p__Actinobacteriota; c__Actinobacteria; o__Micrococcales; f__Micrococcaceae; g__Rothia	< 0.001
p__Chloroflexi; c__Anaerolineae; o__Ardenticatenales; f__norank_o__Ardenticatenales; g__norank_f__norank_o__Ardenticatenales	< 0.001
p__Patescibacteria; c__unclassified_p__Patescibacteria; o__unclassified_p__Patescibacteria; f__unclassified_p__Patescibacteria; g__unclassified_p__Patescibacteria	< 0.001

2. The description of the contents and statements in many parts of this article is vague: For example, some contents of the method steps are not clearly written, the sentence "We studied two cohorts of wild Francois' langurs with stable numbers over the periods" led me to believe that this study studied two cohorts of wild Francois' langurs with stable numbers over the periods. In addition, why does the determination of fecal hormones in the results not use the samples in Table s1, but use the fecal samples in Table s2 collected at different times? This makes me very confused when reading this manuscript!

We thank you for pointing out this problem. The sentence " we studied two cohorts of wild Francois' languages with stable numbers over the periods " is poorly formulated and we revised, please see L108-112.

It should be mentioned that we supplemented the second period experiment by discovering in the first stage that there is no significant seasonal variation in the relative abundance of most bacterial taxa of the langurs, trying to probe the seasonal regulation of energy metabolism of the langurs from physiological hormone studies, and further understand the adaptation by the langurs to limestone forest. This may have reduced the support for the results of gut microbiota during the first stage, and we acknowledge that the combined data on gut microbiota, physiological hormones, and activity time in the same period are more convincing. However, our study is the first report to combine these data to provide a comprehensive analysis of the adaptation by Francois' langurs to limestone habitat. The current study should properly reflect the characteristics and seasonal variations in the gut microbiota of the langurs living in the limestone forest.

3. Many of the words used in the full text are inaccurate: for example, the title in Table s16 uses " several primates", but this table clearly listed 11 animals, so it can be written directly as "11 primates". In addition, the remarks under this table are not properly written as

"1-Firmicutes"; the "invsimpson" in Table s17 does not need to be italicized; The notional words in the title of Figure 2 need not be capitalized, and the four figures in the figure 2 need to be written with a, b, c and d respectively; What do a and b in Figure 3 represent? Figures 2 and 5 also need to refer to a, b, c and d.

Thanks for your valuable comments and help, we modified. Please see Table S16, Fig.2, Fig.3 and Fig.5. In addition, based on the comments of other reviewers, we deleted the original Table S16 and changed the original Table S17 to Table S16.

4. Some writing of the full text are inconsistent: for example, the letter P should be capitalized; "triiodothyronine (T3) and tetraiodothyronine (T4)" are unified if the initial letters are all capitalized.

Thank you for your tips, we have modified it. Please see L251-252.

Reviewer 2:

The manuscript described microbiome characterization by seasonal variation in the wild Francois's langurs living in limestone forest. This paper revealed that microbiome represented relatively stable during the seasons. Only the lower bacteria taxa at the family level showed variable depending on seasons.

Major revision

1. Significant different microbiota composition at phylum and family level was identified in wild Francois langurs reared in the same place between this paper and a previous paper published in this research team (<https://doi.org/10.21203/rs.3.rs-2377898/v1>) (Bacterioidetes 10.29% {plus minus} 8.50% vs 4.82% {plus minus} 1.41%; Actinobacteria 4.05% {plus minus} 5.35% vs 9.11% {plus minus} 8.20%; Ocsillospiraceae 17.74%{plus minus}6.86, 30.21% {plus minus} 4.87% etc.). The reason for this difference should be explained.

Thanks for your interest in this problem. Our study is different from the previous paper. Current study aims to examine the seasonal shift pattern of the gut microbiota of François's langurs, whereas the previous one attempts to detect the effect of captivity on the gut microbiota of the wild François's langurs. Moreover, current works cover a completed year circle and provide data based on 152 fecal samples from the wild langurs. The previous paper is based on 32 fecal samples (15 from the provisioned langurs and 17 from the wild individuals), which provides limited information without any data on seasonality. In addition, we combined behavioral and fecal hormone data into current study. Actually, the fecal samples used in the two studies were different because of the different sampling times and number of samples.

2. Direct proof should be added regarding increased level of Christensenellaceae is associated with dealing with foods such as young leaves and fruits.

Thanks for your suggestion. We had data on feeding activities of the langurs and could not directly demonstrate the association of young leaves and fruits with Christensenellaceae. based only on previous studies (Waters and Ley, 2019, Baniel et al., 2021). Moreover, in combination with the comments of other reviewers, we cannot over interpret the results, and we rewrite this part. Please see L288, L309.

References mentioned above:

Waters JL, Ley RE. 2019. The human gut bacteria *Christensenellaceae* are widespread, heritable, and associated with health. *BMC Biol.* 17: 83. doi: <http://doi.org/10.1186/s12915-019-0699-4>.

Baniel A, Amato KR, Beehner JC, Bergman TJ, Mercer A, Perlman RF, Petruccio L, Reitsema L, Sams S, Lu A, Snyder-Mackler N. 2021. Seasonal shifts in the gut microbiome indicate plastic responses to diet in wild geladas. *Microbiome* 9: 26. doi: <https://doi.org/10.1186/s40168-020-00977-9>

3. The energy metabolism have been known for association with various factors including other hormones.

Thank you for your information. Actually, after the first period found that there was no significant seasonal variation in the relative abundance of most bacterial taxa in the gut microbiota of the François's langurs, we examined the seasonal difference of thyroid hormones in their feces during the second period to explore their energy status, which may help us better understand the adaptations of the langurs.

Reviewer 3:

The manuscript reflects a broad and robust dataset that reports on the fecal (not the gut) microbiota of wild langurs, and also reports on the collection of some other animal data and the correlation between these values and microbiota abundance. The authors use numerous statistical tests and they are generally appropriately applied; however there are superior tests that the authors could and should use in their analyses. Finally, I am very concerned that the paper is not very easy to read, and the authors could do some work to improve the structure of the results and discussion.

Tests: The authors use ANOVA and adonis to discriminate differences in overall microbiota composition between treatments, and also use Wilcoxon tests to discriminate changes in

abundance of individual microbes. I recommend the authors replace these analyses with PERMANOVA (adonis is PERMANOVA, but when run through some applications such as the qiime2 software, it may not be possible to use multiple covariates. If the authors move to R, they should be able to perform a PERMANOVA using the vegan package (function is named adonis). Also, a method such as ANCOM or LefSe could be used to identify individual microbes that vary in abundance between treatments.

Thanks for your suggestion. We used the “vegan” package in R (4.2.2) for PERMANOVA (adonis function) to evaluate the differences in gut microbiota communities between seasonal groupings, please see Fig S2.

In addition, LefSe can display bacterial taxa with inter-group differences, but some studies also use Wilcoxon tests (Holmes et al., 2022, Huang et al., 2022, Rocha et al., 2022), which is also a method that effectively tests for difference in the relative abundance of bacterial taxa, and FDR is used to correct the *P* value of Wilcoxon tests, removing false positives. We believe that this method can effectively display whether there are seasonal differences in bacterial taxa in our study.

References mentioned above:

Holmes ZC, Villa MM, Durand HK, Jiang S, Dallow EP, Petrone BL, Silverman JD, Lin PH, David LA. 2022. Microbiota responses to different prebiotics are conserved within individuals and associated with habitual fiber intake. *Microbiome*. 10, 114. doi: <http://doi.org/10.1186/s40168-022-01307-x>

Huang GP, Wang L, Li J, Hou R, Wang M, Wang ZL, Qu QY, Zhou WL, Nie YG, Hu Y, Ma Y, Yan L, Wei H, Wei F. 2022. Seasonal shift of the gut microbiome synchronizes host peripheral circadian rhythm for physiological adaptation to a low-fat diet in the giant panda. *Cell Rep*. 38: 110203. doi: <http://doi.org/10.1016/j.celrep.2021.110203>

Rocha FP, Ronque MUV, Lyra ML, Bacci MJ, Oliveira PS. 2022. Habitat and Host Species Drive the Structure of Bacterial Communities of Two Neotropical Trap-Jaw *Odontomachus Ants*. *Microb Ecol*. doi: <http://doi.org/10.1007/s00248-022-02064-y>

Also, I am concerned about how the authors interpret their tests. For example, at line 245-56 the authors report that there is no difference in T3 or T4 levels with season; yet then they go on to report microbial abundances that covaried with T3 or T4 levels. This does not make sense, since there is no variation to calculate correlations for. Note that L251 says the analysis revealed bacteria that had significant effects on T3 and T4, when in reality they only identified taxa whose abundance is correlated with T3 and T4.

We have carefully considered your suggestion. These statistical results show no conflict with

our theme. There were no significant seasonal differences in fecal T3 and T4 concentrations, and the relative abundances of most bacterial taxa did not vary significantly on the seasonal scale, and the Random Forest model indicated that only two low abundant taxa were associated with T3. Surely, we only identified the relative abundances of bacterial taxa correlated with T3 and T4, which should be mentioned without misinterpretation.

Language and interpretations: There are some issues with interpretations. For example, at L279-90, 283-4, 287-8, 292-6, 298, 299-302, 302-3, 303-4, 305-6, 311 and 312-4 (I don't list beyond here, but these represent many issues in a short span), the interpretations are overstated. In some instances I think this is because of language - for example, at L283-4 the authors are likely citing a previous finding, but it is phrased as a conclusion. Others, maybe also be languages, such as L 311 "gut microbiota varied significantly on smaller timescales [similar to humans]" the structure of the sentence implies that human microbiomes vary across small time scales but not large ones and cites papers that only support stability. I think the authors only intend those references to apply to the statement about long stability, but the sentence needs to be restructured to reflect this. Finally, for this issue about long-term stability and short-term variation I think this needs to be better explained in the text, as any short term variability must by definition mean there is long-term variability (but the cited studies are correct, the difference between the results just needs to be reconciled better in the text). Generally, I think the authors need to pay very close attention to the intro and discussion to make sure findings are not over interpreted.

We deeply appreciate for your comments. First, we have rewritten the original sentences of L279-L281, please see L280. Second, we are aware that the difference in physiological structure and diet between western lowland gorillas (*Gorilla gorilla*) and François's langurs mentioned by L281-L282, which may not be a strong support for our results, and we chose to delete the sentence.

It was previously suggested that the ratio of Firmicutes to Bacteroidetes (F/B) may be involved in energy absorption (Sun et al., 2019, Xia et al., 2021, Lai et al., 2023), but we reviewed more literatures and found that F/B presented different states in different species without a uniform presentation (Ley. et al., 2006, Turnbaugh et al., 2006, Filippo et al., 2010, Barelli et al., 2020), so we decided to delete L283-L288 in the discussion. At the same time, we reworded these sentences because of possible misrepresentation problems in the L292-296, L298-302, L302-303, L303-304, L305-306, Please see L284-294. And we have modified some sentences on the introduction and the discussion that may be poorly formulated.

Finally, we supplement the references with the L311-314, illustrating the elastic changes of the gut microbiota. For the long-term stability and short-term flexibility mentioned in this paper

are relative, the existing reports mostly also set long-term and short-term groups in their studies. Our study demonstrates that the gut microbiota of François' langurs exhibits significant shifts in the relative abundances of bacterial taxa, community structures, and alpha diversity during relatively short months, whereas those did not exhibit significant differences in relative abundances, community structures and alpha diversity over the relatively long seasonal scale.

References mentioned above:

Barelli, C, Albanese D, Stumpf RM, Asangba A, Donati C, Rovero F, Hauffe HC. 2020. The gut microbiota communities of wild arboreal and ground-feeding tropical primates are affected differently by habitat disturbance. *mSystems* 5: e00061-20. doi: <https://doi.org/10.1128/mSystems.00061-20>

Filippo CD, Cavalieri D, Paola MD, Ramazzotti M, Poullet JB, Massart S, Collini S, Pieraccini G, Lionetti P. 2010. Impact of diet in shaping gut microbiota revealed by a comparative study in children from Europe and rural Africa. *Proc Natl Acad Sci U S A*. 107: 14691-14696. doi: <https://doi.org/10.1073/pnas.1005963107>

Lai Y, Chen Y, Zheng J, Liu Z, Nong D, Liang J, Li Y, Huang Z. 2023. Gut microbiota of white-headed black langurs (*Trachypithecus leucocephalus*) in responses to habitat fragmentation. *Front Microbiol*. 14. doi: <http://doi.org/10.3389/fmicb.2023.1126257>

Ley RE, Turnbaugh PJ, Klein SI, Gordon JI. 2006. Human gut microbes associated with obesity. *Nature*. 444: 1022-1023. doi: <http://doi.org/10.1038/nature4441021a>

Sun YW, Sun YJ, Shi ZH., Liu ZS, Zhao C, Lu TF, Gao H, Zhu F, Chen R, Zhang J, Pan RL, Li BG, Teng LW, Guo ST. 2019. Gut microbiota of wild and captive alpine musk deer (*Moschus chrysogaster*). *Front Microbiol*. 10. doi: <http://doi.org/10.3389/fmicb.2019.03156>

Turnbaugh PJ, Ley RE, Mahowald MA, Magrini V, Mardis ER, Gordon JI. 2006. An obesity-associated gut microbiome with increased capacity for energy harvest. *Nature*. 444: 1027-1031. doi: <http://doi.org/10.1038/nature05414>

Xia T, Yao Y, Wang C, Dong M, Wu Y, Li D, Xie M, Ni Q, Zhang M, Xu H. 2021. Seasonal dynamics of gut microbiota in a cohort of wild Tibetan macaques (*Macaca thibetana*) in western China. *Glob Ecol Conserv*. 25: e01409. doi: <https://doi.org/10.1016/j.gecco.2020.e01409>

Use of supporting information: I think some the information in the supporting information could be moved to the main text to keep the reading clearer. Certainly there need to be more details in the methods (L121 - what are the kits? what are the per parameters? L123 - what software packages were used on the MajorBio Cloud platform, etc). Also, the authors' selection of figures adds some confusion to the manuscript. For example, there are no main text figures supporting measuring T3 or T4 levels, animal activity (these are in supplemental), and there is

no taxon plot showing the overall microbiomes of the samples. These changes may not be required, but I recommend the authors pay attention to them to ensure the manuscript is clear and direct.

We have carefully considered your suggestion. Firstly, we supplement the method description in the L121. Secondly, the DNA extraction, RCR amplification and sequencing of this study are provided by MajorBio Cloud platform, and the detailed steps are only listed in the supplementary information. Then, we put the monthly changes of T3 levels, T4 levels and activity budgets into the text, as shown in Fig.6 and Fig.9. Finally, Fig. 2 shows the main community structure of the gut microbiota of the langurs, and Table S4 and S5 show overall bacterial taxa at the phylum and family level.

April 21, 2023

Dr. Zhonghao Huang
Guangxi Normal University
Guilin
China

Re: Spectrum05091-22R1 (Evaluation of gut microbiota stability and flexibility as a response to seasonal variation in the wild François' langurs (*Trachypithecus francoisi*) in limestone forest)

Dear Dr. Zhonghao Huang:

Thank you for submitting your manuscript to Microbiology Spectrum. Two of the three previous reviewers have sent comments back, and both agree that you have addressed some, but not all of the previous issues in the manuscript. I believe that if you are willing to address these issues the manuscript would be suitable for acceptance, and I encourage you to consider submitting a revised version of the manuscript. If you choose to do so, please follow the guidelines below. I hope you find the feedback encouraging.

Link Not Available

Sincerely,

John Chaston

Journals Department
Reviewer comments:

Reviewer #2 (Comments for the Author):

This reviewer acknowledge the efforts undertaking to improve the manuscripts. However, there are additional questions.
1. Different sampling times and number can influence microbiome difference between two reports. But some results are hard to understand. For example, monthly microbiome results can represented showed significant variable microbiome composition

which were distributed within a certain range. At family level, Oscillospiraceae were significantly different (approximately 10% to 25% vs 30.21% {plus minus} 4.87%). Moreover, Eggerthellaceae was not identified in all months in this report but was fourth bacteria (8.24% {plus minus} 8.37%) in wild langur in the same place. You should explain for this difference especially Eggerthellaceae.

2. Energy metabolism is regulated by a complex interplay of hormones and other signaling molecules. Hormones play important roles in energy metabolism, including insulin, glucagon, cortisol, growth hormone as well as thyroid. Furthermore, fecal thyroid is not commonly used in clinical practice for evaluating metabolism as blood tests are the standard method for measuring thyroid hormones. Therefore, I can't trust the result that there was no metabolism variation according to seasonality based on only fecal thyroid hormone level. Also, sample source should be added for thyroid hormone evaluation. You should add the proof that only evaluation of fecal thyroid can reflect for metabolism in NHPs.

Reviewer #3 (Comments for the Author):

The authors' response has addressed some of my concerns in part, but I am concerned that the foundational concerns have not fully been addressed. My remaining concerns are below. Line numbers are based on the track-changes version. I apologize if my brief writing is terse - I am simply trying to be brief, and have not recommended exactly how the authors should make each change.

Language and interpretations. The authors continue to overstate their findings, including at L26-7, 29-30, 30-31, 35, and 35-7, 277 (need to rephrase as being correlated with, not affecting), 311-2, 313 (Bacteroides can play), 324-5, 327 (can efficiently), 330 (can help), 331 (may respond), 411 (no evidence), 419 (no evidence), 421 (may effectively), 421-2 (they did not increase to cope with the change, their response was correlated with dry season onset and their increase may have provided a coping mechanism to the animals), 425-6, 483-4, 499 (may assist), 530, and 532-7. All of these lines of text are over interpreted and must be rephrased. Two common issues are: 1) the authors conclude that microbes in the langurs are performing a function because some strains of the microbial group that strain belongs to have the function; this is inappropriate because of strain-level differences. 2) inferring that correlation is causation (because the microbes vary they cause the change or result from the change, for example, in season).

Additionally, I remain concerned about the authors' comparison between a dry and wet season. There is no replication for season, so it is impossible to discriminate if season versus any number of other possible covariates may underlie the differences in a few sampling groups versus a few other sampling groups. My main concern remains: the authors say there is no long term change but there is short term change. This is impossible. The human literature reconciles this in terms of a discussion of steady states (David 2014 Genome Biology comes to mind first), despite the idea that major perturbations, including changes in diets, medicine, or lifestyle may cause shifts between states. The authors do appear to hint at this, but none of it is explicit and it should be. Additionally, the interpretations are not consistent. At 224-9 and 425 there are seasonal differences, but at 337-8 there are no significant differences. The authors should be consistent and frame accordingly. I believe they are leveraging that there are no major differences in the five most abundant taxa, but this is not the same thing as saying there are no differences. Additionally, because season is not replicated, it is not clear to me that the authors have any power to conclude anything about season.

I continue to recommend that methods details be placed in the main text. Spectrum does not have space limitations, and there is insufficient detail in the methods section for readers to evaluate if the methods are appropriate.

Staff Comments:

Preparing Revision Guidelines

For complete guidelines on revision requirements, please see the journal Submission and Review Process requirements at

<https://journals.asm.org/journal/Spectrum/submission-review-process>. **Submissions of a paper that does not conform to Microbiology Spectrum guidelines will delay acceptance of your manuscript. "**

Please return the manuscript within 60 days; if you cannot complete the modification within this time period, please contact me. If you do not wish to modify the manuscript and prefer to submit it to another journal, please notify me of your decision immediately so that the manuscript may be formally withdrawn from consideration by Microbiology Spectrum.

Responses to Reviewers' Comments

NOTE: The reviewers' comments are in black, and our responses are in red. The line numbers in our responses are those in the clean version of current manuscript.

Associate Editor Comments to Author:

Thank you for submitting your manuscript to Microbiology Spectrum. Two of the three previous reviewers have sent comments back, and both agree that you have addressed some, but not all of the previous issues in the manuscript. I believe that if you are willing to address these issues the manuscript would be suitable for acceptance, and I encourage you to consider submitting a revised version of the manuscript. If you choose to do so, please follow the guidelines below. I hope you find the feedback encouraging.

Thanks for your comments. In the latest manuscript, we have made modifications based on the reviewer's suggestions.

Reviewer #2 (Comments for the Author):

This reviewer acknowledge the efforts undertaking to improve the manuscripts. However, there are additional questions.

1. Different sampling times and number can influence microbiome difference between two reports. But some results are hard to understood. For example, monthly microbiome results can represented showed significant variable microbiome composition which were distributed within a certain range. At family level, Oscillospiraceae were significant different (approximately 10% to 25% vs 30.21% {plus minus} 4.87%). Moreover, Eggerthellaceae was not identified in all months in this report but was fourth bacteria (8.24% {plus minus} 8.37%) in wild langur in the same place. You should explain for this difference especially Eggerthelaceae.

Thank you for your information. 1) Although both studies contain data from February 2019, the sampling dates, number of samples, and even research focus are different. Therefore, the relative abundance of Oscillospiraceae in February in our study was 21.91% \pm 7.23% and 30.21% \pm 4.87% in another study, which is understandable. Moreover, we want to explore the patterns of gut microbiota in the François' langurs at a seasonal scale, grouping is not on different dates within the same month, but rather on two seasons. 2) The previous preprint only had 17 sample data from February 2019, although Eggerthelaceae was the fourth largest group with a relative abundance of 8.24% \pm 8.37%. This shows significant intra group differences, and the relative abundance of Eggerthelaceae was 3.56% \pm 1.69%. In fact, this is still because

the two research focuses are different. We are exploring the temporal changes of gut microbiota in wild populations. The preprint compares the differences in gut microbiota composition between wild and captive populations during a certain period.

2. Energy metabolism is regulated by a complex interplay hormones and other signaling molecules. Hormones play important roles in energy metabolism, including insulin, glucagon, cortisol, growth hormone as well as thyroid. Furthermore, fecal thyroid is not commonly used in clinical practice for evaluating metabolism as blood tests are the standard method for measuring thyroid hormones. Therefore, I can't trust the result that there was no metabolism variation according to seasonality based on only fecal thyroid hormone level. Also, sample source should be added for thyroid hormone evaluation. You should add the proof that only evaluation of fecal thyroid can reflect for metabolism in NHPs.

Previous studies have used thyroid hormone to characterize the energy budget of wild animals (Kim, 2008, Thompson et al., 2017, Behringer et al., 2018, Gesquiere et al., 2018, Speakman et al., 2021, Touitou et al., 2021), including primates (Schaebs et al., 2016, Thompson et al., 2017, Gesquiere et al., 2018, Touitou et al., 2021). Actually, due to the difficulty in blood collection of wild François' langurs, as well as animal ethics, we are not able to do blood tests in clinical applications. Alternatively, we used fecal thyroid hormones as proxy for energy balance, which is adopted by previous studies and regarded as relative hormone values (Schaebs et al., 2016, Behringer et al., 2018, Speakman et al., 2021).

References mentioned above:

Thompson CL, Powell BL, Williams SH, Hanya G, Glander KE, Vinyard CJ. 2017. Thyroid hormone fluctuations indicate a thermoregulatory function in both a tropical (*Alouatta palliata*) and seasonally cold-habitat (*Macaca fuscata*) primate. *Am J Primatol* 79: e22714. <http://doi.org/10.1002/ajp.22714>

Behringer V, Deimel C, Hohmann G, Negrey J, Schaebs FS, Deschner T. 2018. Applications for non-invasive thyroid hormone measurements in mammalian ecology, growth, and maintenance. *Horm Behav* 105: 66-85. <http://doi.org/10.1016/j.yhbeh.2018.07.011>

Gesquiere LR, Pugh M, Alberts SC, Markham AC. 2018. Estimation of energetic condition in wild baboons using fecal thyroid hormone determination. *Gen Comp Endocrinol* 260: 9-17. <http://doi.org/10.1016/j.ygcen.2018.02.004>

Speakman JR, Chi Q, Oldakowski L, Fu H, Fletcher QE, Hambly C, Togo J, Liu X, Piernney SB, Wang X, Zhang L, Redman P, Wang L, Tang G, Li Y, Cui J, Thomson PJ, Wang Z, Glover P, Robertson OC, Zhang Y, Wang D. 2021. Surviving winter on the Qinghai-Tibetan Plateau: Pikas suppress energy demands and exploit yak feces to survive winter. *Proc Natl Acad Sci U S A* 118.

<http://doi.org/10.1073/pnas.2100707118>

Touitou S, Heistermann M, Schulke O, Ostner J. 2021. Triiodothyronine and cortisol levels in the face of energetic challenges from reproduction, thermoregulation and food intake in female macaques. *Horm Behav* 131: 104968. <http://doi.org/10.1016/j.yhbeh.2021.104968>

Kim B. 2008. Thyroid hormone as a determinant of energy expenditure and the basal metabolic rate. *Thyroid* 18: 141-144. <http://doi.org/10.1089=thy.2007.0266>

Schaebbs FS, Wolf TE, Behringer V, Deschner T. 2016. Fecal thyroid hormones allow for the noninvasive monitoring of energy intake in capuchin monkeys. *J Endocrinol* 231: 1-10. <http://doi.org/10.1530/JOE-16-0152>

Reviewer #3 (Comments for the Author):

The authors' response has addressed some of my concerns in part, but I am concerned that the foundational concerns have not fully been addressed. My remaining concerns are below. Line numbers are based on the track-changes version. I apologize if my brief writing is terse - I am simply trying to be brief, and have not recommended exactly how the authors should make each change.

Language and interpretations. The authors continue to overstate their findings, including at L26-7, 29-30, 30-31, 35, and 35-7, 277 (need to rephrase as being correlated with, not affecting), 311-2, 313 (Bacteroides can play), 324-5, 327 (can efficiently), 330 (can help), 331 (may respond), 411 (no evidence), 419 (no evidence), 421 (may effectively), 421-2 (they did not increase to cope with the change, their response was correlated with dry season onset and their increase may have provided a coping mechanism to the animals), 425-6, 483-4, 499 (may assist), 530, and 532-7. All of these lines of text are over interpreted and must be rephrased. Two common issues are: 1) the authors conclude that microbes in the langurs are performing a function because some strains of the microbial group that strain belongs to have the function; this is inappropriate because of strain-level differences. 2) inferring that correlation is causation (because the microbes vary they cause the change or result from the change, for example, in season).

Many thanks for your comments. 1) We have revised or rewritten these sentences. 2) In addition, we also attempted to explore the function of gut microbiota after isolating strains, but the realistic sampling difficulties did not allow us to do such analysis, requiring higher precision sequencing technology. 3) Actually, some articles describe that diet "affects" the gut microbiota of host (Leshem et al., 2020, Huang et al., 2022, Bourdeau-Julien et al., 2023), but your suggestion is a more rigorous expression that we accept and modify.

References mentioned above:

Leshem A, Segal E, Elinav E. 2020. The gut microbiome and individual-specific responses to

diet. *mSystems* 5: e00665-20. <http://doi.org/10.1128/mSystems.00665-20>

Bourdeau-Julien I, Castonguay-Paradis S, Rochefort G, Perron J, Lamarche B, Flamand N, Di Marzo V, Veilleux A, Raymond F. 2023. The diet rapidly and differentially affects the gut microbiota and host lipid mediators in a healthy population. *Microbiome* 11: 26. <http://doi.org/10.1186/s40168-023-01469-2>

Huang GP, Wang L, Li J, Hou R, Wang M, Wang ZL, Qu QY, Zhou WL, Nie YG, Hu Y, Ma Y, Yan L, Wei H, Wei F. 2022. Seasonal shift of the gut microbiome synchronizes host peripheral circadian rhythm for physiological adaptation to a low-fat diet in the giant panda. *Cell Rep* 38: 110203. <http://doi.org/10.1016/j.celrep.2021.110203>

Additionally, I remain concerned about the authors' comparison between a dry and wet season. There is no replication for season, so it is impossible to discriminate if season versus any number of other possible covariates may underlie the differences in a few sampling groups versus a few other sampling groups. My main concern remains: the authors say there is no long term change but there is short term change. This is impossible. The human literature reconcile this in terms of a discussion of steady states (David 2014 *Genome Biology* comes to mind first), despite the idea that major perturbations, including changes in diets, medicine, or lifestyle may cause shifts between states. The authors do appear to hint at this, but none of it is explicit and it should be. Additionally, the interpretations are not consistent. At 224-9 and 425 there are seasonal differences, but at 337-8 there are no significant differences. The authors should be consistent and frame accordingly. I believe they are leveraging that there are no major differences in the five most abundant taxa, but this is not the same thing as saying there are no differences. Additionally, because season is not replicated, it is not clear to me that the authors have any power to conclude anything about season.

Thank you for your comments. We admitted that there should be weaknesses in current study (as you pointed), however, this is the first reports on the seasonal variations in the gut microbiota of these karst-dwelling langurs and should provide general pattern in the seasonality in their gut microbiota. We added these into the discussion section, please see line L414-7. 1) We acknowledge the limitations of this study, with incomplete sample data spanning 16 months. Current study could reflect the compositional structure and basic seasonal regularity of the gut microbiota in the François' langurs, which has a similar setting in other studies (Smits et al., 2017, Wu et al., 2017, Sun et al., 2018, Orkin et al., 2019, Xia et al., 2021). 2). We describe nonsignificant seasonal changes in relative abundance of bacterial taxa (relative stability) and significant changes between months (relative short-term resilience), which we speculate is related to the high leaf feeding of these langurs in previous behavioral ecology studies. Such features are exhibited in the gut microbiota of humans. 3) Finally, we

corrected for the previously mentioned significant seasonal variations in L224-9 and L425, and insignificant differences in L337-8, We expressed statistically significant seasonal variations in the relative abundances of bacterial taxa, but variable bacterial community structures at the family level. Please see L340-2.

References mentioned above:

Smits SA, Leach J, Sonnenburg ED, Gonzalez CG, Lichtman JS, Reid G, Knight R, Manjurano A, Chungalucha J, Elias JE, Dominguez-Bello MG, Sonnenburg JL. 2017. Seasonal cycling in the gut microbiome of the Hadza hunter-gatherers of Tanzania. *Science* 357: 802-806. <http://doi.org/10.1126/science.aan4834>

Wu Q, Wang X, Ding Y, Hu Y, Nie Y, Wei W, Ma S, Yan L, Zhu L, Wei F. 2017. Seasonal variation in nutrient utilization shapes gut microbiome structure and function in wild giant pandas. *Proc Biol Sci* 284: 20170955. <http://dx.doi.org/10.1098/rspb.2017.0955>

Sun B, Gu Z, Wang X, Huffman MA, Garber PA, Sheeran LK, Zhang D, Zhu Y, Xia D-P, Li J-h. 2018. Season, age, and sex affect the fecal mycobiota of free-ranging Tibetan macaques (*Macaca thibetana*). *Am J Primatol* 80: e22880. <https://doi.org/10.1002/ajp.22880>

Orkin JD, Campos FA, Myers MS, Cheves Hernandez SE, Guadamuz A, Melin AD. 2019. Seasonality of the gut microbiota of free-ranging white-faced capuchins in a tropical dry forest. *ISME J* 13: 183-196. <https://doi.org/10.1038/s41467-019-10191-3>

Xia T, Yao Y, Wang C, Dong M, Wu Y, Li D, Xie M, Ni Q, Zhang M, Xu H. 2021. Seasonal dynamics of gut microbiota in a cohort of wild Tibetan macaques (*Macaca thibetana*) in western China. *Glob Ecol Conserv* 25: e01409. <https://doi.org/10.1016/j.gecco.2020.e01409>

I continue to recommend that methods details be placed in the main text. Spectrum does not have space limitations, and there is insufficient detail in the methods section for readers to evaluate if the methods are appropriate.

Thanks for your comments. We put the methods in the supplementary material into the main text.

May 24, 2023

Dr. Zhonghao Huang
Guangxi Normal University
Guilin
China

Re: Spectrum05091-22R2 (Evaluation of gut microbiota stability and flexibility as a response to seasonal variation in the wild François' langurs (*Trachypithecus francoisi*) in limestone forest)

Dear Dr. Zhonghao Huang:

Thank you for submitting your manuscript to Microbiology Spectrum. As you will see your paper is very close to acceptance. Please modify the manuscript along the lines reviewer 2 has recommend. As these revisions are quite minor, I expect that you should be able to turn in the revised paper in less than 30 days, if not sooner. If your manuscript was reviewed, you will find the reviewers' comments below.

When submitting the revised version of your paper, please provide (1) point-by-point responses to the issues raised by the reviewers as file type "Response to Reviewers," not in your cover letter, and (2) a PDF file that indicates the changes from the original submission (by highlighting or underlining the changes) as file type "Marked Up Manuscript - For Review Only". Please use this link to submit your revised manuscript. Detailed instructions on submitting your revised paper are below.

Link Not Available

Sincerely,

John Chaston

Reviewer comments:

Reviewer #2 (Comments for the Author):

The authors have addressed the comments and I recommend acceptance of the manuscript

Reviewer #3 (Comments for the Author):

All line numbers are from the tracked changes version. the authors have partially addressed my concerns, but I think there is some room for further revision.

Language and interpretations. I think the authors have generally addressed this concern, although some issues remain, for example L 288 it says relative humidity has the largest effect, but it should be rephrased as 'explains the most variation'. Additionally, there are no experiments to support conjecture at L 29-32 and while such statements could be included in the discussion they are inappropriate for the abstract.

Season. The text at 375-84 remains inappropriate and needs to be revised. There cannot be variation on short periods but not on long periods since the long periods include the short periods. The text does an insufficient job of explaining the subtleties here and needs to be made very explicit and clear. I do wonder if using an ASV-based approach might resolve some of the

discrepancy (see comment below), and I would recommend that the authors re-analyze their data using an ASV- rather than OTU-based approach; however, this may not be necessary based on how the authors can respond to this concern below.

Methods. The additional methods are helpful in some cases but remain incomplete. For example:

L146 - the authors should state how the PCR was performed, which primers were used, and how barcodes and Illumina adapters were added

L146 - please explained what 'qualified' PCR products are. Also, there are usually steps between PCR and submitting for sequencing (sometimes not if the company did the work for you)

L147 - it is not clear which software packages the authors used to filter the data. maybe the microbiome package on major bio? which executables? I'm unfamiliar with this platform, but there must be some level of detail they can explain.

L158 - please rephrase what are "non-repetitive sequences in repetitive sequences"

L163-4 - I don't understand this line. please rephrase "equaling all sequences according to the minimum sequence",

L180 - train > training?

L165 - you say you couldn't refine most OTUs to the genus level, but then you report on genus and species levels at L246; this seems inconsistent but I may be misunderstanding, please add some text to help explain the distinction

The authors' additional details show that they are using an OTU instead of an ASV clustering method. The authors may find differences in their results if they used an ASV method, which I think is generally more common and widely accepted in recent years. This is not an objective requirement, but if they wish to do so, it may help provide insight into the short- vs. long timescales issue.

Preparing Revision Guidelines

Please return the manuscript within 60 days; if you cannot complete the modification within this time period, please contact me. If you do not wish to modify the manuscript and prefer to submit it to another journal, please notify me of your decision immediately so that the manuscript may be formally withdrawn from consideration by Microbiology Spectrum.

Responses to Reviewers' Comments

NOTE: The reviewers' comments are in black, and our responses are in red. The line numbers in our responses are those in the clean version of current manuscript.

Associate Editor Comments to Author:

Thank you for submitting your manuscript to Microbiology Spectrum. As you will see your paper is very close to acceptance. Please modify the manuscript along the lines reviewer 2 has recommend. As these revisions are quite minor, I expect that you should be able to turn in the revised paper in less than 30 days, if not sooner. If your manuscript was reviewed, you will find the reviewers' comments below.

Thanks for your comments. In the clean manuscript, we have made modifications based on the reviewer's suggestions.

Reviewer #3 (Comments for the Author):

All line numbers are from the tracked changes version. the authors have partially addressed my concerns, but I think there is some room for further revision.

Language and interpretations. I think the authors have generally addressed this concern, although some issues remain, for example L 288 it says relative humidity has the largest effect, but it should be rephrased as 'explains the most variation'. Additionally, there are no experiments to support conjecture at L 29-32 and while such statements could be included in the discussion they are inappropriate for the abstract.

Thank you very much, we have made the corresponding modification, please see L172.

Season. The text at 375-84 remains inappropriate and needs to be revised. There cannot be variation on short periods but not on long periods since the long periods include the short periods. The text does an insufficient job of explaining the subtleties here and needs to be made very explicit and clear. I do wonder if using an ASV-based approach might resolve some of the discrepancy (see comment below), and I would recommend that the authors re-analyze their data using an ASV- rather than OTU-based approach; however, this may not be necessary based on how the authors can respond to this concern below.

Thanks a lot. We have decided to retain the analysis on the seasonal scale and remove the analysis between months, which is more in line with the theme of exploring the

seasonal patterns of the gut microbiota of these langurs. In fact, there are still studies using OTU clustering (Huang et al., 2022, Chen et al., 2023, Gan et al., 2023, Zhou et al., 2023), and ASV clustering sequencing is more accurate, but it may also mistakenly remove sequences with low abundance. Both clustering methods have their own advantages and disadvantages. We are considering combining ASV clustering, OTU clustering, metagenomics and other technologies to explore the structure and function of the gut microbiota of François's langurs.

References mentioned above:

Chen, Q., Zhang, X., Shi, W., Du, X., Ma, L., Wang, W., et al. 2023. Longitudinal investigation of enteric virome signatures from parental-generation to offspring pigs. *Microbiology Spectrum*. doi: <http://doi.org/10.1128/spectrum.00023-23>.

Gan, L., Feng, Y., Du, B., Fu, H., Tian, Z., Xue, G., et al. 2023. Bacteriophage targeting microbiota alleviates non-alcoholic fatty liver disease induced by high alcohol-producing *Klebsiella pneumoniae*. *Nat Commun.* 14, 3215. doi: <http://doi.org/10.1038/s41467-023-39028-w>.

Huang, G. P., Wang, L., Li, J., Hou, R., Wang, M., Wang, Z. L., et al. 2022. Seasonal shift of the gut microbiome synchronizes host peripheral circadian rhythm for physiological adaptation to a low-fat diet in the giant panda. *Cell Rep.* 38, 110203. doi: <http://doi.org/10.1016/j.celrep.2021.110203>.

Zhou, Z., Tran, P. Q., Adams, A. M., Kieft, K., Breier, J. A., Fortunato, C. S., et al. 2023. Sulfur cycling connects microbiomes and biogeochemistry in deep-sea hydrothermal plumes. *ISME J.* doi: <http://doi.org/10.1038/s41396-023-01421-0>.

Methods. The additional methods are helpful in some cases but remain incomplete.

For example:

L146 - the authors should state how the PCR was performed, which primers were used, and how barcodes and Illumina adapters were added.

We have added the PCR operation, see L128-136. barcodes help filter sequences in L146-147 and Illumina adapters help build library in L133-138.

L146 - please explained what 'qualified' PCR products are. Also, there are usually steps between PCR and submitting for sequencing (sometimes not if the company did the work for you)

Quantitative PCR product, more than 0.5ng/ul was qualified, see L135-136. Libraries were built following PCR using the NEXTFLEX Rapid DNA-Seq Kit, see L136-139, which was provided by the company.

L147 - it is not clear which software packages the authors used to filter the data. maybe the microbiome package on major bio? which executables? I'm unfamiliar with this platform, but there must be some level of detail they can explain.

We used Fastp and FLASH to filter the original sequence and splicing sequence, please see L141-143

L158 - please rephrase what are "non-repetitive sequences in repetitive sequences"

We have corrected it, please see L148

L163-4 - I don't understand this line. please rephrase "equaling all sequences according to the minimum sequence",

We used the minimum sequence number to unify the sequence number of all samples, and then carried out subsequent analysis. Please see L152-153

l180 - train > training?

We corrected it, please see L168.

L165 - you say you couldn't refine most OTUs to the genus level, but then you report on genus and species levels at L246; this seems inconsistent but I may be misunderstanding, please add some text to help explain the distinction

The sequencing results can determine the number of taxa at the species level and genus level, but there are also many unranked or unclassified bacterial taxa, so, we analyzed the bacterial taxa at the phylum level and family level. Let's put it another way, see L154-155

The authors' additional details show that they are using an OTU instead of an ASV clustering method. The authors may find differences in their results if they used an ASV method, which I think is generally more common and widely accepted in recent years. This is not an objective requirement, but if they wish to do so, it may help provide insight into the short- vs. long timescales issue.

Thank you very much. Both OTU clustering and ASV clustering have their own advantages and disadvantages, and both can reflect the composition characteristics of the microbial community of the research objects to a certain extent. In the subsequent analysis, we can combine ASV clustering, metagenomics and metabolomics to explore the structure and function of the gut microbiota of François's langurs.

June 10, 2023

Dr. Zhonghao Huang
Guangxi Normal University
Guilin
China

Re: Spectrum05091-22R3 (Evaluation of gut microbiota stability and flexibility as a response to seasonal variation in the wild François' langurs (*Trachypithecus francoisi*) in limestone forest)

Dear Dr. Zhonghao Huang:

Your manuscript has been accepted, and I am forwarding it to the ASM Journals Department for publication. You will be notified when your proofs are ready to be viewed.

Sincerely,

John Chaston
Editor, Microbiology Spectrum
